# Bayesian Neural Networks for Functional ANOVA model

**Seokhun Park**[1*], **Choeun Kim**[1*], **Jihu Lee**[1], **Yunseop Shin**[1], **Insung Kong**[2], **Yongdai Kim**[1†]

[1]Department of Statistics, Seoul National University
[2]Department of Applied Mathematics, University of Twente
`{shrdid, kimchoeun, rieky0426, dbstjq48}@snu.ac.kr,`
`insung.kong@utwente.nl, ydkim0903@gmail.com`

## ABSTRACT

With the increasing demand for interpretability in machine learning, functional ANOVA decomposition has gained renewed attention as a principled tool for breaking down high-dimensional function into low-dimensional components that reveal the contributions of different variable groups. Recently, Tensor Product Neural Network (TPNN) has been developed and applied as basis functions in the functional ANOVA model, referred to as ANOVA-TPNN. A disadvantage of ANOVA-TPNN, however, is that the components to be estimated must be specified in advance, which makes it difficult to incorporate higher-order TPNNs into the functional ANOVA model due to computational and memory constraints. In this work, we propose Bayesian-TPNN, a Bayesian inference procedure for the functional ANOVA model with TPNN basis functions, enabling the detection of higher-order components with reduced computational cost compared to ANOVA-TPNN. We develop an efficient MCMC algorithm and demonstrate that Bayesian-TPNN performs well by analyzing multiple benchmark datasets. Theoretically, we prove that the posterior of Bayesian-TPNN is consistent.

## 1 INTRODUCTION

As artificial intelligence (AI) models become increasingly complex, the demand for interpretability has grown accordingly. To address this need, various interpretable models—including both post-hoc explanations (Ribeiro et al., 2016; Lundberg & Lee, 2017) and inherently transparent models (Agarwal et al., 2021; Koh et al., 2020; Radenovic et al., 2022; Park et al., 2025)—have been studied. Among various interpretable approaches, our study focuses on the functional ANOVA model, a particularly important class of interpretable models that decompose a high-dimensional function into a sum of low-dimensional functions called *componenets* or *interactions*. Notable examples of the functional ANOVA model are the generalized additive Model (Hastie & Tibshirani, 1986), SS-ANOVA (Gu & Wahba, 1993) and MARS (Friedman, 1991). Because complex structures of a given high-dimensional model can be understood by interpreting low-dimensional components, the functional ANOVA models have been extensively used in interpretable AI applications (Lengerich et al., 2020; Märtens & Yau, 2020; Choi et al., 2025; Herren & Hahn, 2022).

In recent years, various neural networks have been developed to estimate components in the functional ANOVA model. Neural Additive Models (NAM, Agarwal et al. (2021)) estimates each component of the functional ANOVA model using deep neural networks (DNN), and Neural Basis Models (NBM, Radenovic et al. (2022)) significantly reduce the computational burden of NAM by using basis deep neural networks (DNN). NODE-GAM (Chang et al., 2021) can select and estimate the components in the functional ANOVA model simultaneously, and Thielmann et al. (2024) proposes NAMLSS, which modifies NAM to estimate the predictive distribution. Park et al. (2025) proposes ANOVA-TPNN, which estimates the components under the uniqueness constraint and thus provides a stable estimate of each component.

---

[*]Equal contribution.
[†]Corresponding author.

Existing neural-network approaches to functional ANOVA model require prohibitive computation when the input dimension $p$ is large, because the number of components—and thus the required networks—grows exponentially. As a result, only 1–2 dimensional components are typically used, yielding suboptimal prediction when higher-order interactions matter.

In this paper, we propose a Bayesian neural network (BNN) for the functional ANOVA model which can estimate higher-order interactions (i.e., components whose input dimension is greater than 2) without requiring huge amounts of computing resources. *The main idea of the proposed BNN is to infer the architecture (the architectures of neural networks for each component) as well as the parameters (the weights and biases in each neural network). To explore higher posterior regions of the architecture, a specially designed MCMC algorithm is developed which searches the architectures in a stepwise manner (i.e., growing or pruning the current architecture) and thus huge computing resources for memorizing and processing all of the predefined neural networks for the components can be avoided.*

Bayesian Neural Networks (BNN; MacKay (1992); Neal (2012); Wilson & Izmailov (2020); Izmailov et al. (2021)) provide a principled Bayesian framework for training DNNs and have received considerable attention in machine learning and AI. Compared to frequentist approaches, BNN offers stronger generalization and better-calibrated uncertainty estimates (Wilson & Izmailov, 2020; Izmailov et al., 2021), which enhance decision making. These properties have motivated applications in areas such as recommender systems (Wang et al., 2015), topic modeling (Gan et al., 2015), and medical diagnosis (Filos et al., 2019). More recently, Bayesian neural networks (BNN) that learn their own architectures have been actively studied. In particular, Kong et al. (2023) introduced a node-sparse BNN, referred to as the masked BNN (mBNN), and established its theoretical properties. Nguyen et al. (2024) proposes S-RJMCMC, which explores architectures and weights by jointly sampling parameters and altering the number of nodes.

This is the first work on BNN that efficiently estimates higher-order components in the functional ANOVA model without requiring substantial computing resources. Our main contributions can be outlined as follows.

- We propose a BNN for the functional ANOVA model called Bayesian-TPNN which treats the architecture as a learnable parameter, and develop an MCMC algorithm which efficiently explores high-posterior regions of the architecture.
- For theoretical justifications of the proposed BNN, we prove the posterior consistency of the prediction model as well as each component.
- Through experiments on multiple real datasets, we show that the proposed BNN provides more accurate and stable estimation and uncertainty quantification than other neural networks for the functional ANOVA model. On various synthetic datasets, we further show that Bayesian-TPNN effectively estimates important higher-order components.

## 2 PRELIMINARIES

### 2.1 NOTATION

Let $\mathbf{x} = (x_1, \ldots, x_p)^\top \in \mathcal{X}$ be a $p$-dimensional input vector, where $\mathcal{X} = \mathcal{X}_1 \times \cdots \times \mathcal{X}_p \subseteq [0,1]^p$. We write $[p] = \{1, \ldots, p\}$ and its power set with cardinality $d$ as power$([p], d)$. For any component $S \subseteq [p]$, we denote $\mathbf{x}_S = (x_j, j \in S)^\top$ and define $\mathcal{X}_S = \prod_{j \in S} \mathcal{X}_j$. A function defined on $\mathcal{X}_S$ is denoted by $f_S$. For any real-valued function $f : \mathcal{X} \to \mathbb{R}$, we define the empirical $\ell_2$-norm as $\|f\|_{2,n} := (\sum_{i=1}^n f(\mathbf{x}_i)^2/n)^{1/2}$, where $\mathbf{x}_1, \ldots, \mathbf{x}_n$ are observed input vectors. We denote $\sigma(\cdot)$ as the sigmoid function, i.e., $\sigma(x) := 1/(1 + \exp(-x))$. We denote by $\mu_n$ the empirical distribution of $\{\mathbf{x}_1, \ldots, \mathbf{x}_n\}$, and by $\mu_{n,j}$ the marginal distribution of $\mu_n$ on $\mathcal{X}_j$.

### 2.2 PROBABILITY MODEL FOR THE LIKELIHOOD

We consider a nonparametric regression model in which the conditional distribution of $Y_i$ given $\mathbf{x}_i$ follows an exponential family (Brown et al., 2010; Chen, 2024):

$$Y_i | \mathbf{x}_i \sim \mathbb{Q}_{f(\mathbf{x}_i), \eta} \tag{1}$$

for $i = 1, ..., n$, where $f : \mathcal{X} \to \mathbb{R}$ is a regression function and $\eta$ is a nuisance parameter. Here, we assume that $\mathbb{Q}_{f(\mathbf{x}),\eta}$ admits the density function $q_{f(\mathbf{x}),\eta}$ defined as

$$q_{f(\mathbf{x}),\eta}(y) = \exp\left(\frac{f(\mathbf{x})y - A(f(\mathbf{x}))}{\eta} + S(y, \eta)\right), \tag{2}$$

where $A(\cdot)$ is the log-partition function, ensuring that the density integrates to one. We assume that each input vector $\mathbf{x}_i$ has been rescaled, yielding $\mathbf{x}_i \in [0, 1]^p$ for $i = 1, ..., n$.

**Example 1. Gaussian regression model:** Consider the gaussian regression $Y = f(\mathbf{x}) + \epsilon$, where $\epsilon \sim N(0, \sigma_\epsilon^2)$. In this case, the density in (2), corresponds to $A(f(\mathbf{x})) := f(\mathbf{x})^2/2$ and $S(y, \eta) := -y^2/2\eta - (\log 2\pi\eta)/2$ with $\eta = \sigma_\epsilon^2$.

**Example 2. Logistic regression model:** For a binary outcome $Y \in \{0, 1\}$, consider the logistic regression model $Y|\mathbf{x} \sim \text{Bernoulli}(\sigma(f(\mathbf{x})))$. In this case, there is no nuisance parameter, i.e., $\eta = 1$. This distribution can be expressed as the exponential family with $A(f(\mathbf{x})) := \log(1 + e^{f(\mathbf{x})})$ and $S(y, \eta) := 0$.

**Likelihood:** Let $\mathcal{D}^{(n)} = \{(\mathbf{x}_1, y_1), \ldots, (\mathbf{x}_n, y_n)\}$ be given data which consist of $n$ pairs of observed input vectors and response variables. For the likelihood, we assume that $y_i$s are independent realizations of $Y_i|\mathbf{x}_i \sim \mathbb{Q}_{f(\mathbf{x}_i),\eta}$, where $f$ and $\eta$ are the parameters to be inferred.

## 2.3 Functional ANOVA model

For $S \subseteq [p]$, we say that $f_S$ satisfies the sum-to-zero condition with respect to a probability measure $\mu$ on $\mathcal{X}$ if

$$\text{For} \quad S \subseteq [p], \ \forall j \in S \text{ and } \forall \mathbf{x}_{S \setminus \{j\}} \in \mathcal{X}_{S \setminus \{j\}}, \ \int_{\mathcal{X}_j} f_S(\mathbf{x}_S)\mu_j(dx_j) = 0 \tag{3}$$

holds, where $\mu_j$ is the marginal probability measure of $\mu$ on $\mathcal{X}_j$.

**Theorem 2.1** (Functional ANOVA Decomposition (Hooker, 2007; Owen, 2013)). *Any real-valued function $f$ defined on $\mathbb{R}^p$ can be uniquely decomposed as*

$$f(\mathbf{x}) = \sum_{S \subseteq [p]} f_S(\mathbf{x}_S), \tag{4}$$

*almost everywhere with respect to $\Pi_{j=1}^p \mu_j$, where each component $f_S$ satisfies the sum-to-zero condition with respect to $\mu$.*

Theorem 2.1 guarantees a unique decomposition of any real-valued multivariate function $f$ into the components satisfying the sum-to-zero condition with respect to the probability measure $\mu$. In (4), we refer to $f_S$ as main effects when $|S| = 1$, as second-order interactions when $|S| = 2$, and so on. For brevity, we use the empirical distribution $\mu_n$ for $\mu$ when referring to the sum-to-zero condition.

## 2.4 Tensor Product Neural Networks

In this subsection, we review Tensor Product Neural Network (TPNN) proposed by Park et al. (2025) since we use it as a building block of our proposed BNN. TPNN is a specially designed neural network to satisfy the sum-to-zero condition.

For each $S \subseteq [p]$, TPNN is defined as $f_S(\mathbf{x}_S) = \sum_{k=1}^{K_S} \beta_{S,k}\phi(\mathbf{x}_S|S, \mathfrak{B}_{S,k}, \mathfrak{R}_{S,k})$ for component $f_S$, where $\beta_{S,k} \in \mathbb{R}$, $\mathfrak{B}_{S,k} = (b_{S,j,k}, j \in S) \in \mathbb{R}^{|S|}$, and $\mathfrak{R}_{S,k} = (\gamma_{S,j,k}, j \in S) \in (0, \infty)^{|S|}$. Here, $\phi(\mathbf{x}_S|S, \mathfrak{B}_{S,k}, \mathfrak{R}_{S,k})$ is defined as

$$\phi(\mathbf{x}_S|S, \mathfrak{B}_{S,k}, \mathfrak{R}_{S,k}) := \prod_{j \in S}\left(1 - \sigma\left(\frac{x_j - b_{S,j,k}}{\gamma_{S,j,k}}\right) + c_j(b_{S,j,k}, \gamma_{S,j,k})\sigma\left(\frac{x_j - b_{S,j,k}}{\gamma_{S,j,k}}\right)\right), \tag{5}$$

where

$$c_j(b,\gamma) := -\left(1 - \int_{\mathcal{X}_j} \sigma\left(\frac{x_j - b}{\gamma}\right)\mu_{n,j}(dx_j)\right)\bigg/\int_{\mathcal{X}_j}\sigma\left(\frac{x_j - b}{\gamma}\right)\mu_{n,j}(dx_j). \quad (6)$$

The term $c_j(b,\gamma)$ is introduced to make $\phi(\mathbf{x}_S|S, \mathfrak{B}_{S,k}, \mathfrak{R}_{S,k})$ satisfy the sum-to-zero condition. Finally, Park et al. (2025) proposes ANOVA-T$^d$PNN, which assumes that:

$$f(\mathbf{x}) = \sum_{S\subseteq[p],|S|\leq d}\sum_{k=1}^{K_S}\beta_{S,k}\phi(\mathbf{x}_S|S,\mathfrak{B}_{S,k},\mathfrak{R}_{S,k}), \quad (7)$$

where $d \in \mathbb{N}_+$ and $\{K_S, S \subseteq [p], |S| \leq d\}$ are hyperparameters. Since $\phi(\cdot|S,\mathfrak{B}_{S,k},\mathfrak{R}_{S,k})$ satisfies the sum-to-zero condition for any $S \subseteq [p]$, $f_{\text{ANOVA-T}^d\text{PNN}}$ also satisfies the sum-to-zero condition. Therefore, we can estimate the components uniquely by estimating the parameters in ANOVA-T$^d$PNN.

Here, $d$ is the maximum order of components. Note that as the maximum order $d$ increases, the number of TPNNs in (7) grows exponentially; therefore, in practice $d$ is set to 1 or 2 due to the limitation of computing resources. In addition, choosing $K_S$s is not easy. To further illustrate these limitations, the experiments on the runtime of Bayesian-TPNN and ANOVA-T$^2$PNN are presented in Section G of Appendix.

## 3 BAYESIAN TENSOR PRODUCT NEURAL NETWORKS

In (7), instead of fixing $S$, we treat $S$ also as learnable parameters. That is, we consider the following model:

$$f(\mathbf{x}) = \sum_{k=1}^{K}\beta_k\phi(\mathbf{x}|\Theta_k), \quad (8)$$

where $\Theta_k = (S_k, \mathbf{b}_{S_k,k}, \Gamma_{S_k,k})$, $S_k \subseteq [p]$, and aim to learn $K$ and $(S_k, k \in [K])$ as well as the other parameters. Here,

$$\mathbf{b}_{S_k,k} := (b_{j,k}, j \in S_k) \in [0,1]^{|S_k|},$$
$$\Gamma_{S_k,k} := (\gamma_{j,k}, j \in S_k) \in (0,\infty)^{|S_k|}.$$

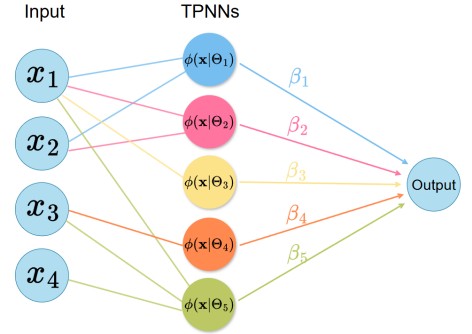

Figure 1: **Bayesian-TPNN with** $p = 4$ **and** $K = 5$.

for $k \in [K]$. *Note that $K$ and $S_k$ are considered to be the parameters defining the architecture, but they cannot be updated by a gradient descent algorithm since $K$ and $S_k$s are not numeric parameters. Instead, we adopt a Bayesian approach in which $K$ and $S_k$s are explored via an MCMC algorithm.* We refer to the resulting model as *Bayesian Tensor Product Neural Networks (Bayesian-TPNN)*. Bayesian-TPNN can be understood as an edge-sparse shallow neural network when $K$ is the number of hidden nodes and $S_K$ is the set of input variables linked to the $k$-th hidden node through active edges. See Figure 1 for an illustration.

### 3.1 PRIOR

The parameters in Bayesian-TPNN consist of $K$, $\mathcal{B}_K := (\beta_1, ..., \beta_K)$, $\mathbf{S}_K := (S_k, k \in [K])$, $\mathbf{b}_{\mathbf{S}_K,K} := (\mathbf{b}_{S_k,k}, k \in [K])$, $\Gamma_{\mathbf{S}_K,K} := (\Gamma_{S_k,k}, k \in [K])$ and the nuisance parameter $\eta$ if it exists (e.g. the variance of the noise in the gaussian regression model). The parameters can be categorized into the three groups: (1) $K$ for the node-sparsity, (2) $S_k, k = 1, ..., K$ for the edge sparsity, and (3) all the other parameters including $(\mathbf{b}_{S_k,k}, \Gamma_{S_k,k}, k = 1, ..., K)$. We use a hierarchical prior for these three groups of parameters.

**Prior for $K$:** We consider the following prior distribution for $K$:

$$\pi(K = k) \propto \exp(-C_0 k \log n), \quad \text{for} \quad k = 0, ..., K_{\max}, \quad (9)$$

where $K_{\max} \in \mathbb{N}_+$ and $C_0 > 0$ are hyperparameters. This prior is motivated by Kong et al. (2023).

**Prior for $\mathbf{S}_K | K$:** Conditional on $K$, we assume a prior that $S_k$s are independent and each $S_k$ follows the mixture distribution:

$$\sum_{d=1}^{p} w_d \text{Uniform}\big(\text{power}([p], d)\big), \tag{10}$$

where $w_d$s are defined recursively as follows: $w_d \propto \big(1 - p_{\text{adding}}(d)\big) \prod_{\ell < d} p_{\text{adding}}(\ell)$ with $p_{\text{adding}}(\ell) := \alpha_{\text{adding}}(1 + \ell)^{-\gamma_{\text{adding}}}$. Here, $p_{\text{adding}}$ is the probability of adding a variable to $S_k$, controlled by hyperparameters $\alpha_{\text{adding}}$ and $\gamma_{\text{adding}}$. This prior is inspired by Bayesian CART (Chipman et al., 1998), where $S_k$ denotes split variables.

**Prior for the numeric parameters given $K$ and $\mathbf{S}_K$:** All the remaining parameters are numerical ones and hence we use standard priors for them.

- Conditional on $K$, we assume a prior that $\beta_k$s are independent and follow $\beta_k \sim N(0, \sigma_\beta^2)$, where $\sigma_\beta > 0$ is a hyperparameter.

- Conditional on $S_k$, we let $b_{j,k}$s and $\gamma_{j,k}$s be all independent and $b_{j,k} \sim \text{Uniform}(0, 1)$ and $\gamma_{j,k} \sim \text{Gamma}(a_\gamma, b_\gamma)$ for $j \in S_k$ and $k \in [K]$, where $a_\gamma > 0$ and $b_\gamma > 0$ are hyperparameters.

- For the nuisance parameter in the gaussian regression model, where the nuisance parameter $\eta$ corresponds to $\sigma^2$, we set $\sigma^2 \sim \text{IG}\big(\frac{v}{2}, \frac{v\lambda}{2}\big)$, where $v > 0$ and $\lambda > 0$ are hyperparameters and $\text{IG}(\cdot, \cdot)$ is the inverse gamma distribution.

### 3.2 MCMC Algorithm for Posterior Sampling

We now develop an MCMC algorithm for posterior sampling of Bayesian-TPNN. Our overall sampling strategy is to update $K$, $\mathbf{S}_K$ and the remaining numeric parameters iteratively using the corresponding Metropolis-Hastings (MH) algorithms, which is motivated by the MCMC algorithm of Bayesian additive regression tree (Chipman et al., 2010). A novel part of our MCMC algorithm, however, is to devise a specially designed proposal distribution in the MH algorithm such that the proposal distribution encourages the MCMC algorithm to visit important higher-order interactions more frequently. For this purpose, we introduce two special tools. First, we employ a pretrained probability mass function $p_{\text{input}}(\cdot)$ on $[p]$, which represents the importance of each input variable. Further, let $p_{\text{input}}(\cdot | S)$ be the distribution $p_{\text{input}}(\cdot)$ restricted to $S \subseteq [p]$. See Remark at the end of this subsection for the choice of $p_{\text{input}}(\cdot)$.

The second tool is a stepwise search. The stepwise search adds a new node by first copying one of existing nodes and add an edge. By doing so, a newly added node has one more edges than the copied node and thus corresponds to an interaction whose order is larger than the copied one by 1. By keeping the copied node also in the model, we can avoid dramatic loss of accuracy.

To be more specific, let $\theta := (K, \mathbf{S}_K, \mathbf{b}_{\mathbf{S}_K, K}, \Gamma_{\mathbf{S}_K, K}, \mathcal{B}_K, \eta)$ be given current parameters. We update these parameters by sequentially updating $K$, $(\mathbf{S}_K, \mathbf{b}_{\mathbf{S}_K, K}, \Gamma_{\mathbf{S}_K, K}, \mathcal{B}_K)$ and the nuisance parameter $\eta$. We now describe these 3 updates.

**Updating $K$:** First, we devise a proposal distribution of $K^{\text{new}}$ given $K$ used in the MH algorithm. For a given $K$, we set $K^{\text{new}}$ as $K - 1$ or $K + 1$ with probability $K/K_{\max}$ and $1 - K/K_{\max}$ respectively. If $K^{\text{new}} = K - 1$, we remove one of $(S_k, \mathbf{b}_{S_k, k}, \Gamma_{S_k, k}, \beta_k), k \in [K]$ from $\theta$ with probability $1/K$ to have $\theta^{\text{new}}$.

For the case $K^{\text{new}} = K + 1$, the crucial mission is to design an appropriate proposal of $(S_{K+1}^{\text{new}}, \mathbf{b}_{S_{K+1}^{\text{new}}, K+1}^{\text{new}}, \Gamma_{S_{K+1}^{\text{new}}, K+1}^{\text{new}}, \beta_{K+1}^{\text{new}})$. Specifically, we first generate $S_{K+1}^{\text{new}}$ and then generate $(\mathbf{b}_{S_{K+1}^{\text{new}}, K+1}^{\text{new}}, \Gamma_{S_{K+1}^{\text{new}}, K+1}^{\text{new}}, \beta_{K+1}^{\text{new}})$ conditional on $S_{K+1}^{\text{new}}$. The proposal of $S_{K+1}^{\text{new}}$ consists of the following two alternations:

- **Random**: Generate $S_{K+1}^{\text{new}}$ from the prior distribution.
- **Stepwise**: Propose $S_{K+1}^{\text{new}} = S_{k^*} \cup \{j_{k^*}\}$, where $k^* \sim \text{Uniform}[K]$ and $j_{k^*} \sim p_{\text{input}}(\cdot | S_{k^*}^c)$.

The MH algorithm randomly selects one of {**Random**, **Stepwise**} with probability $M/(M + K)$, and $K/(M + K)$, where $M > 0$ is a hyperparameter. This proposal combines random and stepwise search, where $S_{K+1}^{\text{new}}$ is sampled as a completely new index set from the prior with probability $M/(M + K)$, or taken as a higher-order modification of one of $S_1, \ldots, S_K$ with probability $K/(M + K)$. We employ **Stepwise** move to encourage the proposal distribution to explore higher-order interactions more frequently without losing much information in the current model (i.e. keeping all of the components in the current model). Once $S_{K+1}^{\text{new}}$ is given, we generate $(\mathbf{b}_{S_{K+1}^{\text{new}}, K+1}^{\text{new}}, \Gamma_{S_{K+1}^{\text{new}}, K+1}^{\text{new}}, \beta_{K+1}^{\text{new}})$ from the prior distribution. See Section A.1 of Appendix for the acceptance probability for this proposal $\theta^{\text{new}}$ and see Section C.5 of Appendix for experimental results demonstrating the effectiveness of the proposed MH.

**Updating** $(S_k, \mathbf{b}_{S_k,k}, \Gamma_{S_k,k}, \beta_k)$ **for** $k \in [K]$**:**   For a given $k$, we consider the following three possible alterations of $S_k$ and $(\mathbf{b}_{S_k,k}, \Gamma_{S_k,k})$ for the proposal of $(S_k^{\text{new}}, \mathbf{b}_{S_k^{\text{new}},k}^{\text{new}}, \Gamma_{S_k^{\text{new}},k}^{\text{new}})$:

- **Adding**: Adding a new variable $j^{\text{new}}$, which is selected randomly from $S_k^c$ according to the probability distribution $p_{\text{input}}(\cdot|S_k^c)$, and generating $b_{j^{\text{new}},k}$ and $\gamma_{j^{\text{new}},k}$ from the prior distribution.

- **Deleting**: Uniformly at random, select an index $j$ in $S_k$ and delete it from $S_k$.

- **Changing**: Select an index $j$ uniformly at random from $S_k$ and index $j^{\text{new}}$ from $S_k^c$ according to the probability distribution of $p_{\text{input}}(\cdot|S_k^c)$ and delete $j$ from $S_k$ and add $j^{\text{new}}$ to $S_k$. Then, generate $b_{j^{\text{new}},k}$ and $\gamma_{j^{\text{new}},k}$ from the prior distribution.

The MH algorithm randomly selects one of {**Adding**, **Deleting**, **Changing**} with probability $(q_{\text{add}}, q_{\text{delete}}, q_{\text{change}})$. This proposal distribution is a modification of one used in BART (Chipman et al., 1998; Kapelner & Bleich, 2016) to grow/prune or modify a current decision tree. See Section A.2 of Appendix for the acceptance probability of $(S_k^{\text{new}}, \mathbf{b}_{S_k^{\text{new}},k}^{\text{new}}, \Gamma_{S_k^{\text{new}},k}^{\text{new}})$.

Once $(S_k, \mathbf{b}_{S_k,k}, \Gamma_{S_k,k})$ are updated, we update all of the numeric parameters $(\mathbf{b}_{S_k,k}, \Gamma_{S_k,k}, \beta_k)$ by the MH algorithm with the Langevin proposal (ros, 1978) to accelerate the convergence of the MCMC algorithm further. Finally, we repeat this update for $k \in [K]$ sequentially. See Appendix A.3 for details and Section I for a toy example illustrating the proposed MCMC algorithm.

**Updating the nuisance parameter** $\eta$ **:**   In the gaussian regression model, the nuisance parameter $\eta$ corresponds to the error variance $\sigma_g^2$. Since the conditional posterior distribution of $\sigma_g^2$ is Inverse Gamma distribution, it is straightforward to draw $\sigma_g^2$ from $\pi(\sigma_g^2|\text{others})$. Details are provided in Section A.4 of Appendix.

---

**Algorithm 1** MCMC algorithm of Bayesian TPNN.

---

**Input** $\{(\mathbf{x}_i, y_i)\}_{i=1}^n$ : data, $K$ : initial number of hidden nodes, $M_{\text{mcmc}}$ : the number of MCMC iterations,

1: **for** i : 1 to $M_{\text{mcmc}}$ **do**
2:     Update $K$
3:     **for** k : 1 to $K$ **do**
4:         Update $S_k, \mathbf{b}_{S_k,k}, \Gamma_{S_k,k}$
5:         Update $\mathbf{b}_{S_k,k}, \Gamma_{S_k,k}, \beta_k$
6:     **end for**
7:     Update $\eta$
8: **end for**

---

**Predictive Inference.**   Let $\hat{\theta}_1, ..., \hat{\theta}_N$ denote samples drawn from the posterior distribution. The predictive distribution is then estimated as $\hat{p}(y|\mathbf{x}) = \sum_{i=1}^N p(y|\mathbf{x}, \hat{\theta}_i)/N$.

**Remark 3.1.** *When no prior information is available on the importance of input variables, we use a uniform distribution for $p_{input}$. However, this noninformative choice often performs poorly when the dimension $p$ is large and higher-order interactions exist. Our numerical studies in Section C.4 reveal that the choice of a good $p_{input}$ is important for exploring higher-posterior regions. In practice, we*

*could specify $p_{input}$ based on the importance measures of each input variable obtained by a standard method such as Molnar (2020). That is, we let $p_{input}(j) \propto \omega_j$, where $\omega_j$ is an importance measure of the input variable $j \in [p]$. In our numerical study, we use the global SHAP value (Molnar, 2020) based on a pretrained Deep Neural Network (DNN) for the importance measure or the feature importance using a pretrained eXtreme Gradient Boosting (XGB, Chen & Guestrin (2016)).*

### 3.3 POSTERIOR CONSISTENCY

For theoretical justification of Bayesian-TPNN, in this section, we prove the posterior consistency of Bayesian-TPNN. To avoid unnecessary technical difficulties, we assume that $\phi(\mathbf{x}|\Theta_k)$ in (8) satisfies the sum-to-zero condition with respect to the uniform distribution. This can be done by using the uniform distribution instead of the empirical distribution in (6).

We assume that $(\mathbf{x}_1, y_1), ..., (\mathbf{x}_n, y_n)$ are realizations of independent copies $(\mathbf{X}_1, Y_1), ..., (\mathbf{X}_n, Y_n)$ of $(\mathbf{X}, Y)$ whose distribution $\mathbb{Q}_0$ is given as

$$\mathbf{X} \sim \mathbb{P}_{\mathbf{X}} \quad \text{and} \quad Y|\mathbf{X} = \mathbf{x} \sim \mathbb{Q}_{f_0(\mathbf{x}),1},$$

where $f_0$ is the true regression function. We let $\eta = 1$ for technical simplicity. Suppose that $f_0(\mathbf{x}) = \sum_{S \subseteq [p]} f_{0,S}(\mathbf{x}_S)$, where each $f_{0,S}$ satisfies the sum-to-zero condition with respect to the uniform distribution. We denote $\mathbf{X}^{(n)} = \{\mathbf{X}_1, ..., \mathbf{X}_n\}$ and $Y^{(n)} = \{Y_1, ..., Y_n\}$. Let $\pi_\xi(\cdot) \propto \pi(\cdot)\mathbb{I}(\|f\|_\infty \leq \xi)$, where $\pi(\cdot)$ is the prior distribution of $f$ defined in Section 3.1. Under regularity conditions $(S.1)$, $(S.2)$, $(S.3)$ and $(S.4)$ in Section M.2 of Appendix, Theorem 3.2 proves the posterior consistency of each component of Bayesian-TPNN.

**Theorem 3.2** (Posterior Consistency of Bayesian-TPNN). *Assume that $0 < \inf_{\mathbf{x} \in \mathcal{X}} p_{\mathbf{X}}(\mathbf{x}) \leq \sup_{\mathbf{x} \in \mathcal{X}} p_{\mathbf{X}}(\mathbf{x}) < \infty$, where $p_{\mathbf{X}}(\mathbf{x})$ is the density of $\mathbb{P}_{\mathbf{X}}$. Then, there exists $\xi > 0$ such that for any $\varepsilon > 0$, we have*

$$\pi_\xi\left(f : \|f_{0,S} - f_S\|_{2,n} > \varepsilon \Big| \mathbf{X}^{(n)}, Y^{(n)}\right) \to 0 \tag{11}$$

*for all $S \subseteq [p]$ in $\mathbb{Q}_0^n$ as $n \to \infty$, where $\pi_\xi(\cdot|\mathbf{X}^{(n)}, Y^{(n)})$ is the posterior distribution of Bayesian-TPNN with the prior $\pi_\xi$.*

## 4 EXPERIMENTS

We present the results of the numerical experiments in this section, while further results and comprehensive details regarding the datasets, implementations of baseline models, and hyperparameter selections are provided in Sections B to H of Appendix.

### 4.1 PREDICTION PERFORMANCE

We compare the prediction performance of Bayesian-TPNN with baseline models including ANOVA-TPNN (Park et al., 2025), Neural Additive Models (NAM, Agarwal et al. (2021)), Linear model, XGB (Chen & Guestrin, 2016), Bayesian Additive Regression Trees (BART, Chipman et al. (2010), Linero (2025)) and mBNN (Kong et al., 2023). We analyze eight real datasets and split each dataset into training and test sets with a ratio of 0.8 to 0.2. This random split is repeated five times to obtain five prediction performance measures.

Table 1 reports the prediction accuracies (the Root Mean Square Error (RMSE) for regression tasks and the Area Under the ROC Curve (AUROC) for classification tasks) of the Bayes estimator of Bayesian-TPNN along with those of its competitors, where the best results are highlighted by **bold**. Overall, Bayesian-TPNN achieves prediction performance comparable to that of the baseline models. Further details of the experiments are provided in Section B.3 of Appendix.

Table 2 compares Bayesian-TPNN with the baseline Bayesian models in view of uncertainty quantification. As uncertainty quantification measures, we consider Continuous Ranked Probability Score (CRPS, Gneiting & Raftery (2007)) and Negative Log-Likelihood (NLL) for regression tasks, and Expected Calibration Error (ECE, Kumar et al. (2019)) together with NLL for classification tasks. The results indicate that Bayesian-TPNN compares favorably with the baseline models in uncertainty quantification, which is a bit surprising since Bayesian-TPNN is a transparent model while

Table 1: **The averaged prediction accuracies (the standard errors) on real datasets.**

| | | Interpretable model | | | | Blackbox model | | |
|---|---|---|---|---|---|---|---|---|
| Dataset | Measure | Bayesian TPNN | ANOVA TPNN | NAM | Linear | XGB | BART | mBNN |
| ABALONE (Warwick et al., 1995) | | 2.053 (0.26) | **2.051** (0.21) | 2.062 (0.23) | 2.244 (0.22) | 2.157 (0.24) | 2.197 (0.26) | 2.081 (0.24) |
| BOSTON (Harrison Jr & Rubinfeld, 1978) | RMSE ↓ (SE) | **3.654** (0.49) | 3.671 (0.56) | 3.832 (0.67) | 5.892 (0.77) | 4.130 (0.56) | 4.073 (0.67) | 4.277 (0.51) |
| MPG (Quinlan, 1993) | | **2.386** (0.41) | 2.623 (0.38) | 2.755 (0.41) | 3.748 (0.41) | 2.531 (0.26) | 2.699 (0.43) | 2.897 (0.42) |
| SERVO (Ulrich, 1986) | | 0.351 (0.02) | 0.594 (0.04) | 0.802 (0.04) | 1.117 (0.04) | 0.314 (0.04) | 0.342 (0.04) | **0.301** (0.04) |
| FICO (fic, 2018) | | 0.793 (0.009) | **0.802** (0.008) | 0.764 (0.019) | 0.690 (0.010) | 0.793 (0.009) | 0.701 (0.015) | 0.740 (0.008) |
| BREAST (Wolberg et al., 1993) | AUROC ↑ (SE) | 0.998 (0.001) | **0.998** (0.001) | 0.976 (0.003) | 0.922 (0.010) | 0.995 (0.002) | 0.977 (0.006) | 0.978 (0.002) |
| CHURN (chu, 2017) | | **0.849** (0.008) | 0.848 (0.006) | 0.835 (0.008) | 0.720 (0.002) | 0.848 (0.006) | 0.835 (0.008) | 0.833 (0.008) |
| MADELON (Guyon, 2004) | | 0.854 (0.013) | 0.587 (0.013) | 0.644 (0.005) | 0.548 (0.011) | **0.884** (0.006) | 0.751 (0.011) | 0.650 (0.018) |

Table 2: **Comparison of Bayesian models in view of uncertainty quantification on real datasets.**

| | Bayesian-TPNN | | BART | | mBNN | |
|---|---|---|---|---|---|---|
| Dataset | CRPS | NLL | CRPS | NLL | CRPS | NLL |
| ABALONE | **1.372** (0.19) | 2.260 (0.16) | 1.384 (0.18) | 2.261 (0.16) | 1.399 (0.16) | **2.226** (0.16) |
| BOSTON | **2.202** (0.23) | 3.411 (0.37) | 2.623 (0.25) | **3.400** (0.42) | 3.144 (0.39) | 3.488 (0.26) |
| MPG | **1.510** (0.43) | **2.511** (0.21) | 1.553 (0.27) | 2.530 (0.20) | 2.142 (0.42) | 2.710 (0.24) |
| SERVO | 0.194 (0.01) | 0.836 (0.10) | 0.202 (0.02) | 0.849 (0.08) | **0.185** (0.02) | **0.321** (0.08) |
| Dataset | ECE | NLL | ECE | NLL | ECE | NLL |
| FICO | **0.036** (0.004) | **0.554** (0.007) | 0.054 (0.011) | 0.632 (0.012) | 0.219 (0.032) | 0.773 (0.046) |
| BREAST | 0.129 (0.009) | 0.211 (0.014) | **0.118** (0.010) | **0.143** (0.032) | 0.292 (0.018) | 0.523 (0.025) |
| CHURN | **0.031** (0.001) | **0.418** (0.008) | 0.035 (0.001) | 0.430 (0.010) | 0.168 (0.037) | 0.531 (0.036) |
| MADELON | 0.076 (0.004) | **0.478** (0.009) | **0.066** (0.004) | 0.685 (0.032) | 0.252 (0.020) | 0.840 (0.031) |

the other Bayesian models are black-box models. The results of uncertainty quantification for non-Bayesian models are presented in Section H.1 of Appendix, which are inferior to Bayesian models.

## 4.2 PERFORMANCE IN COMPONENT SELECTION

Table 3: **Performance of component selection on synthetic datasets.**

| True model | $f^{(1)}$ | | | $f^{(2)}$ | | | $f^{(3)}$ | | |
|---|---|---|---|---|---|---|---|---|---|
| Order | Bayesian TPNN | ANOVA T²PNN | NA²M | Bayesian TPNN | ANOVA T²PNN | NA²M | Bayesian TPNN | ANOVA T²PNN | NA²M |
| 1 | **1.000** (0.000) | 0.999 (0.001) | 0.528 (0.023) | 0.831 (0.008) | **0.859** (0.010) | 0.417 (0.015) | **1.000** (0.000) | 0.781 (0.021) | 0.522 (0.011) |
| 2 | **1.000** (0.000) | 0.978 (0.007) | 0.508 (0.024) | **0.985** (0.003) | 0.949 (0.003) | 0.838 (0.009) | 0.922 (0.019) | 0.704 (0.007) | 0.542 (0.017) |
| 3 | **0.740** (0.022) | — | — | **0.966** (0.018) | — | — | **0.661** (0.022) | — | — |

Table 4: **Top 5 components: the important scores are normalized by their maximum.**

| | Rank 1 | | Rank 2 | | Rank 3 | | Rank 4 | | Rank 5 | |
|---|---|---|---|---|---|---|---|---|---|---|
| Dataset | Component | Score | Component | Score | Component | Score | Component | Score | Component | Score |
| MADELON | (49, 242, 319, 339) | 1.000 | (129, 443, 494) | 0.472 | (379, 443) | 0.374 | 106 | 0.322 | (242, 443) | 0.301 |
| SERVO | 1 | 1.000 | (1, 3, 4, 5) | 0.554 | 4 | 0.202 | (4, 6) | 0.193 | 8 | 0.173 |

We investigate whether Bayesian-TPNN identifies the true signal components well similarly to the setting in Park et al. (2025); Tsang et al. (2017). Synthetic datasets are generated from $Y = f^{(k)}(\mathbf{x}) + \epsilon$, $k = 1, 2, 3$, where $f^{(k)}$ is the true regression model and $\mathbf{x} \in \mathbb{R}^{50}$. Details of the experiment are described in Section B.5.

We define the importance score of each component as its $\ell_2$-norm, i.e., $\|f_S\|_{2,n}$. A large $\|f_S\|_{2,n}$ implies $f_S$ is a signal. Table 3 reports the averages (standard errors) of AUROCs of the importance scores obtained by Bayesian-TPNN, ANOVA-T²PNN, and NA²M for interaction order up to 3. Note that extending ANOVA-T²PNN and NA²M to include the third order interactions requires additional

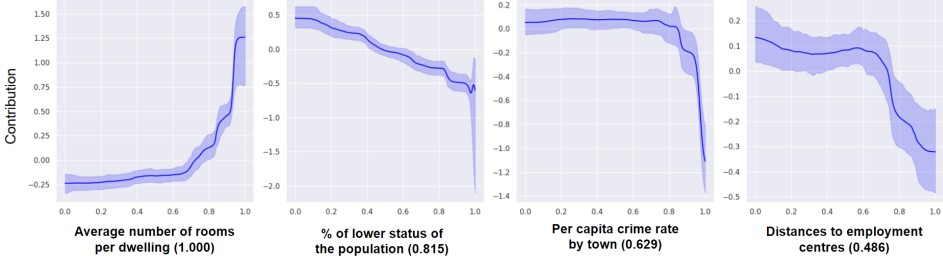

Figure 2: **Plots of the functional relations of the important main effects estimated by Bayesian-TPNN on the BOSTON dataset.** Each plot shows the Bayes estimate and 95% credible interval of each component. Labels indicate the names of the input variables along with the normalized importance scores.

19, 600 neural networks, and so we give up ANOVA-T$^3$PNN and NA$^3$M due to the limitations of our computational environment. Overall, Bayesian-TPNN achieves the best performance in component selection across orders and datasets, and detects higher-order interactions reasonably well.

Table 4 presents the five most important components selected by Bayesian-TPNN on MADELON and SERVO datasets. We use these datasets as they highlight the performance gap between models with and without higher-order interactions. Notably, Bayesian-TPNN identifies a 4th-order interaction as the most important component in the MADELON data, suggesting that its ability to capture higher-order interactions largely explains its superior prediction performance over ANOVA-TPNN on these datasets. See Section B.2 of Appendix for descriptions of the variables in MADELON and SERVO.

### 4.3 INTERPRETATION OF BAYESIAN-TPNN

The functional ANOVA model can provide various interpretations of the estimated prediction model through the estimated components as Park et al. (2025) illustrates. In particular, by visualizing the estimated components, we can understand how each group of input variables affects the response variable. Figure 2 presents the plots of the functional relations for the important main effects estimated by Bayesian-TPNN on the BOSTON dataset. Each plot shows the Bayes estimate and the 95% credible interval of the selected component. The leftmost plot shows increasing trend, indicating that as the average number of rooms per dwelling increases, the price of the housing increases as well. The second plot reveals a strictly decreasing relationship between the proportion of lower status of the population and the housing price. The third plot indicates that housing prices decrease sharply once the crime rate exceeds a certain threshold. The fourth plot shows that houses located farther from major employment centers are generally less expensive than those situated closer to such hubs. More discussions about interpretation of Bayesian-TPNN are provided in Section E of Appendix.

### 4.4 APPLICATION TO CONCEPT BOTTLENECK MODELS

Concept Bottleneck Model (CBM, Koh et al. (2020)) is an interpretable model in which a CNN first receives an image and predicts its concepts. These predicted concepts are then used to infer the target label, enabling explainable predictions. To illustrate that Bayesian-TPNN can be amply combined with CBM, we consider Independent Concept Bottleneck Models (ICBM, Koh et al. (2020)), in which a CNN is first trained and then frozen, after which a final classifier is trained on the predicted concepts. We compare Bayesian-TPNN with other baselines for learning the final classifier. In the experiment, we use CELEBA-HQ (Lee et al., 2020) and CATDOG (Jikadara, 2023) datasets, where we generate 5 concepts using GPT-5 (OpenAI, 2025), and we obtain the concept labels for each image via CLIP (Radford et al., 2021). The target labels for CELEBA-HQ and CATDOG are gender and cat/dog classification, respectively. The details are provided in Section B.4 of Appendix.

Table 5: **Prediction performance with CBM on image datasets.**

| Dataset | Measure | Bayesian-TPNN | ANOVA-T$^2$PNN | NA$^2$M | Linear |
|---------|---------|---------------|----------------|---------|--------|
| CELEBA-HQ | AUROC ↑ | **0.936** (0.002) | 0.923 (0.002) | 0.922 (0.002) | 0.893 (0.003) |
| CATDOG | AUROC ↑ | **0.878** (0.002) | 0.853 (0.002) | 0.851 (0.002) | 0.711 (0.001) |

Table 5 presents the averages and standard errors of AUROCs when Bayesian-TPNN, ANOVA-T$^2$PNN, NA$^2$M, and Linear model are used in the final classifier. Among them, Bayesian-TPNN

attains the highest prediction performance, which can be attributed to its capability to estimate higher-order components.

## 5 CONCLUSION

We proposed Bayesian-TPNN, a novel Bayesian neural network for the functional ANOVA model that can detect higher-order signal components effectively and thus achieve superior prediction performance in view of prediction accuracy and uncertainty quantification. In addition, Bayesian-TPNN is also theoretically sound since it achieves the posterior consistency.

We used a predefined distribution $p_{\text{input}}$ for the selection probability of each input variable in the MH algorithm. It would be interesting to update $p_{\text{input}}$ along with the other parameters. For example, it would be possible to let $p_{\text{input}}(j)$ be proportional to the number of basis functions in the current Bayesian-TPNN model which uses $x_j$. This would be helpful when $p$ is large. We will pursue this algorithm in the near future.

**Reproducibility Statement.** We have made significant efforts to ensure the reproducibility of our results. The source code implementing our proposed model and experiments is provided in the supplementary material. Detailed descriptions of the experimental setup, hyperparameters and datasets are provided in Section B of Appendix. Additional ablation studies are reported in Section C of Appendix.

## ACKNOWLEDGEMENTS

This work was supported by the National Research Foundation of Korea(NRF) grant funded by the Korea government(MSIT) (RS-2025-00556079), Institute of Information & communications Technology Planning & Evaluation (IITP) grant funded by the Korea government(MSIT) [NO.RS-2021-II211343, Artificial Intelligence Graduate School Program (Seoul National University)] and by the National Research Foundation of Korea(NRF) grant funded by the Korea government(MSIT)(No. 2022R1A5A7083908)

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

# APPENDIX

## A  DETAILS OF THE MCMC ALGORITHM

For given data $\mathcal{D}^{(n)}$, we denote $\mathbf{x}^{(n)} = \{\mathbf{x}_1, ..., \mathbf{x}_n\}$. Let $\omega_j = p_{\text{input}}(j)$.

### A.1  SAMPLING $K$ VIA MH ALGORITHM

#### A.1.1  CASE OF $K^{\text{new}} = K + 1$

From current state $\theta = (K, \mathbf{S}_K, \mathbf{b}_{\mathbf{S}_K,K}, \Gamma_{\mathbf{S}_K,K}, \mathcal{B}_K, \eta)$, we propose a new state $\theta^{\text{new}}$ using one of {**Random**, **Stepwise**}. Here, $\theta^{\text{new}}$ is defined as

$$\theta^{\text{new}} = (K + 1, \mathbf{S}_{K+1}, \mathbf{b}_{\mathbf{S}_{K+1},K+1}, \Gamma_{\mathbf{S}_{K+1},K+1}, \mathcal{B}_{K+1}, \eta),$$

where

$$\mathbf{S}_{K+1} = (\mathbf{S}_K, S_{K+1}^{\text{new}}),$$
$$\mathbf{b}_{\mathbf{S}_{K+1},K+1} = (\mathbf{b}_{\mathbf{S}_K,K}, \mathbf{b}_{S_{K+1}^{\text{new}},K+1}),$$
$$\Gamma_{\mathbf{S}_{K+1},K+1} = (\Gamma_{\mathbf{S}_K,K}, \Gamma_{S_{K+1}^{\text{new}},K+1}),$$
$$\mathcal{B}_{K+1} = (\mathcal{B}_K, \beta_{K+1}^{\text{new}}).$$

We accept the new state $\theta^{\text{new}}$ with probability

$$P_{\text{accept}} = \min\left\{1, \prod_{i=1}^{n} \frac{q_{f_{\theta^{\text{new}}}(\mathbf{x}_i),\eta}(y_i)}{q_{f_\theta(\mathbf{x}_i),\eta}(y_i)} \frac{\pi(\theta^{\text{new}})}{\pi(\theta)} \frac{q(\theta|\theta^{\text{new}})}{q(\theta^{\text{new}}|\theta)}\right\},$$

where

$$f_\theta(\mathbf{x}) = \sum_{k \in [K]} \beta_k \phi(\mathbf{x}|S_k, \mathbf{b}_{S_k,k}, \Gamma_{S_k,k})$$

and

$$f_{\theta^{\text{new}}}(\mathbf{x}) = f_\theta(\mathbf{x}) + \beta_{K+1}^{\text{new}} \phi(\mathbf{x}|S_{K+1}^{\text{new}}, \mathbf{b}_{S_{K+1}^{\text{new}},K+1}, \Gamma_{S_{K+1}^{\text{new}},K+1}).$$

To compute the acceptance probability, we calculate the prior ratio $\pi(\theta^{\text{new}})/\pi(\theta)$, and then the proposal ratio $q(\theta|\theta^{\text{new}})/q(\theta^{\text{new}}|\theta)$.

**Prior Ratio.**  The prior ratio is given as

$$\frac{\pi(\theta^{\text{new}})}{\pi(\theta)} = \frac{\pi(K+1)\pi(\mathbf{S}_{K+1}|K+1)\pi(\mathbf{b}_{\mathbf{S}_{K+1},K+1}|\mathbf{S}_{K+1})\pi\left(\Gamma_{\mathbf{S}_{K+1},K+1}|\mathbf{S}_{K+1}\right)\pi\left(\mathcal{B}_{K+1}|K+1\right)}{\pi(K)\pi(\mathbf{S}_K|K)\pi(\mathbf{b}_{\mathbf{S}_K,K}|\mathbf{S}_K)\pi\left(\Gamma_{\mathbf{S}_K,K}|\mathbf{S}_K\right)\pi\left(\mathcal{B}_K|K\right)}$$
$$= \frac{\pi(S_{K+1})\pi(\mathbf{b}_{S_{K+1},K+1})\pi(\Gamma_{S_{K+1},K+1})\pi(\beta_{K+1})}{\exp(C_0 \log n)}.$$

**Proposal Ratio.**  For $q(\theta|\theta^{\text{new}})$, we have

$$q(\theta|\theta^{\text{new}}) = \Pr(K = K^{\text{new}} - 1)\Pr(\text{Choose one of } K^{\text{new}} \text{ TPNNs for deletion})$$
$$= \frac{K^{\text{new}}}{K_{\text{max}}} \frac{1}{K^{\text{new}}}.$$

For a given $\theta$, a new state $\theta^{\text{new}}$ is proposed in two ways: (1) **Random** move or (2) **Stepwise** move.

For **Random** move, we have

$$q(\theta^{\text{new}}|\theta, \textbf{Random}) = \pi(S_{K^{\text{new}}})\pi(\mathbf{b}_{S_{K+1}^{\text{new}},K+1})\pi(\Gamma_{S_{K+1}^{\text{new}},K+1})\pi(\beta_{K+1}^{\text{new}}). \tag{12}$$

For **Stepwise** move, we have

$$q(\theta^{\text{new}}|\theta, \textbf{Stepwise}) = \Pr(S_{K+1}^{\text{new}})\pi(\mathbf{b}_{S_{K+1}^{\text{new}},K+1})\pi(\Gamma_{S_{K+1}^{\text{new}},K+1})\pi(\beta_{K+1}^{\text{new}}).$$

Here, $\Pr(S_{K+1}^{\text{new}})$ is defined as

$$\Pr(S_{K+1}^{\text{new}}) = \sum_{k=1}^{K} \Pr(\text{Choose } S_k \text{ from } \mathbf{S}_K)\Pr(S_{K+1}^{\text{new}} = S_k \cup \{j^{\text{new}}\}, j^{\text{new}} \in S_k^c)$$

$$= \sum_{k=1}^{K} \frac{1}{K}\mathbb{I}(\exists j^{\text{new}} \in S_k^c \text{ s.t } S_k \cup \{j^{\text{new}}\} = S_{K+1}^{\text{new}})\frac{\omega_{j^{\text{new}}}}{\sum_{l \in S_k^c} \omega_l}.$$

To sum up, we have

$$q(\theta^{\text{new}}|\theta) = q(\theta^{\text{new}}|\theta, \textbf{Random})\Pr(\textbf{Random}) + q(\theta^{\text{new}}|\theta, \textbf{Stepwise})\Pr(\textbf{Stepwise}).$$

### A.1.2 Case of $K^{\text{new}} = K - 1$

Since the acceptance probability of the case $K^{\text{new}} = K - 1$ can be easily computed by reversing the steps in Section A.1.1, we omit the details here.

### A.2 Sampling $S_k, \mathbf{b}_k, \Gamma_k$ via MH algorithm

Here, we consider three moves - {**Adding, Deleting** and **Changing**}. Each move is chosen with the probabilities $\Pr(\textbf{Adding}) = q_{\text{add}}, \Pr(\textbf{Deleting}) = q_{\text{delete}}, \Pr(\textbf{Changing}) = q_{\text{change}}$, respectively.

In **Adding** move, the proposal distribution generates $S_k^{\text{new}} = S_k \cup \{j^{\text{adding}}\}$, where $j^{\text{adding}} \in [p]\backslash S_k$ is chosen with a given weight vector $\boldsymbol{\omega} := (\omega_1, ..., \omega_p)$. Note that the likelihood cannot be calculated using $S_k^{\text{new}}$ alone, where $S_k^{\text{new}}$ is the index set generated by the proposal distribution. To address this, we also generate $b_{j^{\text{adding}},k}$ and $\gamma_{j^{\text{adding}},k}$ from Uniform$(0, 1)$ and Gamma$(a_\gamma, b_\gamma)$, respectively.

Furthermore, in **Deleting** move, a variable to be deleted is uniformly selected from $S_k$ and the new component $S_k^{\text{new}} = S_k\backslash\{j^{\text{deleting}}\}$ is proposed accordingly. This move also involves removing the associated numeric parameters $b_{j^{\text{deleting}},k}$ and $\gamma_{j^{\text{deleting}},k}$ from $\mathbf{b}_{S_k,k}$ and $\Gamma_{S_k,k}$, respectively.

Finally, in **Changing** move, we choose an element $j^{\text{change}}$ in $S_k$ and replace it with a randomly selected $j^{\text{new}} \in S_k^c$. The corresponding $b_{j^{\text{change}},k}$ and $\gamma_{j^{\text{change}},k}$ are then replaced by new values generated from Uniform$(0, 1)$ and Gamma$(a_\gamma, b_\gamma)$, respectively. This move results in $S_k^{\text{new}} = (S_k\backslash\{j^{\text{change}}\}) \cup \{j^{\text{new}}\}$.

Here, **Adding** and **Deleting** affect the dimensions of $\mathbf{b}_{S_k,k}$ and $\Gamma_{S_k,k}$, thus the algorithm corresponds to RJMCMC (Green (1995)) which requires Jacobian computations. However, since we applied the identity transformation on the auxiliary variables which are generated to match the dimensions, the Jacobian is simply 1. This allows us to easily compute the acceptance probability.

### A.2.1 Transition probability for proposal distribution

For a given weight vector $\boldsymbol{\omega}$, the proposal distributions $q_{\boldsymbol{\omega}}$ of $\Theta_k^{\text{new}} = (S_k^{\text{new}}, \mathbf{b}_{S_k^{\text{new}},k}, \Gamma_{S_k^{\text{new}},k})$ are defined as:

$$q_{\boldsymbol{\omega}}(\Theta_k^{\text{new}}|\Theta_k, \textbf{Adding}) = \frac{\omega_{j^{\text{adding}}}}{\sum_{j \in S_k^c} \omega_j}\pi(b_{j^{\text{adding}},k})\pi(\gamma_{j^{\text{adding}},k})$$

$$q_{\boldsymbol{\omega}}(\Theta_k^{\text{new}}|, \Theta_k, \textbf{Deleting}) = \frac{1}{|S_k|}$$

$$q_{\boldsymbol{\omega}}(\Theta_k^{\text{new}}|\Theta_k, \textbf{Changing}) = \frac{1}{|S_k|}\frac{\omega_{j^{\text{new}}}}{\sum_{j \in S_k^c} \omega_j}\pi(b_{j^{\text{new}},k})\pi(\gamma_{j^{\text{new}},k}).$$

To sum up, we have

$$q_{\boldsymbol{\omega}}(\Theta_k^{\text{new}}|\Theta_k) = q_{\boldsymbol{\omega}}(\Theta_k^{\text{new}}|\Theta_k, \textbf{Adding})\Pr(\textbf{Adding})$$
$$+ q_{\boldsymbol{\omega}}(\Theta_k^{\text{new}}|\Theta_k, \textbf{Deleting})\Pr(\textbf{Deleting})$$
$$+ q_{\boldsymbol{\omega}}(\Theta_k^{\text{new}}|\Theta_k, \textbf{Changing})\Pr(\textbf{Changing}).$$

### A.2.2 POSTERIOR RATIO

We define $\boldsymbol{\lambda}_k := (\lambda_{k,1}, \ldots, \lambda_{k,n})$ where $\lambda_{k,i} = \sum_{j \neq k} \beta_j \phi(\mathbf{x}_i|\Theta_j)$ for $i = 1, \ldots, n$ and the likelihood $\mathcal{L}(\Theta_k, \beta_k, \boldsymbol{\lambda}_k, \eta) := \prod_{i=1}^n q_{\lambda_{k,i}+\beta_k\phi(\mathbf{x}_i|\Theta_k),\eta}(y_i)$.

Then, we have

$$\pi(\Theta_k|\beta_k, \boldsymbol{\lambda}_k, \mathcal{D}^{(n)}, \eta) \propto \pi(y_1, \ldots, y_n|\Theta_k, \beta_k, \boldsymbol{\lambda}_k, \mathbf{x}^{(n)}, \eta)\pi(\Theta_k)$$
$$= \mathcal{L}(\Theta_k, \beta_k, \boldsymbol{\lambda}_k, \eta)\pi(\Theta_k).$$

Thus the posterior ratio of $\Theta_k^{\text{new}} = (S_k^{\text{new}}, \mathbf{b}_{S_k^{\text{new}},k}, \Gamma_{S_k^{\text{new}},k})$ to $\Theta_k = (S_k, \mathbf{b}_{S_k,k}, \Gamma_{S_k,k})$ is given as

$$\frac{\pi(\Theta_k^{\text{new}}|\beta_k, \boldsymbol{\lambda}_k, \mathcal{D}^{(n)}, \eta)}{\pi(\Theta_k|\beta_k, \boldsymbol{\lambda}_k, \mathcal{D}^{(n)}, \eta)} = \frac{\mathcal{L}(\Theta_k^{\text{new}}, \beta_k, \boldsymbol{\lambda}_k, \eta)}{\mathcal{L}(\Theta_k, \beta_k, \boldsymbol{\lambda}_k, \eta)} \frac{\pi(\Theta_k^{\text{new}})}{\pi(\Theta_k)}.$$

### A.2.3 ACCEPTANCE PROBABILITY

In this section, for notational simplicity, we denote the hyperparameters $\alpha_{\text{adding}}$ and $\gamma_{\text{adding}}$ as $\alpha$ and $\gamma$, respectively.

For a proposed new state $\Theta_k^{\text{new}}$, we accept it with probability

$$P_{\text{accept}} = \min\left\{1, \frac{\pi(\Theta_k^{\text{new}}|\beta_k, \boldsymbol{\lambda}_k, \mathcal{D}^{(n)}, \eta)}{\pi(\Theta_k|\beta_k, \boldsymbol{\lambda}_k, \mathcal{D}^{(n)}, \eta)} \frac{q_{\boldsymbol{\omega}}(\Theta_k|\Theta_k^{\text{new}})}{q_{\boldsymbol{\omega}}(\Theta_k|\Theta_k^{\text{new}})}\right\}$$
$$= \min\left\{1, \frac{\mathcal{L}(\Theta_k^{\text{new}}, \beta_k, \boldsymbol{\lambda}_k, \eta)}{\mathcal{L}(\Theta_k, \beta_k, \boldsymbol{\lambda}_k, \eta)} \frac{\pi(\Theta_k^{\text{new}})}{\pi(\Theta_k)} \frac{q_{\boldsymbol{\omega}}(\Theta_k|\Theta_k^{\text{new}})}{q_{\boldsymbol{\omega}}(\Theta_k^{\text{new}}|\Theta_k)}\right\}.$$

Now, we will show how the product of the prior and proposal ratios is calculated in the case of **Adding**, **Deleting**, and **Changing**.

For **Adding**, we have

$$\frac{\pi(\Theta_k^{\text{new}})}{\pi(\Theta_k)} \frac{q_{\boldsymbol{\omega}}(\Theta_k|\Theta_k^{\text{new}})}{q_{\boldsymbol{\omega}}(\Theta_k^{\text{new}}|\Theta_k)}$$
$$= \alpha|S_k^{\text{new}}|^{-\gamma} \frac{1 - \alpha(1 + |S_k^{\text{new}}|)^{-\gamma}}{1 - \alpha|S_k^{\text{new}}|^{-\gamma}} \frac{1}{p - |S_k^{\text{new}}| + 1} \frac{\Pr(\textbf{Deleting})}{\Pr(\textbf{Adding})} \frac{\sum_{l \in S_k^c} \omega_l}{\omega_{j^{\text{adding}}}}.$$

For **Deleting**, we have

$$\frac{\pi(\Theta_k^{\text{new}})}{\pi(\Theta_k)} \frac{q_{\boldsymbol{\omega}}(\Theta_k|\Theta_k^{\text{new}})}{q_{\boldsymbol{\omega}}(\Theta_k^{\text{new}}|\Theta_k)}$$
$$= \frac{1}{\alpha(1 + |S_k^{\text{new}}|)^{-\gamma}} \frac{1 - \alpha(1 + |S_k^{\text{new}}|)^{-\gamma}}{1 - \alpha(2 + |S_k^{\text{new}}|)^{-\gamma}}(p - |S_k^{\text{new}}|) \frac{\Pr(\textbf{Adding})}{\Pr(\textbf{Deleting})} \frac{\omega_{j^{\text{deleting}}}}{\sum_{l \in S_k^c} \omega_l}.$$

For **Changing**, we have

$$\frac{\pi(\Theta_k^{\text{new}})}{\pi(\Theta_k)} \frac{q_{\boldsymbol{\omega}}(\Theta_k|\Theta_k^{\text{new}})}{q_{\boldsymbol{\omega}}(\Theta_k^{\text{new}}|\Theta_k)} = \frac{\omega_{j^{\text{change}}} \sum_{l \in S_k^c} \omega_l}{\omega_{j^{\text{new}}} \sum_{l \in (S_k^{\text{new}})^c} \omega_l}.$$

## A.3 SAMPLING $\mathbf{b}_{S_k,k}$, $\Gamma_{S_k,k}$ AND $\beta_k$ VIA MH ALGORITHM

We use Langevin Dynamics (ros (1978)) as a proposal distribution for $\mathbf{b}_{S_k,k}$, $\Gamma_{S_k,k}$ and $\beta_k$. That is, $\mathbf{b}_{S_k,k}^{\text{new}}$, $\Gamma_{S_k,k}^{\text{new}}$ and $\beta_k^{\text{new}}$ are proposed as

$$(\mathbf{b}_{S_k,k}^{\text{new}}, \Gamma_{S_k,k}^{\text{new}}, \beta_k^{\text{new}}) = (\mathbf{b}_{S_k,k}, \Gamma_{S_k,k}, \beta_k) + \frac{\epsilon^2}{2} U(\mathbf{b}_{S_k,k}, \Gamma_{S_k,k}, \beta_k) + \epsilon \mathbb{M},$$

where

$$U(\mathbf{b}_{S_k,k}, \Gamma_{S_k,k}, \beta_k) = \nabla_{(\mathbf{b}_{S_k,k}, \Gamma_{S_k,k}, \beta_k)} \log \pi(\mathbf{b}_{S_k,k}, \Gamma_{S_k,k}, \beta_k | \boldsymbol{\lambda}_k, S_k, \mathcal{D}^{(n)}, \eta).$$

Here, $\mathbb{M} \sim N(0, \mathbf{I})$, where $\mathbf{I}$ is the $(2|S_k|+1) \times (2|S_k|+1)$ identity matrix and $\epsilon > 0$ is a step size.

We accept the proposal $(\mathbf{b}_{S_k,k}^{\text{new}}, \Gamma_{S_k,k}^{\text{new}}, \beta_k^{\text{new}})$ with a probability $P_{\text{accept}}$ given as

$$P_{\text{accept}} = \left\{ 1, \frac{\mathcal{L}(S_k, \mathbf{b}_{S_k,k}^{\text{new}}, \Gamma_{S_k,k}^{\text{new}}, \beta_k^{\text{new}}, \boldsymbol{\lambda}_k, \eta)}{\mathcal{L}(S_k, \mathbf{b}_{S_k,k}, \Gamma_{S_k,k}, \beta_k, \boldsymbol{\lambda}_k, \eta)} \frac{\pi(\mathbf{b}_{S_k,k}^{\text{new}})}{\pi(\mathbf{b}_{S_k,k})} \frac{\pi(\Gamma_{S_k,k}^{\text{new}})}{\pi(\Gamma_{S_k,k})} \frac{\pi(\beta_k^{\text{new}})}{\pi(\beta_k)} \exp \left( -\frac{1}{2} (\|\mathbb{M}^{\text{new}}\|_2^2 - \|\mathbb{M}\|_2^2) \right) \right\},$$

where $\| \cdot \|_2$ is the Euclidean norm for a vector and

$$\mathbb{M}^{\text{new}} = \mathbb{M} + \frac{\epsilon}{2} U(\mathbf{b}_{S_k,k}, \Gamma_{S_k,k}, \beta_k) + \frac{\epsilon}{2} U(\mathbf{b}_{S_k,k}^{\text{new}}, \Gamma_{S_k,k}^{\text{new}}, \beta_k^{\text{new}}).$$

For $\nabla_{(\mathbf{b}_{S_k,k}, \Gamma_{S_k,k}, \beta_k)} \log \pi(\mathbf{b}_{S_k,k}, \Gamma_{S_k,k}, \beta_k | \boldsymbol{\lambda}_k, S_k, \mathcal{D}^{(n)}, \eta)$, we will calculate

$$\nabla_{\mathbf{b}_{S_k,k}} \log \pi(\mathbf{b}_{S_k,k}, | \boldsymbol{\lambda}_k, \beta_k, S_k, \Gamma_{S_k,k}, \mathcal{D}^{(n)}, \eta),$$

$$\nabla_{\Gamma_{S_k,k}} \log \pi(\Gamma_{S_k,k} | \boldsymbol{\lambda}_k, \beta_k, S_k, \mathbf{b}_{S_k,k}, \mathcal{D}^{(n)}, \eta),$$

and

$$\nabla_{\beta_k} \log \pi(\beta_k | \boldsymbol{\lambda}_k, S_k, \mathbf{b}_{S_k,k}, \Gamma_{S_k,k}, \mathcal{D}^{(n)}, \eta).$$

### A.3.1 CALCULATING THE GRADIENT OF THE LOG-POSTERIOR WITH RESPECT TO $\mathbf{b}_{S_k,k}$

Without loss of generality, let $S_k = \{1, ..., d\}$.

Since

$$\pi(\mathbf{b}_{S_k,k} | \boldsymbol{\lambda}_k, \beta_k, S_k, \Gamma_{S_k,k}, \mathcal{D}^{(n)}, \eta) \propto \mathcal{L}(\boldsymbol{\lambda}_k, \beta_k, S_k, \mathbf{b}_{S_k,k}, \Gamma_{S_k,k}, \eta),$$

the $j$-th gradient is given as

$$\frac{\partial}{\partial b_{j,k}} \log \pi(\mathbf{b}_{S_k,k} | \boldsymbol{\lambda}_k, \beta_k, S_k, \Gamma_{S_k,k}, \mathcal{D}^{(n)}, \eta) = \frac{\partial}{\partial b_{j,k}} \sum_{i=1}^{n} \log q_{f(\mathbf{x}_i), \eta}(y_i),$$

where $f(\mathbf{x}_i) = \lambda_{k,i} + \beta_k \prod_{l \in S_k} \phi(x_{i,l} | \{l\}, b_{l,k}, \gamma_{l,k})$.

In turn, we have

$$\frac{\partial}{\partial b_{j,k}} \sum_{i=1}^{n} \log q_{f(\mathbf{x}_i), \eta}(y_i)$$

$$= \sum_{i=1}^{n} \left( \frac{\partial \log q_{f(\mathbf{x}_i), \eta}(y_i)}{\partial f(\mathbf{x}_i)} \frac{\partial f(\mathbf{x}_i)}{\partial b_{j,k}} \right)$$

$$= \beta_k \sum_{i=1}^{n} \left( \frac{\partial \log q_{f(\mathbf{x}_i), \eta}(y_i)}{\partial f(\mathbf{x}_i)} \frac{\partial \phi(x_{i,j} | \{j\}, b_{j,k}, \gamma_{j,k})}{\partial b_{j,k}} \prod_{l \neq j, l \in S_k} \phi(x_{i,l} | \{l\}, b_{l,k}, \gamma_{l,k}) \right).$$

Here,

$$\phi(x_{i,j}|\{j\}, b_{j,k}, \gamma_{j,k}) = 1 - \sigma\left(\frac{x_{i,j} - b_{j,k}}{\gamma_{j,k}}\right) + c_j(b_{j,k}, \gamma_{j,k})\sigma\left(\frac{x_{i,j} - b_{j,k}}{\gamma_{j,k}}\right),$$

$$c_j(b_{j,k}, \gamma_{j,k}) = -\left(1 - \tilde{c}_j(b_{j,k}, \gamma_{j,k})\right)\Big/\tilde{c}_j(b_{j,k}, \gamma_{j,k}),$$

where $\tilde{c}_j(b, \gamma) := \int_{\mathcal{X}_j} \sigma\left(\frac{u-b}{\gamma}\right)\mu_{n,j}(du)$.

Then, we have

$$\begin{aligned}
\frac{\partial \phi(x_{i,j}|\{j\}, b_{j,k}, \gamma_{j,k})}{\partial b_{j,k}} =& -\frac{1}{\gamma_{j,k}}\sigma\left(\frac{x_{i,j} - b_{j,k}}{\gamma_{j,k}}\right)\int_{\mathcal{X}_j}\tilde{\sigma}\left(\frac{u - b_{j,k}}{\gamma_{j,k}}\right)\mu_{n,j}(du) \\
&+ \frac{1}{\gamma_{j,k}\tilde{c}_j(b_{j,k}, \gamma_{j,k})}\tilde{\sigma}\left(\frac{x_{i,j} - b_{j,k}}{\gamma_{j,k}}\right),
\end{aligned}$$

where $\tilde{\sigma}(x) := \sigma(x)(1 - \sigma(x))$.

### A.3.2 CALCULATING THE GRADIENT OF THE LOG-POSTERIOR WITH RESPECT TO $\Gamma_{S_k,k}$

Without loss of generality, we let $S_k = \{1, ..., d\}$. Similarly to Section A.3.1 of Appendix, we can derive the gradient of the log posterior with respect to $\gamma_{j,k}$ as

$$\begin{aligned}
&\frac{\partial}{\partial \gamma_{j,k}} \log \pi(\Gamma_{S_k,k}|\boldsymbol{\lambda}_k, \beta_k, S_k, \mathbf{b}_{S_k,k}, \mathcal{D}^{(n)}, \eta) \\
&= \left(\frac{\partial}{\partial \gamma_{j,k}}\sum_{i=1}^{n}\log q_{f(\mathbf{x}_i),\eta}(y_i)\right) + (a_\gamma - 1)\frac{1}{\gamma_{j,k}} - \frac{1}{b_\gamma}
\end{aligned}$$

From $f(\mathbf{x}_i) = \lambda_{k,i} + \beta_k \prod_{l \in S_k} \phi(x_{i,l}|\{l\}, b_{l,k}, \gamma_{l,k})$, we have

$$\begin{aligned}
&\frac{\partial}{\partial \gamma_{j,k}}\sum_{i=1}^{n}\log q_{f(\mathbf{x}_i),\eta}(y_i) \\
&= \sum_{i=1}^{n}\left(\frac{\partial \log q_{f(\mathbf{x}_i),\eta}(y_i)}{\partial f(\mathbf{x}_i)}\frac{\partial f(\mathbf{x}_i)}{\partial \gamma_{j,k}}\right) \\
&= \beta_k\sum_{i=1}^{n}\left(\frac{\partial \log q_{f(\mathbf{x}_i),\eta}(y_i)}{\partial f(\mathbf{x}_i)}\frac{\partial \phi(x_{i,j}|\{j\}, b_{j,k}, \gamma_{j,k})}{\partial \gamma_{j,k}}\prod_{l\neq j, l\in S_k}\phi(x_{i,l}|\{l\}, b_{l,k}, \gamma_{l,k})\right).
\end{aligned}$$

Here,

$$\begin{aligned}
&\frac{\partial \phi(x_{i,j}|\{j\}, b_{j,k}, \gamma_{j,k})}{\partial \gamma_{j,k}} \\
&= -\frac{\int_{\mathcal{X}_j}\frac{u-b_{j,k}}{\gamma_{j,k}^2}\tilde{\sigma}\left(\frac{u-b_{j,k}}{\gamma_{j,k}}\right)\mu_{n,j}(du)}{\tilde{c}_j(b_{j,k}, \gamma_{j,k})^2}\sigma\left(\frac{x_{i,j} - b_{j,k}}{\gamma_{j,k}}\right) - (c_j(b_{j,k}, \gamma_{j,k}) - 1)\frac{x_{i,j} - b_{j,k}}{\gamma_{j,k}^2}\tilde{\sigma}\left(\frac{x_{i,j} - b_{j,k}}{\gamma_{j,k}}\right).
\end{aligned}$$

### A.3.3 CALCULATING THE GRADIENT OF THE LOG-POSTERIOR WITH RESPECT TO $\beta_k$

The gradient of the log posterior for $\beta_k$ is given as

$$\nabla_{\beta_k}\log\pi(\beta_k|\boldsymbol{\lambda}_k, S_k, \mathbf{b}_{S_k,k}, \Gamma_{S_k,k}, \mathcal{D}^{(n)}, \eta) = \sum_{i=1}^{n}\frac{\partial \log q_{f(\mathbf{x}_i),\eta}(y_i)}{\partial f(\mathbf{x}_i)}\phi(\mathbf{x}_i|\Theta_k) - \frac{\beta_k}{\sigma_\beta^2}.$$

### A.4 Sampling Nuisance parameter $\eta$

We only consider the nuisance parameter in the gaussian regression model:

$$Y_i|\mathbf{x}_i \sim N(\cdot|f(\mathbf{x}_i), \sigma_g^2)$$

for $i = 1, ..., n$, where $\sigma^2$ is a nuisance parameter. When the prior distribution is an inverse gamma distribution

$$\sigma_g^2 \sim \text{IG}\left(\frac{v}{2}, \frac{v\lambda}{2}\right), \tag{13}$$

we have

$$\sigma_g^2|K, \mathcal{B}_K, \mathbf{S}_K, \mathbf{b}_{S_K,K}, \Gamma_{S_K,K}, \mathcal{D}^{(n)} \sim \text{IG}\left(\frac{v}{2}, \frac{\frac{1}{n}\sum_{i=1}^{n}(y_i - f(\mathbf{x}_i))^2 + v\lambda}{2}\right), \tag{14}$$

and thus $\sigma_g^2$ can be sampled from the conditional posterior easily.

# B DETAILS OF THE EXPERIMENTS

## B.1 DATA DESCRIPTION

Table 6: **Descriptions of real datasets.**

| Dataset | $n$ | $p$ | Task |
|---------|-----|-----|------|
| ABALONE | 4,178 | 8 | Regression |
| BOSTON | 506 | 13 | Regression |
| MPG | 398 | 7 | Regression |
| SERVO | 167 | 4 | Regression |
| FICO | 10,459 | 23 | Classification |
| BREAST | 569 | 30 | Classification |
| CHURN | 7,043 | 20 | Classification |
| MADELON | 4,400 | 500 | Classification |
| CELEBA-HQ | 30,000 | — | Classification |
| CATDOG | 24,998 | — | Classification |

## B.2 FEATURE DESCRIPTIONS FOR MADELON AND SERVO DATASETS

Table 7: **Feature index and its corresponding description for SERVO dataset.**

| Feature index | Feature description |
|---------------|---------------------|
| 1 | Proportional gain setting for the servo motor. |
| 2 | Velocity gain setting for the servo motor. |
| 3 | Presence of Motor type A |
| 4 | Presence of Motor type B |
| 5 | Presence of Motor type C |
| 6 | Presence of Motor type D |
| 7 | Presence of Motor type E |
| 8 | Presence of Screw type A |
| 9 | Presence of Screw type B |
| 10 | Presence of Screw type C |
| 11 | Presence of Screw type D |
| 12 | Presence of Screw type E |

Table 7 presents the feature descriptions for SERVO dataset (Ulrich, 1986). MADELON (Guyon, 2004), introduced in the NIPS 2003 feature selection challenge, is a synthetic binary classification dataset with 500 features, only a few of which are informative while many are redundant or irrelevant.

## B.3 EXPERIMENT DETAILS FOR TABULAR DATASET

**Data Preprocessing.** All of the categorical input variables are encoded using the one-hot encoding. For continuous ones, the inverse of the empirical marginal CDF is used to transform them to their marginal ranks for Bayesian-TPNN and ANOVA-TPNN, whereas they are transformed via the mean-variance standardization for other baseline models.

**Implementation of baseline models.** For implementation of baseline models, we proceed as follows.

- ANOVA-TPNN : we use the official source code provided in `https://github.com/ParkSeokhun/ANOVA-TPNN`.
- NA[1]M : we use the official source code provided in `https://github.com/AmrMKayid/nam` and NA[2]M is implemented by extending the code of NA[1]M.
- Linear : We use 'scikit-learn' python package (Pedregosa et al., 2011).
- XGB : We use 'xgboost' python package (Chen & Guestrin, 2016).
- BART : We use 'BayesTree' R package (Chipman et al., 2010).
- mBNN : We use official code at `https://github.com/ggong369/mBNN`.

**Hyperparameters.** For each model, we perform 5-fold cross validation over the following hyper-parameter candidates to select the best configuration.

- Bayesian-TPNN
  - We set the step size in Langevin proposal as 0.01 and $q_{\text{add}} = 0.28$, $q_{\text{delete}} = 0.28$ and $q_{\text{change}} = 0.44$ as in Kapelner & Bleich (2016).
  - We fix $\alpha_{\text{adding}} = 0.95$ and $\gamma_{\text{adding}} = 2$, as in Chipman et al. (2010).
  - $C_0 \in \{0.001, 0.005, 0.01\}$
  - $a_\gamma \in \{1, 2, 4\}$
  - $b_\gamma \in \{10^{-3}, 5 \cdot 10^{-3}, 10^{-2}\}$
  - $K_{\max} \in \{100, 200, 300\}$
  - $\sigma_\beta^2 \in \{10^{-4}, 10^{-3}, 10^{-2}\}$
  - $M \in \{1, 5\}$
  - As in Chipman et al. (2010), for $\lambda$, we reparameterize it as $q_\lambda$, where $q_\lambda = \pi(\sigma^2 \leq \hat{\sigma}_{\text{OLS}}^2)$ and $\hat{\sigma}_{\text{OLS}}^2$ denotes the residual variance from estimated Linear model. The candidate values for $q_\lambda$ are $\{0.90, 0.95, 0.99\}$.
  - We set MCMC iterations as 1000 after 1000 burn-in iterations.

- ANOVA-TPNN
  - We set the hyperparameter candidates to be the same as those used in Park et al. (2025).
  - $K_S \in \{10, 30, 50, 100\}$
  - Adam optimizer with learning rate 5e-3.
  - Batch size = 4,096
  - Maximum order of component $\in \{1, 2\}$
  - Epoch $\in \{500, 1000, 2000\}$

- NAM
  - We set the architecture of the deep neural networks to three hidden layers with 64, 64, and 32 units, following Agarwal et al. (2021).
  - Adam optimizer with learning rate 5e-3 and weight decay 7.483e-9.
  - Batch size = 4,096
  - Maximum order of component $\in \{1, 2\}$
  - Epoch $\in \{500, 1000, 2000\}$

- BART
  - We set the hyperparmeter candidates similar to those in Chipman et al. (2010).
  - Number of trees $T \in \{50, 100, 200\}$
  - $\alpha = 0.95$ and $\beta = 2$
  - $v \in \{1, 3, 5\}$
  - $q_\lambda \in \{0.90, 0.95, 0.99\}$
  - For $\sigma_\mu = 3/(k\sqrt{T})$, $k \in \{1, 2, 3, 5\}$.
  - We set MCMC iterations as 1000 after 1000 burn-in iterations.

- XGB
  - We consider the hyperparameter candidates used in Park et al. (2025).

- mBNN
  - We consider the hyperparameter candidates similarly to Kong et al. (2023).
  - Architecture $\in \{$ 2 hidden layers with 500 and 500 units, 2 hidden layers with 1000 and 1000 units $\}$
  - Sparsity hyperparameter $\lambda \in \{0.01, 0.1, 0.5\}$
  - We set MCMC iterations as 1000 after 1000 burn-in iterations.

**Computational environments.** In this paper, all experiments are conducted on a machine equipped with an NVIDIA RTX 4000 GPU (24GB memory), an Intel(R) Xeon(R) Silver 4310 CPU @ 2.10GHz, and 128GB RAM.

### B.4 EXPERIMENT DETAILS FOR IMAGE DATASET

**CNN model.** For CNN that predicts concepts, we attach a linear head for each concept on top of the pretrained ResNet18, and train both the ResNet-18 and the linear heads jointly.

**Concept generating.** Following Oikarinen et al. (2023), we use GPT-5 (OpenAI, 2025) to generate concept dictionaries for CELEBA-HQ and CATDOG dataset. Specifically, we prompted GPT-5 as follows:

- CelebAMask-HQ is a large-scale face image dataset containing 30,000 high-resolution face images selected from CelebA, following CelebA-HQ. In this context, we aim to classify gender using the CelebAMask-HQ dataset. Could you list five high-level binary concepts that you consider most important for gender classification?
- When classifying images of cats and dogs, what are the five most important concepts to consider?

Through GPT-5, we obtained a concept dictionary

$$\{\text{'Facial hair', 'Makeup', 'Long hair', 'Angular contour', 'Accessories'}\}$$

for dataset CELEBA-HQ and another dictionary

$$\{\text{'Pointed ear', 'Short snout', 'Almond eye', 'Slender/flexible body', 'Fine/uniform fur'}\}$$

for dataset CATDOG. Each concept $c$ is divided into a positive part $c_+$ and a negative part $c_-$. For example, concept 'Makeup' can be divided into 'Makeup' and 'No Makeup', and 'Slender/flexible body' can be divided into 'Slender/flexible body' and 'Bulky/varied body'. In turn, we use the pre-trained CLIP encoder to convert $c_+$ and $c_-$ as well as each image into embedding vectors. For each concept, each image is then assigned a binary label by measuring which of the embeddings of $c_+$ and $c_-$ the image embedding is closer to.

**Hyperparameters.** For ANOVA-T$^2$PNN and NA$^2$M are trained using the Adam optimizer with a learning rate of 1e-3 and batch size of 512. For ANOVA-T$^2$PNN, the numbers of basis $K_S$ are all equal to $K$ and $K$ is determined using grid search on $\{10, 50, 100\}$. For the neural network in NA$^2$M, we set hidden layer with sizes (64,64,32). We implement Linear model as the linear logistic regression using the 'scikit-learn' package (Pedregosa et al., 2011).

### B.5 EXPERIMENT DETAILS FOR COMPONENT SELECTION

Table 8: **Definitions of $f^{(1)}$, $f^{(2)}$ and $f^{(3)}$.**

| Function | Equation |
|---|---|
| $f^{(1)}(\mathbf{x})$ | $\pi^{x_1 x_2}\sqrt{2|x_3|} - \sin^{-1}(0.5x_4) + \log(|x_3 + x_5| + 1) + \dfrac{x_9}{1 + |x_{10}|}\sqrt{\dfrac{x_7}{1 + |x_8|}} - x_2 x_7$ |
| $f^{(2)}(\mathbf{x})$ | $x_1 x_2 + 2^{x_3 + x_5 + x_6} + 2^{x_3 + x_4 + x_5 + x_7} + \sin(x_7 \sin(x_8 + x_9)) + \arccos(0.9x_{10})$ |
| $f^{(3)}(\mathbf{x})$ | $\tanh(x_1 x_2 + x_3 x_4)\sqrt{|x_5|} + \exp(x_5 + x_6) + \log((x_6 x_7 x_8)^2 + 1) + x_9 x_{10} + \dfrac{1}{1 + |x_{10}|}$ |

Table 9: **Distributions of input features for each synthetic function.**

| Function | Distribution |
|---|---|
| $f^{(1)}(\mathbf{x})$ | $X_1, X_2, X_3, X_6, X_7, X_9 \sim^{iid} U(0,1), X_4, X_5, X_8, X_{10} \sim^{iid} U(0.6,1)$ and $X_{11}, ..., X_{50} \sim^{iid} U(-1,1)$ |
| $f^{(2)}(\mathbf{x})$ | $X_1, ...., X_{50} \sim^{iid} U(-1,1)$ |
| $f^{(3)}(\mathbf{x})$ | $X_1, ...., X_{50} \sim^{iid} U(-1,1)$ |

We generate synthetic datasets from the regression model defined as

$$Y = f^{(k)}(\mathbf{x}) + \epsilon,$$

where $\epsilon \sim N(0, \sigma_\epsilon^2)$ and $\mathbf{x} \in \mathbb{R}^{50}$. Here, $f^{(k)}, k = 1, 2, 3$ are true prediction model used in Tsang et al. (2017) and defined in Table 8 and the input variables are generated from the distributions in Table 9. Input variables indexed 1–10 are informative, as they affect the output, whereas input variables 11–50 are non-informative. We choose $\sigma_\epsilon^2$ such that the signal-to-noise ratio is 5.

To evaluate the ability to detect signal components, we conduct experiments in the same manner as in Park et al. (2025). That is, we use AUROC based on the pairs of $\|\hat{f}_S^{(k)}\|_{2,n}$ and $r_S^{(k)}$, computed for all subsets $S \subseteq [p]$ with $|S| = 1, 2, 3$, where $\hat{f}_S^{(k)}$ denotes the estimate of $f_S^{(k)}$ in $f^{(k)}$ and $r_S^{(k)} = \mathbb{I}(\|f_S^{(k)}\|_{2,n} > 0)$ for $k \in \{1, 2, 3\}$.

## C    ABLATION STUDIES

### C.1    THE NUMBER OF BASIS $K$ FOR VARIOUS VALUES $C_0$

To evaluate the effect of $C_0$ in (9) on the number of bases $K$, we conduct experiments with the maximum number of bases $K_{\max}$ set to 200, and 1000 iterations for both burn-in and MCMC updates. Also, $a_\gamma$ and $b_\gamma$ are set to be 0.5 and we use ABALONE dataset. Figure 3 shows that $K$ decreases and RMSE increases as $C_0$ increases. This result demonstrates that the hyperparameter $C_0$ effectively controls model complexity by regulating the number of bases $K$. A small value of $C_0$ is recommended since an excessively large $C_0$ can be detrimental to predictive performance.

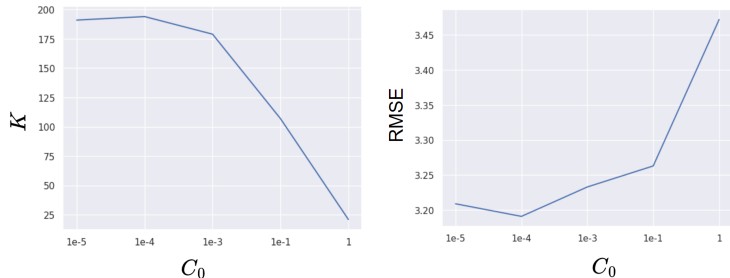

Figure 3: **Plots of the number of basis $K$ and RMSEs on various $C_0$ values.**

### C.2    IMPACT OF THE HYPERPARAMETERS $a_\gamma$ AND $b_\gamma$ ON PREDICTION PERFORMANCE

We conduct an experiment to evaluate the effect of shape parameter $a_\gamma$ and scale parameter $b_\gamma$ on prediction performance. Except for $a_\gamma$ and $b_\gamma$, the other hyperparameters of Bayesian-TPNN are set identical to those in Section C.1 of Appendix, and we analyze ABALONE dataset. We observe that prediction performance is relatively insensitive to the choice of the shape parameter $a_\gamma$, whereas it is somehow sensitive to the choice of the scale parameter $b_\gamma$. Note that the scale of $\gamma$ controls the smoothness of each TPNN basis $\phi(\mathbf{x}|\Theta)$ and thus the smoothness of Bayesian-TPNN model.

Table 10: **Prediction performance depends on various values of $a_\gamma$ and $b_\gamma$.**

| $b_\gamma \backslash a_\gamma$ | 0.5 | 1 | 2 | 3 |
|---|---|---|---|---|
| 1e-5 | 3.247 | 3.202 | 3.278 | 3.228 |
| 1e-4 | 3.224 | 3.215 | 3.184 | 3.175 |
| 0.01 | 3.211 | 3.182 | 3.184 | 3.175 |
| 0.1 | 3.213 | 3.258 | 3.282 | 3.343 |

### C.3    IMPACT OF THE STEP SIZE IN THE LANGEVIN PROPOSAL

We conduct an experiment to investigate the effect of the step size in the Langevin proposal for $(\mathbf{b}_{S_k,k}, \Gamma_{S_k,k}, \beta_k)$. Except for the step size, the other hyperparameters of Bayesian-TPNN are set identical to those in Section C.1 of Appendix, and we analyze ABALONE dataset. Table 11 presents the prediction performances of Bayesian-TPNN for various step sizes. Our results show that overly large step sizes in the Langevin proposal can degrade the prediction performance due to poor acceptance and unstable exploration, whereas a moderate range yields the best performance. Therefore, a not too large step size is recommended in practice.

Table 11: **Prediction performances of Bayesian-TPNN for various step sizes in the Langevin proposal .**

| Step size | 0.01 | 0.02 | 0.04 | 0.08 | 0.1 | 0.2 | 0.3 | 0.4 | 0.5 |
|---|---|---|---|---|---|---|---|---|---|
| RMSE | 3.199 | 3.216 | 3.209 | 3.269 | 3.160 | 3.243 | 4.308 | 4.549 | 4.578 |

## C.4 Impact of $p_{\text{INPUT}}$ on estimating higher-order components

We conduct an experiment to evaluate the effects of using $p_{\text{input}}$ other than the uniform distribution in the MH algorithm. We refer to the model with the uniform distribution for $p_{\text{input}}$ as Uniform Bayesian-TPNN, and the model where $p_{\text{input}}$ is determined using the feature importance from a pretrained XGB as Bayesian-TPNN. Table 12 compares prediction performances of Uniform Bayesian-TPNN (UBayesian-TPNN) and Bayesian-TPNN on MADELON dataset. To investigate why the prediction performance improvement occurs when using the nonuniform $p_{\text{input}}$, we identify the 5 most important components for each model whose results are presented in Table 13. UBayesian-TPNN only detects two thrid-order interactions as signals and ignores even all of the main effects. In contrast, Bayesian-TPNN captures the fourth-order component as the most important but is also able to capture other meaningful lower-order components including two main effects effectively.

We also analyze the synthetic datasets in Section 4.2 with UBayesian-TPNN, and the corresponding results are reported in Table 14. These results amply imply that $p_{\text{input}}$ plays an important role in detecting higher-order components and leading to substantial improvements in both prediction performance and component selection.

Table 12: **Prediction performance on MADELON dataset.**

| Model | UBayesian-TPNN | Bayesian-TPNN |
|---|---|---|
| AUROC ↑ (SE) | 0.739 (0.002) | **0.854** (0.007) |

Table 13: **Top 5 components with the important scores normalized by the maximum.**

| Model | Rank 1 | | Rank 2 | | Rank 3 | | Rank 4 | | Rank 5 | |
|---|---|---|---|---|---|---|---|---|---|---|
| | Comp. | Score | Comp. | Score | Comp. | Score | Comp. | Score | Comp. | Score |
| UBayesian-TPNN | (203,289,421) | 1.000 | (30,149,212) | 0.950 | (148,176,298) | 0.006 | (75,232,442) | 0.005 | (64,373,379) | 0.004 |
| Bayesian-TPNN | (49,242,319,339) | 1.000 | (129,443,494) | 0.472 | (379,443) | 0.374 | 106 | 0.322 | (242,443) | 0.301 |

Table 14: **Performance of component selection on the synthetic datasets.**

| True model | $f^{(1)}$ | | $f^{(2)}$ | | $f^{(3)}$ | |
|---|---|---|---|---|---|---|
| Order | UBayesian TPNN | Bayesian TPNN | UBayesian TPNN | Bayesian TPNN | UBayesian TPNN | Bayesian TPNN |
| 1 | **1.000** (0.000) | **1.000** (0.000) | 0.826 (0.024) | **0.831** (0.008) | 0.824 (0.009) | **1.000** (0.000) |
| 2 | 0.988 (0.010) | **1.000** (0.000) | 0.953 (0.006) | **0.985** (0.003) | 0.750 (0.006) | **0.922** (0.019) |
| 3 | 0.736 (0.050) | **0.740** (0.022) | 0.878 (0.020) | **0.966** (0.018) | 0.658 (0.011) | **0.661** 0.022 |

## C.5 Impact of stepwise search in the proposal of $K$

We conduct an experiment to evaluate the effectiveness of **Stepwise** move in the proposal distribution of $K$ suggested in Section 3.2. We compare the performances of Bayesian-TPNN with and without **Stepwise** move on MADELON dataset. Table 15 reports the averages and standard errors of AUROCs, ECEs, and NLLs over 5 trials and Table 16 shows the top 5 important components. The results suggest that the **Stepwise** move is helpful to detect higher-order interactions which in turn leads to improvements in both prediction performance and uncertainty quantification.

Table 15: **Results of performance with and without Stepwise move.**

|  | With **Stepwise** move | Without **Stepwise** move |
|---|---|---|
| AUROC ↑ (SE) | **0.854** (0.007) | 0.820 (0.002) |
| ECE ↓ (SE) | **0.076** (0.004) | 0.106 (0.007) |
| NLL ↓ (SE) | **0.479** (0.009) | 0.650 (0.005) |

Table 16: **Top 5 components with the important scores normalized by the maximum.**

| Model | Rank 1 | | Rank 2 | | Rank 3 | | Rank 4 | | Rank 5 | |
|---|---|---|---|---|---|---|---|---|---|---|
|  | Comp. | Score | Comp. | Score | Comp. | Score | Comp. | Score | Comp. | Score |
| With **Stepwise** move | (49,242,319,339) | 1.000 | (129,443,494) | 0.472 | (379,443) | 0.374 | 106 | 0.322 | (242,443) | 0.301 |
| Without **Stepwise** move | (129,242) | 1.000 | (29,339,379) | 0.986 | 339 | 0.622 | 337 | 0.544 | (242,443) | 0.526 |

# D    EXPERIMENT FOR THE POISSON REGRESSION

In this section, we compare the prediction performance and uncertainty quantification of Bayesian-TPNN with GBART (Linero, 2025) on the Poisson regression model. We consider the poisson regression model defined as

$$Y_i|\mathbf{x}_i \sim \text{Poisson}(\exp(f(\mathbf{x}_i))),$$

where $f$ is the regression function. We generate a synthetic dataset of size 15,000 using the true regression function $f_0$ defined as

$$f_0(\mathbf{x}) = \pi^{x_1 x_2}\sqrt{2|x_3|} - \sin^{-1}(0.5x_4) + \log(|x_3 + x_5| + 1) + \frac{x_9}{1 + |x_{10}|}\sqrt{\frac{x_7}{1 + |x_8|}} - x_2 x_7,$$

where input variable $\mathbf{x}_i \in \mathbb{R}^{10}$ are generated from $\text{Uniform}(0,1)^{10}$ for $i = 1, ..., 15,000$. Table 17 presents the RMSE and NLL for Bayesian-TPNN and GBART, demonstrating that Bayesian-TPNN achieves superior performance to GBART even in the Poisson regression. Here, the RMSE is calculated between $\exp(f_0(\mathbf{x}_i))$ and $\exp(\hat{f}(\mathbf{x}_i))$ for $i = 1, .., 15,000$, where $\hat{f}$ is the Bayes estimate. Figure 4 shows the scatter plot of predicted values $\exp(\hat{f}(\mathbf{x}_i))$ versus $\exp(f_0(\mathbf{x}_i))$ for $i = 1, ..., 15,000$. It implies that Bayesian-TPNN yields predictions much closer to the true values compared to GBART.

Table 17: **Prediction performance and uncertainty quantification on Poisson synthetic dataset.**

|  | Bayesian-TPNN | GBART |
|---|---|---|
| RMSE ↓ | **0.094** | 0.141 |
| NLL ↓ | **1.615** | 1.629 |

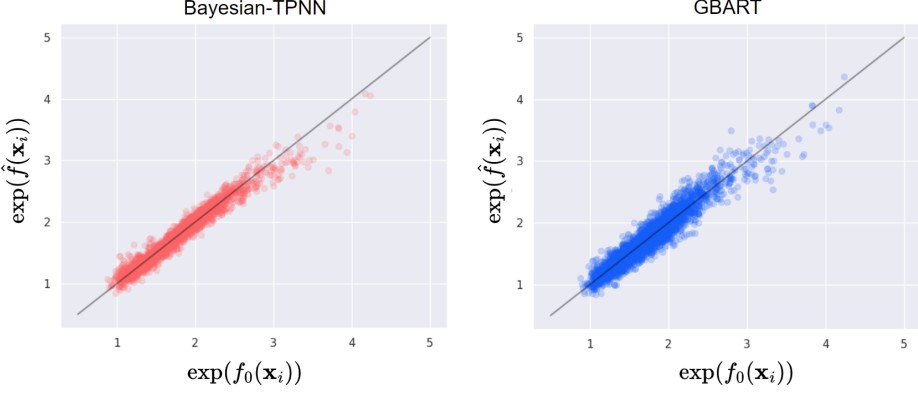

Figure 4: **Scatter Plots between the true expectations and estimated ones.**

# E EXPERIMENTS FOR INTERPRETABILITY

## E.1 INTERPRETABILITY ON THE IMAGE DATASETS

In this section, we describe the local and global interpretations of CBM (Koh et al., 2020) with Bayesian-TPNN on CELEBA-HQ and CATDOG datasets. Table 18 presents the description of concepts used in CELEBA-HQ and CATDOG datasets.

Table 18: **Description of image datasets.**

| Index | CELEBA-HQ | CATDOG |
|-------|-----------|--------|
| 1 | Facial hair | Pointed ear |
| 2 | Makeup | Short snout |
| 3 | Long hair | Almond eye |
| 4 | Angular contour | Slender/flexible body |
| 5 | Accessories | Fine/uniform fur |

Table 19: **Normalized importance scores and ranks for the top 5 important components on the image datasets.**

CELEBA-HQ

| | Rank | 1 | 2 | 3 | 4 | 5 |
|---|------|---|---|---|---|---|
| Bayesian-TPNN | Component index | 2 | 4 | (2,3) | (2,4) | (1,5) |
| | Score | 1.000 | 0.665 | 0.592 | 0.304 | 0.262 |
| ANOVA-$T^2$PNN | Component index | (2,3) | 1 | (1,5) | 4 | 5 |
| | Score | 1.000 | 0.482 | 0.284 | 0.262 | 0.211 |
| Linear | Component index | 2 | 1 | 4 | 5 | 3 |
| | Score | 1.000 | 0.783 | 0.549 | 0.328 | 0.304 |

CATDOG

| | Rank | 1 | 2 | 3 | 4 | 5 |
|---|------|---|---|---|---|---|
| Bayesian-TPNN | Component index | 3 | (3,4) | 2 | 4 | (2,3,4,5) |
| | Score | 1.000 | 0.395 | 0.252 | 0.162 | 0.086 |
| ANOVA-$T^2$PNN | Component index | (4,5) | 3 | (3,5) | 4 | (1,4) |
| | Score | 1.000 | 0.883 | 0.882 | 0.716 | 0.453 |
| Linear | Component index | 5 | 1 | 3 | 2 | 4 |
| | Score | 1.000 | 0.698 | 0.352 | 0.023 | 0.021 |

**Global interpretation.** Table 19 shows the top 5 most important components along with their importance scores (normalized by the maximum score) for Bayesian-TPNN, ANOVA-$T^2$PNN and Linear model. In CATDOG dataset, Bayesian-TPNN identifies the 4th-order component (2,3,4,5) as an important component. It seems that complex interactions exists between the 5 concepts.

**Example of CelebA-HQ**   **Example of CatDog**

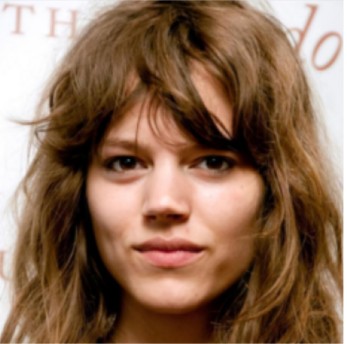 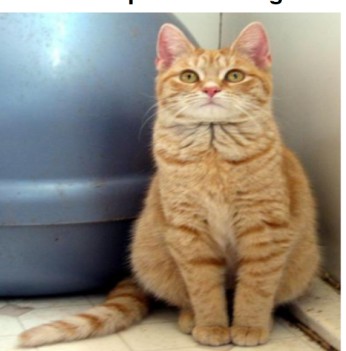

Figure 5: **Examples of images misclassified by Linear model.**

**Local interpretation.** Figure 5 presents two images where Bayesian-TPNN correctly classifies but Linea model does not. For the CELEBA-HQ example image, Linear model incorrectly predicts it as male, whereas the Bayesian-TPNN correctly predicts as female. The contributions of the important components for this image are presented in Table 20. In Linear model, 'Makeup' gives a positive contribution, which leads to a misclassification of the image as male. In contrast, in Bayesian-TPNN, while the main effect of 'Makeup' still provides a positive contribution, the interactions between ('Makeup', 'Long hair') and ('Makeup', 'Angular contour') yield negative contributions, resulting in a correct prediction as female.

For the CATDOG example image, Linear model incorrectly predicts it as 'dog', whereas Bayesian-TPNN correctly predicts as 'cat'. Table 21 indicates that Linear model misclassifis the image as 'dog' due to the positive contribution of 'Almond eye'. In contrast, although Bayesian-TPNN also assigns a positive contribution to 'Almond eye', the higher-order interactions—('Almond eye', 'Slender/flexible body') and ('Short snout', 'Almond eye', 'Slender/flexible body', 'Fine/uniform fur')—provided much stronger negative contributions, leading to the correct classification as a cat.

These two examples strongly suggest that considering higher-order interactions between concepts is necessary for the success of CBM.

Table 20: **Prediction values of the 5 most important components for CELEBA-HQ image.**

| Bayesian-TPNN | Component index | 2 | 4 | (2,3) | (2,4) | (1,5) |
|---|---|---|---|---|---|---|
| | Contribution | 0.297 | 0.184 | -0.444 | -0.323 | -0.207 |
| Linear | Component index | 1 | 2 | 3 | 4 | 5 |
| | Contribution | -0.222 | 3.746 | -1.510 | -2.665 | 1.627 |

Table 21: **Prediction values of the 5 most important components for CATDOG image.**

| Bayesian-TPNN | Component | 3 | (3,4) | 2 | 4 | (2,3,4,5) |
|---|---|---|---|---|---|---|
| | Contribution | 0.618 | -0.767 | 0.181 | -0.778 | -0.355 |
| Linear | Component | 1 | 2 | 3 | 4 | 5 |
| | Contribution | -4.304 | -0.630 | 9.503 | -2.463 | -4.113 |

Table 22: **Prediction performance on the image datasets.**

| | Bayesian-TPNN with 5 concepts | Linear with 10 concepts |
|---|---|---|
| CELEBA-HQ | **0.936** (0.002) | 0.899 (0.001) |
| CATDOG | **0.878** (0.002) | 0.869 (0.002) |

**Fewer concepts, better prediction performance.** One may argue that 5 concepts are too small for Linear model and Linear model would perform well with more concepts. To see the validity of this argument, we compare predictive performance of Bayesian-TPNN with 5 concept and Linear model with 10 concepts, where additional 5 concepts are generated through GPT-5: for CELEBA-HQ dataset,

{'Emphasized eyes', 'Prominent lips', 'Smooth skin',

'Pronounced cheekbones', 'High contrast'}

and for CATDOG dataset,

{'Long tail', 'Retractable claws (hidden)', 'Upright sitting or crouching posture',

'Small mouth / Meowing shape', 'Ambush-like pose (crouched)'}.

Table 22 presents the averages and standrad errors of AUROCs for Bayesian-TPNN with 5 concepts and Linear model with 10 concepts. While using more concepts with Linear model improves prediction accuracy, Bayesian-TPNN is still superior to Linear model even though fewer concepts are used in Bayesian-TPNN. This implies that capturing higher-order interactions plays a more critical role in improving prediction performance than merely increasing the number of concepts. Quality of concepts generated by GPT would be problematic.

### E.2 ADDITIONAL RESULTS OF LOCAL INTERPRETATION ON THE TABULAR DATASET

In this section, we describe the results of local interpretation on BOSTON dataset. Specifically, we examine the contributions of the 5 most important components identified by Bayesian-TPNN in Section 4.3 at a specific input vector $\mathbf{x}$. For a given data point

$$\mathbf{x} = (0.006, 18, 2.31, 0, 0.538, 6.58, 65.2, 4.09, 1, 296, 15.3, 396.9, 4.98),$$

the contributions of the 5 estimated components $\hat{f}_{\{13\}}, \hat{f}_{\{6\}}, \hat{f}_{\{1\}}, \hat{f}_{\{8\}}$, and $\hat{f}_{\{1,6\}}$ by Bayesian-TPNN are given as

$$(\hat{f}_{\{13\}}(\mathbf{x}), \hat{f}_{\{6\}}(\mathbf{x}), \hat{f}_{\{1\}}(\mathbf{x}), \hat{f}_{\{8\}}(\mathbf{x}), \hat{f}_{\{1,6\}}(\mathbf{x})) = (0.575, -0.108, 0.080, -0.002, -0.001).$$

In particular, the component $\hat{f}_{\{13\}}$ makes a substantial positive contribution to the housing price. That is, the price of the house for a given input vector $\mathbf{x}$ is high because of the main effect $x_{13}$.

## F    EXPERIMENT FOR STABILITY OF COMPONENT ESTIMATION

Park et al. (2025) demonstrated, both theoretically and empirically, that TPNN reliably estimates the components of the functional ANOVA model. In this section, we investigate whether Bayesian-TPNN exhibits the same stability in component estimation. For this analysis, we randomly split the dataset into training and test datasets. From this, we obtain estimators for the components. We repeat this procedure five times to obtain five estimators for each component. We then calculate the stability score using these estimators. Specifically, following Park et al. (2025), for predefined data $\{\mathbf{x}_1, ..., \mathbf{x}_n\}$, we use the stability score defined as

$$\mathcal{SC}(f_S) := \frac{1}{n} \sum_{i=1}^{n} \frac{\sum_{j=1}^{5}(f_S^j(\mathbf{x}_i) - \bar{f}_S(\mathbf{x}_i))^2}{\sum_{j=1}^{5}(f_S^j(\mathbf{x}_i))^2},$$

where $f_S^j$ is the estimated component for $S$ obtained from the $j$-th fold and $\bar{f}_S(\mathbf{x}) = \sum_{j=1}^{5} f_S^j(\mathbf{x})/5$. Finally, we use $\mathcal{SC}^d(f) := \frac{1}{\sum_{k=1}^{d}\binom{p}{k}} \sum_{S \subseteq [p], |S| \leq d} \mathcal{SC}(f_S)$ to compare the stability in component estimation between Bayesian-TPNN, ANOVA-TPNN and NAM.

Table 23 presents the results of stability scores $\mathcal{SC}^1(f)$ for Bayesian-TPNN, ANOVA-T[1]PNN and NA[1]M, where ANOVA-T[1]PNN and NA[1]M estimate only the main effects. Table 24 shows of stability scores $\mathcal{SC}^2(f)$ for Bayesian-TPNN, ANOVA-T[2]PNN and NA[2]M, where ANOVA-T[2]PNN and NA[2]M estimate up to second-order components. These results imply that Bayesian-TPNN estimates the components more stably than ANOVA-TPNN and NAM. Note that for MADELON dataset, which has an input dimension of 500, we could not train ANOVA-T[2]PNN and NA[2]M due to the computational environment, and thus their stability scores could not be calculated.

Table 23: **Stability scores of Bayesian-TPNN, ANOVA-T[1]PNN and NA[1]M.**

| Dataset | Bayesian TPNN | ANOVA T[1]PNN | NA[1]M |
|---------|:---:|:---:|:---:|
| ABALONE | **0.087** | 0.405 | 0.555 |
| BOSTON  | **0.368** | 0.425 | 0.583 |
| MPG     | **0.222** | 0.411 | 0.472 |
| SERVO   | **0.339** | 0.651 | 0.481 |
| FICO    | **0.130** | 0.287 | 0.607 |
| BREAST  | **0.100** | 0.286 | 0.569 |
| CHURN   | **0.111** | 0.558 | 0.569 |
| MADELON | **0.520** | 0.685 | 0.785 |

Table 24: **Stability scores of Bayeisan-TPNN, ANOVA-T[2]PNN and NA[2]M.**

| Dataset | Bayesian TPNN | ANOVA T[2]PNN | NA[2]M |
|---------|:---:|:---:|:---:|
| ABALONE | 0.400 | **0.340** | 0.770 |
| BOSTON  | 0.615 | **0.380** | 0.705 |
| MPG     | **0.340** | 0.370 | 0.560 |
| SERVO   | **0.445** | 0.575 | 0.665 |
| FICO    | **0.525** | 0.540 | 0.790 |
| BREAST  | **0.630** | 0.675 | 0.730 |
| CHURN   | **0.520** | 0.755 | 0.730 |
| MADELON | **0.475** | — | — |

## G    COMPARISON OF CONVERGENCE SPEED AND RUNTIME IN MCMC ALGORITHM

In this section, we evaluate the convergence speed and runtime of MCMC algorithms for Bayesian-TPNN. Specifically, we compare its convergence speed with that of mBNN, and its runtime with those of ANOVA-T$^2$PNN and mBNN. In Bayesian-TPNN, we set $K_{\max} = 100$. For mBNN, we use two hidden layers with 500 units each and set the number of HMC steps to 30. For ANOVA-T$^2$PNN, we set $K_S = 10$.

Figure 6 shows the RMSE trajectories across MCMC iterations on BOSTON dataset for Bayesian-TPNN and mBNN. Table 25 presents the runtime comparison of Bayesian-TPNN , mBNN with 2,000 iterations and ANOVA-T$^2$PNN with 2,000 epochs on real datasets. The best results are highlighted by **bold**. In the experiments on FICO, CHURN, and BREAST datasets, the runtime difference between Bayesian-TPNN and ANOVA-T$^2$PNN become more pronounced. This is because, after data preprocessing, the input dimensions are 23, 46, and 30, respectively. As the number of neural networks required in ANOVA-T$^2$PNN increases rapidly with the input dimension, the runtime increases considerably. Note that for the MADELON dataset, where the input dimension is 500, training ANOVA-T$^2$PNN is infeasible because the number of neural networks to be trained is $125, 250$.

These results imply that Bayesian-TPNN converges faster in terms of MCMC iterations compared to mBNN. Moreover, its overall runtime is shorter than both mBNN and ANOVA-T$^2$PNN. In particular, Bayesian-TPNN runs significantly faster than ANOVA-T$^2$PNN, and this advantage becomes more pronounced as the input dimension $p$ increases.

Table 25: **Runtime of Bayesian-TPNN, ANOVA-T$^2$PNN and mBNN (sec).**

| Dataset | Bayesian-TPNN | ANOVA-T$^2$PNN | mBNN |
|---|---|---|---|
| ABALONE | 475 | **326** | 1,273 |
| BOSTON | **181** | 577 | 266 |
| MPG | **156** | 227 | 275 |
| SERVO | **159** | 400 | 242 |
| FICO | **943** | 3,530 | 4,198 |
| BREAST | **181** | 2,363 | 310 |
| CHURN | **686** | 7,772 | 2,756 |
| MADELON | **345** | — | 894 |

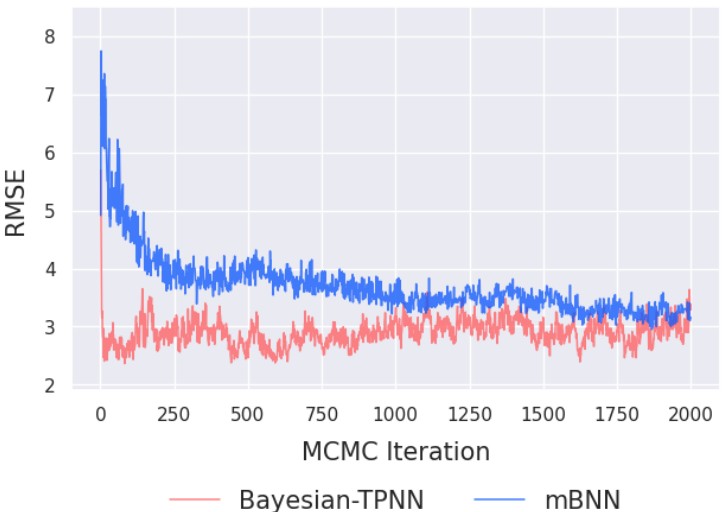

Figure 6: **The RMSE trajectories across MCMC iterations for Bayesian-TPNN and mBNN.**

# H  ADDITIONAL EXPERIMENTS FOR UNCERTAINTY QUANTIFICATION

## H.1  UNCERTAINTY QUANTIFICATION ON NON-BAYESIAN MODELS.

We report the performance of uncertainty quantification for non-Bayesian models including ANOVA-TPNN, NAM, XGB and Linear model, in Table 26. These results indicate that Bayesian-TPNN outperforms the non-bayesian models in view of uncertainty quantification.

Table 26: **Uncertainty quantifications for non-bayesian models on real datasets.**

| Dataset | ANOVA-TPNN | | NAM | | XGB | | Linear | |
|---|---|---|---|---|---|---|---|---|
| | CRPS | NLL | CRPS | NLL | CRPS | NLL | CRPS | NLL |
| ABALONE | 1.578 (0.16) | — | 1.901 (0.27) | — | 1.668 (0.16) | — | 1.638 (0.15) | — |
| BOSTON | 4.464 (0.71) | — | 3.147 (0.35) | — | 3.241 (0.27) | — | 4.291 (0.44) | — |
| MPG | 2.478 (0.45) | — | 3.314 (1.07) | — | 2.343 (0.35) | — | 2.990 (0.32) | — |
| SERVO | 0.595 (0.02) | — | 0.868 (0.39) | — | 0.215 (0.03) | — | 0.910 (0.04) | — |
| | ECE | NLL | ECE | NLL | ECE | NLL | ECE | NLL |
| FICO | 0.063 (0.017) | 0.583 (0.018) | 0.198 (0.007) | 0.681 (0.012) | 0.096 (0.026) | 0.620 (0.015) | 0.055 (0.014) | 0.593 (0.017) |
| BREAST | 0.100 (0.030) | 0.423 (0.071) | 0.284 (0.022) | 0.511 (0.033) | 0.063 (0.012) | 0.878 (0.172) | 0.102 (0.015) | 0.216 (0.039) |
| CHURN | 0.053 (0.004) | 0.444 (0.011) | 0.318 (0.007) | 0.718 (0.008) | 0.131 (0.006) | 0.594 (0.021) | 0.078 (0.004) | 0.573 (0.002) |
| MADELON | 0.354 (0.014) | 0.752 (0.003) | 0.156 (0.009) | 0.735 (0.016) | 0.147 (0.008) | 0.703 (0.035) | 0.232 (0.011) | 0.736 (0.016) |

## H.2  EXPERIMENT FOR OUT-OF-DISTRIBUTION DETECTION

Here, we conduct experiments to evaluate whether each model appropriately captures uncertainty on out-of-distribution data in binary classification. As a measure of uncertainty for out-of-distribution data, we use the maximum predicted probability (Mukhoti et al., 2023). Specifically, we denote the in-distribution dataset by $\{\mathbf{x}_1^{\text{in}}, ..., \mathbf{x}_{N_1}^{\text{in}}\}$ and the out-of-distribution dataset by $\{\mathbf{x}_1^{\text{out}}, ..., \mathbf{x}_{N_2}^{\text{out}}\}$ with corresponding predictive probabilities $\hat{p}(\mathbf{x}_i^{\text{in}})$ for $i = 1, ..., N_1$ and $\hat{p}(\mathbf{x}_i^{\text{out}})$ for $i = 1, ..., N_2$.

Let $\hat{p}_{\max}(\mathbf{x}) = \max\{\hat{p}(\mathbf{x}), 1 - \hat{p}(\mathbf{x})\}$. For evaluation, we assign label 1 to the in-distribution data and label 0 to the out-of-distribution data. Then, we compute the AUROC between the labels and the transformed scores $1 + \log_2 \hat{p}_{\max}(\mathbf{x}_i^{\text{in}})$ or $1 + \log_2 \hat{p}_{\max}(\mathbf{x}_i^{\text{out}})$. Intuitively, predictive probabilities close to 0.5 reflect model uncertainty, and such cases can be identified as out-of-distribution.

We randomly sample a subset which size of 500 from the MADELON dataset, standardized it, and use it as an out-of-distribution dataset. For each dataset FICO, BREAST, and CHURN, we randomly split the data into training and test datasets. In turn, we train Bayesian-TPNN and baseline models using the training dataset. We then compute the AUROC treating the test dataset as the in-distribution dataset. We repeat this procedure 5 times, and Table 27 presents the averages and standard errors of AUROCs for Bayesian-TPNN and baseline models on FICO, BREAST and CHURN datasets. These results demonstrate that Bayesian-TPNN outperforms the baseline models, achieving substantially superior performance in out-of-distribution detection.

Table 27: **AUROC Results on in-distribution and out-of-distribution detection.**

| Dataset | Bayesian-TPNN | ANOVA-TPNN | NAM | Linear | XGB | BART | mBNN |
|---|---|---|---|---|---|---|---|
| FICO | 0.606 (0.013) | 0.446 (0.020) | 0.455 (0.032) | 0.191 (0.002) | 0.605 (0.018) | **0.667** (0.004) | 0.519 (0.014) |
| BREAST | **0.903** (0.015) | 0.542 (0.021) | 0.534 (0.041) | 0.112 (0.010) | 0.827 (0.022) | 0.664 (0.023) | 0.503 (0.051) |
| CHURN | **0.724** (0.006) | 0.570 (0.040) | 0.533 (0.040) | 0.442 (0.006) | 0.420 (0.014) | 0.598 (0.009) | 0.599 (0.039) |

# I   VISUAL ILLUSTRATION FOR PROPOSAL

In this section, we describe the visual explanation of the proposal in Section 3.2. Given Bayesian-TPNN as in Figure 7, we explain the updating of $K$ and the updating of $S_K$.

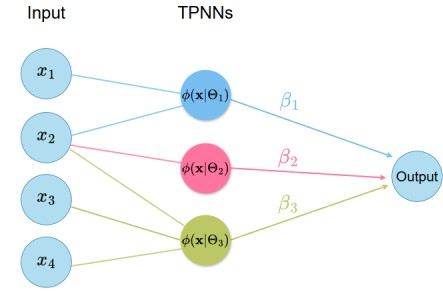

Figure 7: **Bayesian-TPNN with** $p = 4, K = 3$.

## I.1   UPDATING $K$

For a given $K$, we propose $K^{\text{new}} = K - 1$ or $K^{\text{new}} = K + 1$. Here, we describe only **Random** and **Stepwise** moves, corresponding to the case where $K^{\text{new}} = K + 1$.

In the case of **Random** move, a node is randomly generated and its edges are randomly assigned. For **Stepwise** move, a node is first selected from the existing nodes, and then a new edge is added to create a new node. Figure 8 presents an overall illustration for these moves.

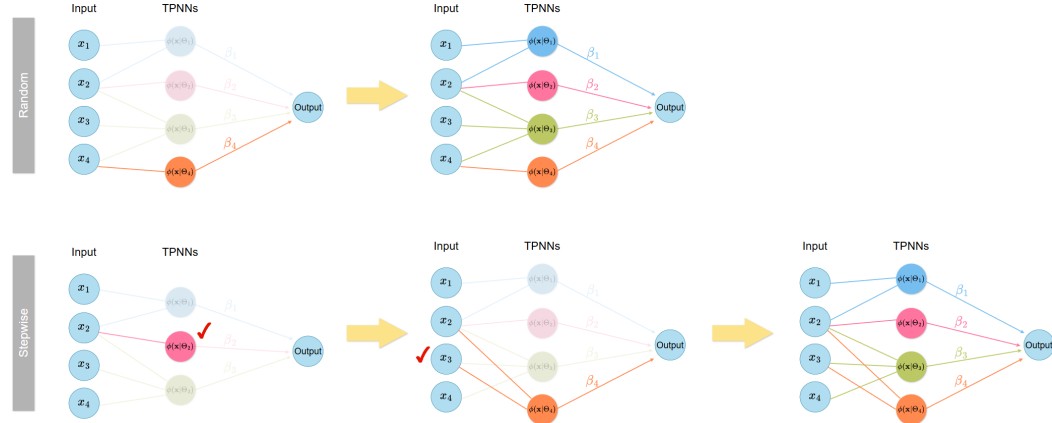

Figure 8: **Visual explanation for alternations in the proposal distribution of** $K$.

## I.2   UPDATING $S_k$

Figure 9 illustrates how the edges change when applying **Adding**, **Deleting**, or **Changing** moves to a given current state.

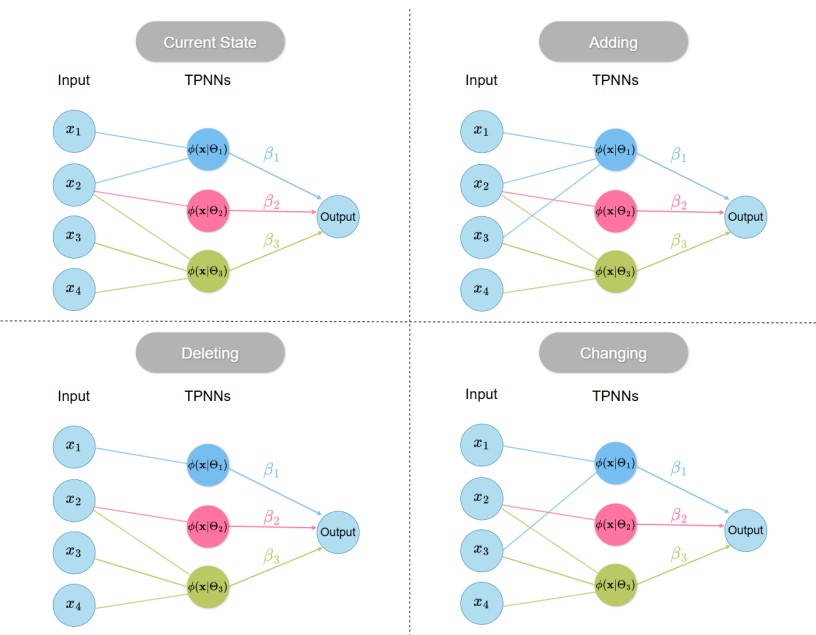

Figure 9: **Visual explanation for alternations in the proposal distribution of $S_k$.**

## J EMPIRICAL EVALUATION UNDER MINIBATCH SETTINGS

We conduct an additional experiment to empirically verify that our MCMC algorithm performs well when mini-batches are used. When estimating Bayesian-TPNN with mini-batched data, we refer to it as MBayesian-TPNN. Here, for ABALONE and FICO datasets, we set the size of mini-batch as 1,000 and 2,000, respectively. Table 28 presents the averages and standard errors of prediction performance and the uncertainty quantifications of Bayesian-TPNN and MBayesian-TPNN for 5 trials on ABALONE and FICO datasets. These results suggest that training with mini-batches does not significantly reduce prediction performance and uncertainty quantification. In practice, these findings indicate that using mini-batches is practically acceptable, as it does not lead to meaningful degradation in performance or uncertainty estimation.

Table 28: Results of MBayesian-TPNN.

| | RMSE/AUROC | | CRPS/ECE | | NLL | |
|---|---|---|---|---|---|---|
| | Bayesian-TPNN | MBayesian-TPNN | Bayesian-TPNN | MBayesian-TPNN | Bayesian-TPNN | MBayesian-TPNN |
| ABALONE | 2.053 (0.26) | 2.081 (0.24) | 1.372 (0.19) | 1.391 (0.17) | 2.260 (0.16) | 2.280 (0.18) |
| FICO | 0.793 (0.009) | 0.788 (0.005) | 0.036 (0.004) | 0.038 (0.003) | 0.554 (0.007) | 0.564 (0.003) |

# K    COMPARISON WITH DEEP ENSEMBLE

In this section, we conduct additional experiment to compare Bayesian-TPNN with Deep Ensemble (Lakshminarayanan et al., 2017). Here, we consider candidates for each hyperparmeter of Deep Ensemble as below.

- The number of MLPs : $\{5, 50, 100\}$
- MLP architectures : $\{(50), (100), (256, 128, 64), (512, 256, 128)\}$
- Learning rates : $\{1e - 4, 1e - 3, 1e - 2\}$
- Epochs : $\{100, 200, 500, 1000\}$
- Weight for $L_2$ regularization : $\{1e - 3, 1e - 2, 1e - 1\}$

Table 29 presents the averages of RMSE, AUROC, CRPS, ECE and NLLs for 5 trials on real datasets. These results show that the performance of Bayesian-TPNN is comparable to that of Deep Ensemble in terms of both prediction accuracy and uncertainty quantification.

Table 29: Results of Bayesian-TPNN and Deep Ensemble.

| | RMSE/AUROC | | CRPS/ECE | | NLL | |
|---|---|---|---|---|---|---|
| | Bayesian-TPNN | Deep Ensemble | Bayesian-TPNN | Deep Ensemble | Bayesian-TPNN | Deep Ensemble |
| ABALONE | **2.053** (0.26) | 2.121 (0.23) | **1.372** (0.19) | 1.498 (0.17) | 2.260 (0.16) | **2.036** (0.15) |
| BOSTON | **3.654** (0.49) | 3.922 (0.57) | **2.202** (0.23) | 2.458 (0.22) | **3.411** (0.37) | 3.747 (0.40) |
| MPG | 2.386 (0.41) | **2.257** (0.14) | 1.510 (0.43) | **1.481** (0.11) | **2.511** (0.21) | 2.769 (0.47) |
| SERVO | **0.351** (0.02) | 0.398 (0.03) | 0.194 (0.01) | **0.179** (0.01) | 0.836 (0.10) | **0.701** (0.04) |
| FICO | **0.793** (0.009) | 0.773 (0.024) | **0.036** (0.004) | 0.057 (0.033) | **0.554** (0.007) | 0.577 (0.034) |
| BREAST | **0.998** (0.001) | 0.993 (0.003) | 0.129 (0.009) | **0.075** (0.017) | 0.211 (0.014) | **0.133** (0.041) |
| CHURN | **0.849** (0.008) | 0.841 (0.013) | **0.031** (0.001) | 0.039 (0.002) | **0.418** (0.008) | 0.424 (0.018) |
| MADELON | **0.854** (0.013) | 0.616 (0.029) | **0.076** (0.004) | 0.137 (0.061) | **0.478** (0.009) | 0.719 (0.049) |

## L  Applications to Genomic Dataset

We conduct additional experiment to explore the applicability of Bayesian-TPNN to genomics dataset GSE43358 (Fumagalli et al., 2013). GSE43358 is a gene expression dataset with $n = 57$ samples and $p = 54,675$ features and we perform a classification task distinguishing between HER2-positive and non–HER2-positive cases. Table 30 shows that the averages and standard errors of prediction performance for Bayesian-TPNN, Linear model and XGB for 5 trials. For Bayesian-TPNN and XGB, the hyperparameters are optimized as in the experiment for other real datasets. Note that because the input dimension $p$ is too large, both ANOVA-TPNN and NAM could not be trained within our computational environment. The results in Table 30 indicate that the interpretable Bayesian-TPNN achieves prediction performance comparable to that of the black-box model XGB on GSE43358 dataset.

Table 31 reports the top 10 most important components in Bayesian-TPNN with the normalized importance score. Here, we use the importance score defined in Section 4.2, and the normalized score represents each importance value divided by the maximum importance score. Note that one of the third order interactions is detected by Bayesian-TPNN. The results in Table 31 indicate that higher-order interactions (beyond the second order) play a crucial role, which provides a plausible explanation for the inferior prediction performance of the linear model. Moreover, this highlights the necessity of an interpretable model such as Bayesian-TPNN, which is capable of estimating such higher-order interactions.

Table 30: Results of baseline models on GSE43358 dataset.

| Model | Bayesian-TPNN | ANOVA-TPNN | NAM | Linear | XGB |
|-------|---------------|------------|-----|--------|-----|
| AUROC | 0.949 (0.017) | – | – | 0.545 (0.001) | 0.953 (0.041) |

Table 31: Top 10 important components.

| Rank | Component of GenBank accession numbers | Normalized Score |
|------|----------------------------------------|------------------|
| 1 | S69189 | 1.000 |
| 2 | BF357738 | 0.924 |
| 3 | (BC000129, R80390) | 0.701 |
| 4 | AF307338 | 0.569 |
| 5 | NM_018297 | 0.410 |
| 6 | BF061275 | 0.375 |
| 7 | AF319440 | 0.365 |
| 8 | (BE741754, AB037854, AK024890) | 0.334 |
| 9 | AI368358 | 0.292 |
| 10 | BE672684 | 0.218 |

# M NOTATIONS AND REGULARITY CONDITIONS FOR THE PROOFS

## M.1 ADDITIONAL NOTATIONS

For two positive sequences $\{a_n\}$ and $\{b_n\}$, we write $a_n \lesssim b_n$ if there exists a constant $C > 0$ such that $a_n \leq Cb_n$ for all $n \in \mathbb{N}$. The notation $a_n = o(b_n)$ indicates that the ratio $a_n/b_n$ converges to zero as $n \to \infty$. We denote $\mathcal{N}(\epsilon, \mathcal{F}, d)$ the $\epsilon$-covering number of the function class $\mathcal{F}$ with respect to the semimetric $d$. For a given vector $\mathbf{v} = (v_1, ..., v_N)$, we define its $\ell_2$ norm as $\|\mathbf{v}\|_2^2 := \sum_{i=1}^{N} v_i^2$. Given a real-valued function $f : \mathcal{X} \to \mathbb{R}$, we define its sup-norm as $\|f\|_\infty := \sup_{\mathbf{x} \in \mathcal{X}} |f(\mathbf{x})|$. We define population $\ell_p$-norm with respect to a probability measure $\mu$ on $\mathcal{X}$ as $\|f\|_{p,\mu} := (\int_{\mathbf{x} \in \mathcal{X}} f(\mathbf{x})^p \mu(d\mathbf{x}))^{1/p}$. Let $\mathbb{P}_{\mathbf{X}}^n = \prod_{i=1}^{n}$, where $\mathbb{P}_{\mathbf{X}_i}$ is the probability distribution of $\mathbf{X}_i$ for $i = 1, ..., n$. For two given densities $p_1$ and $p_2$, we define the Kullback-Leibler (KL) divergence as

$$K(p_1, p_2) := \int \log(p_1(\mathbf{v})/p_2(\mathbf{v}))p_1(\mathbf{v})d\mathbf{v},$$

and let $V(p_1, p_2) := \int |\log(p_1(\mathbf{v})/p_2(\mathbf{v})) - K(p_1, p_2)|^2 p_1(\mathbf{v})d\mathbf{v}$.

## M.2 REGULARITY CONDITIONS

$(S.1)$ For a distribution $\mathbb{P}_{\mathbf{X}}$, there exist a density $p_{\mathbf{X}}$ with respect to the Lebesgue measure on $\mathbb{R}^p$, that is bounded away from zero and infinity, i.e.,

$$0 < \inf_{\mathbf{x} \in \mathcal{X}} p_{\mathbf{X}}(\mathbf{x}) \leq \sup_{\mathbf{x} \in \mathcal{X}} p_{\mathbf{X}}(\mathbf{x}) < \infty.$$

$(S.2)$ The true function $f_{0,S}$ is L-Lipschitz continuous, i.e.,

$$|f_{0,S}(\mathbf{x}) - f_{0,S}(\mathbf{x}')| \leq L\|\mathbf{x} - \mathbf{x}'\|_2$$

for some positive constant $L$ and all $\mathbf{x}, \mathbf{x}' \in \mathcal{X}$. Additionally, $f_{0,S}$ is assumed to be bounded in the supremum norm by a positive constant $F$, i.e., $\|f_{0,S}\|_\infty \leq F$. We denote the above conditions compactly as $f_{0,S} \in \text{Lip}_{L,F}$. Moreover, we say that $f_0 \in \text{Lip}_{0,L,F}$ if $f_{0,S} \in \text{Lip}_{L,F}$ for all $S \subseteq [p]$.

$(S.3)$ The log-partition function $A(\cdot)$ is differentiable with a bounded second derivative over $[-F, F]$, i.e., there exists a positive constant $C_A$ such that

$$1/C_A \leq \ddot{A}(x) \leq C_A$$

for all $x \in [-F, F]$.

$(S.4)$ $K_{\max}$ is assumed to grow at a rate $K_{\max} = O(n)$.

# N POSTERIOR CONSISTENCY OF BAYESIAN-TPNN

We first prove the posterior consistency of $f$ since it plays an important role in the proof of the posterior consistency of each component $f_S$.

## N.1 POSTERIOR CONSISTENCY OF $f_0$

**Theorem N.1** (Posterior Consistency of Bayesian-TPNN)**.** *We assumes that* $(S.1)$, $(S.2)$, $(S.3)$ *and* $(S.4)$*. Then, for any* $\varepsilon > 0$ *and* $\xi \geq 2^p F + \varepsilon\sqrt{\frac{2}{C_A}}$*, it holds that*

$$\pi_\xi\left(f : \|f_0 - f\|_{2,n} > \varepsilon \Big| \mathbf{X}^{(n)}, Y^{(n)}\right) \to 0 \tag{15}$$

*in* $\mathbb{Q}_0^n$ *as* $n \to \infty$*, where* $\mathbb{Q}_0^n$ *is the probability distribution of* $(\mathbf{X}^{(n)}, Y^{(n)})$*.*

## N.2 PROOF OUTLINE

Consider a function class $\mathcal{F} = \bigcup_{K=1}^{K_{\max}} \mathcal{F}(K)$ that satisfies the sum-to-zero condition with respect to uniform distribution on (0,1). Here, $\mathcal{F}(K)$ is defined as

$$\mathcal{F}(K) = \Bigg\{ f : f(\mathbf{x}) = \sum_{k=1}^K \beta_k \phi(\mathbf{x}|S_k, \mathbf{b}_{S_k,k}, \Gamma_{S_k,k}),$$
$$\beta_k \in \mathbb{R},$$
$$\mathbf{b}_{S_k,k} \in [0,1]^{|S_k|},$$
$$\Gamma_{S_k,k} \in [1/n, \infty)^{|S_k|} \text{ for } k = 1, ..., K \Bigg\},$$

where

$$\phi(\mathbf{x}|S_k, \mathbf{b}_{S_k,k}, \Gamma_{S_k}, k) = \prod_{j \in S_k} \left( 1 - \sigma\left(\frac{x_j - b_{j,k}}{\gamma_{j,k}}\right) + c_j(b_{j,k}, \gamma_{j,k})\sigma\left(\frac{x_j - b_{j,k}}{\gamma_{j,k}}\right) \right)$$

and

$$c_j(b_{j,k}, \gamma_{j,k}) = -\frac{1 - \int_0^1 \sigma\left(\frac{x_j - b_{j,k}}{\gamma_{j,k}}\right) dx_j}{\int_0^1 \sigma\left(\frac{x_j - b_{j,k}}{\gamma_{j,k}}\right) dx_j}.$$

For any $f \in \mathcal{F}(K)$, we denote it as $f_{K,\mathcal{B},\mathbf{b},\Gamma}$, where

$$\mathcal{B} = (\beta_k, k \in [K]), \quad \mathbf{b} = (\mathbf{b}_{S_k,k}, k \in [K]) \quad \text{and} \quad \Gamma = (\Gamma_{S_k,k}, k \in [K]).$$

Our goal is to show that

$$\lim_{n\to\infty} \mathbb{E}_0^n[\pi_\xi(\|f - f_0\|_{2,n} > \varepsilon|\mathbf{X}^{(n)}, Y^{(n)})] = 0 \tag{16}$$

for any $\varepsilon > 0$.

We prove (16) using following two steps.

$(P.1)$ For given data $\mathbf{x}^{(n)}$, we prove that

$$\lim_{n\to\infty} \mathbb{E}_0^n[\pi_\xi(\|f - f_0\|_{2,n} > \varepsilon|\mathbf{X}^{(n)}, Y^{(n)})|\mathbf{X}^{(n)} = \mathbf{x}^{(n)}] = 0$$

for any $\varepsilon > 0$.

$(P.2)$ Finally, we show that

$$\lim_{n\to\infty} \mathbb{E}_0^n[\pi_\xi(\|f - f_0\|_{2,n} > \varepsilon|\mathbf{X}^{(n)}, Y^{(n)})] = 0$$

for any $\varepsilon > 0$.

We first verify the following three conditions: there exists $\mathcal{F}^n \subseteq \mathcal{F}$ and positive constants $\delta, C_1, C_2$ such that

$$\log \mathcal{N}\left(\delta, \mathcal{F}^n, \|\cdot\|_\infty\right) < nC_1, \tag{17}$$

$$\pi\left(f \in \mathcal{F} : \|f - f_0\|_\infty \le \varepsilon\sqrt{\frac{2}{C_A}}\right) > \exp(-nC_2), \tag{18}$$

$$\pi(\mathcal{F}\backslash\mathcal{F}^n) < \exp(-(2C_2 + 2)n). \tag{19}$$

After that, we will show that these three conditions imply the posterior consistency in Step $(P.1)$ by checking the conditions in Ghosal et al. (1999).

### N.3 VERIFYING CONDITION (17)

We consider a sieve $\mathcal{F}^n = \cup_{K=1}^{M_n}\mathcal{F}^n(K)$, where

$$\mathcal{F}^n(K) = \left\{f : f(\mathbf{x}) = \sum_{k=1}^K \beta_k\phi(\mathbf{x}|S_k, \mathbf{b}_{S_k,k}, \Gamma_{S_k,k}),\right.$$
$$\beta_k \in [-n, n],$$
$$\mathbf{b}_{S_k,k} \in [0,1]^{|S_k|}$$
$$\left.\Gamma_{S_k,k} \in [1/n, n]^{|S_k|} \quad \text{for} \quad k = 1,..,K\right\},$$

where $M_n = \lfloor\frac{C_3 n\varepsilon^2}{\log n}\rfloor$ and $C_3$ will be determined later.

Also, we consider a more general function class as :

$$\mathcal{G}^n(K) = \left\{f : f(\mathbf{x}) = \sum_{k=1}^K \beta_k\phi(\mathbf{x}|S_k, \mathbf{b}_{S_k,k}, \Gamma_{S_k,k}, \mathbf{c}_{S_k,k}),\right.$$
$$\beta_k \in [-n, n],$$
$$\mathbf{b}_{S_k,k} \in [0,1]^{|S_k|}, \tag{20}$$
$$\Gamma_{S_k,k} \in [1/n, n]^{|S_k|},$$
$$\left.\mathbf{c}_{S_k,k} \in [-2n, 2n]^{|S_k|} \quad \text{for} \quad k = 1,..,K\right\},$$

where the function $\phi$ is defined as

$$\phi(\mathbf{x}|S_k, \mathbf{b}_{S_k,k}, \Gamma_{S_k,k}, \mathbf{c}_{S_k,k}) = \prod_{j\in S_k}\left(1 - \sigma\left(\frac{x_j - b_{j,k}}{\gamma_{j,k}}\right) + c_{j,k}\sigma\left(\frac{x_j - b_{j,k}}{\gamma_{j,k}}\right)\right).$$

and the vector $\mathbf{c}_{S_k,k}$ is defined as $\mathbf{c}_{S_k,k} = (c_{j,k}, j \in S_k)$.

For all $j, k$, we have

$$\int_0^1 \sigma\left(\frac{x - b_{j,k}}{\gamma_{j,k}}\right)dx \ge \int_{b_{j,k}}^1 \sigma\left(\frac{x - b_{j,k}}{\gamma_{j,k}}\right)dx$$
$$\ge C_{\sigma,j,k},$$

where $C_{\sigma,j,k}$ is a positive constant and thus, we have $|c_j(b_{j,k}, \gamma_{j,k})| \le C_\sigma$, $\forall j, k$ for some positive constant $C_\sigma$. Hence, for all $K \in [K_{\max}]$,

$$\mathcal{F}^n(K) \subseteq \mathcal{G}^n(K), \tag{21}$$

whenever $n$ is sufficiently large. Therefore, it suffices to verify Condition (17) over

$$\mathcal{G}^n = \bigcup_{K=1}^{M_n}\mathcal{G}^n(K). \tag{22}$$

**Lemma N.2.** *For any integer $K$, we have*

$$\mathcal{N}(\epsilon, \mathcal{G}^n(K), \|\cdot\|_\infty) \le \left(1 + \frac{K2^{p+4}n^{3p+1}}{\epsilon}\right)^{K(1+3p)}.$$

Proof.)

First, since the maximum dimension of parameters in $\mathcal{G}^n(K)$ is $K(1+3p)$, we consider $K(1+3p)$-dimensional hypercube $[-2n, 2n]^{K(1+3p)}$. Then, we have

$$\mathcal{N}(\epsilon_1, [-2n, 2n]^{K(1+3p)}, \|\cdot\|_1) \le \left(\mathcal{N}(\epsilon_1, [-2n, 2n], \|\cdot\|_1)\right)^{K(1+3p)}$$

$$\le \left(1 + \frac{4n}{\epsilon_1}\right)^{K(1+3p)}.$$

For $\mathbf{S}_K = (S_k, k \in [K])$, we define $\mathfrak{S} := (\mathcal{B}_K, \mathbf{b}_{\mathbf{S}_K,K}, \Gamma_{\mathbf{S}_K,K}, \mathbf{c}_{\mathbf{S}_K,K})$, where

$$\mathcal{B}_K = (\beta_1, ..., \beta_K),$$
$$\mathbf{b}_{\mathbf{S}_K,K} = (\mathbf{b}_{S_k,k}, k \in [K]),$$
$$\Gamma_{\mathbf{S}_K,K} = (\Gamma_{S_k,k}, k \in [K]),$$
$$\mathbf{c}_{\mathbf{S}_K,K} = (\mathbf{c}_{S_k,k}, k \in [K]).$$

Let $\left\{\mathfrak{S}^1, ..., \mathfrak{S}^{\mathcal{N}(\epsilon_1, [-n,n]^{K(1+3p)}, \|\cdot\|_1)}\right\}$ be an $\epsilon_1$-cover of $[-2n, 2n]^{K(1+3p)}$, and for given $\mathfrak{S} \in [-2n, 2n]^{K(1+3p)}$, let $\tilde{\mathfrak{S}}$ be an element in the $\epsilon_1$-cover such that $\|\mathfrak{S} - \tilde{\mathfrak{S}}\|_1 \le \epsilon_1$.

Note that for any $f_\Theta \in \mathcal{G}^n(K)$, we have

$$f_{\mathfrak{S}}(\mathbf{x}) = \sum_{k=1}^K \beta_k \prod_{j \in S_k} \phi(x_j | \{j\}, b_{j,k}, \gamma_{j,k}, c_{j,k}),$$

where

$$\phi(x_j | \{j\}, b_{j,k}, \gamma_{j,k}, c_{j,k}) = 1 - \sigma\left(\frac{x_j - b_{j,k}}{\gamma_{j,k}}\right) + c_{j,k}\sigma\left(\frac{x_j - b_{j,k}}{\gamma_{j,k}}\right)$$

with $|c_{j,k}| \le 2n$. Then, for any $f_{\mathfrak{S}} \in \mathcal{G}^n(K)$, we have

$$\sup_{\mathbf{x}} \left| f_{\mathfrak{S}}(\mathbf{x}) - f_{\tilde{\mathfrak{S}}}(\mathbf{x}) \right|$$

$$\le \sup_{\mathbf{x}} \sum_{k=1}^K \left| \beta_k \prod_{j \in S_k} \phi(x_j | \{j\}, b_{j,k}, \gamma_{j,k}, c_{j,k}) - \tilde{\beta}_k \prod_{j \in S_k} \phi(x_j | \{j\}, \tilde{b}_{j,k}, \tilde{\gamma}_{j,k}, \tilde{c}_{j,k}) \right|$$

$$\le \sup_{\mathbf{x}} \sum_{k=1}^K \left( \left| \beta_k \prod_{j \in S_k} \phi(x_j | \{j\}, b_{j,k}, \gamma_{j,k}, c_{j,k}) - \tilde{\beta}_k \prod_{j \in S_k} \phi(x_j | \{j\}, b_{j,k}, \gamma_{j,k}, c_{j,k}) \right| \right. \tag{23}$$

$$\left. + \left| \tilde{\beta}_k \prod_{j \in S_k} \phi(x_j | \{j\}, b_{j,k}, \gamma_{j,k}, c_{j,k}) - \tilde{\beta}_k \prod_{j \in S_k} \phi(x_j | \{j\}, \tilde{b}_{j,k}, \tilde{\gamma}_{j,k}, \tilde{c}_{j,k}) \right| \right).$$

**Upper bound of first term in (23).** Since

$$\left| \prod_{j \in S_k} \phi(x_j | \{j\}, b_{j,k}, \gamma_{j,k}, c_{j,k}) \right| = \left| \prod_{j \in S_k} \left(1 - \sigma\left(\frac{x_j - b_{j,k}}{\gamma_{j,k}}\right) + c_{j,k}\sigma\left(\frac{x_j - b_{j,k}}{\gamma_{j,k}}\right)\right) \right|$$

$$\le \prod_{j \in S_k} \left( \left|1 - \sigma\left(\frac{x_j - b_{j,k}}{\gamma_{j,k}}\right) + c_{j,k}\sigma\left(\frac{x_j - b_{j,k}}{\gamma_{j,k}}\right)\right| \right)$$

$$\le \prod_{j \in S_k} (1 + 2n)$$

$$\le (1 + 2n)^p,$$

we have

$$\sup_{\mathbf{x}} \sum_{k=1}^{K} \left| \beta_k \prod_{j \in S_k} \phi(x_j | \{j\}, b_{j,k}, \gamma_{j,k}, c_{j,k}) - \tilde{\beta}_k \prod_{j \in S_k} \phi(x_j | \{j\}, b_{j,k}, \gamma_{j,k}, c_{j,k}) \right|$$

$$\leq \sup_{\mathbf{x}} \sum_{k=1}^{K} (1 + 2n)^{|S_k|} |\beta_k - \tilde{\beta}_k|$$

$$\leq (1 + 2n)^p \epsilon_1.$$

**Upper bound of second term in (23).** Using direct calculation and triangle inequality, we have

$$\left| \tilde{\beta}_k \prod_{j \in S_k} \left( \phi(x_j | \{j\}, b_{j,k}, \gamma_{j,k}, c_{j,k}) - \phi(x_j | \{j\}, \tilde{b}_{j,k}, \tilde{\gamma}_{j,k}, \tilde{c}_{j,k}) \right) \right|$$

$$= \left| \tilde{\beta}_k \prod_{j \in S_k} \left( \sigma\left( \frac{x_j - \tilde{b}_{j,k}}{\tilde{\gamma}_{j,k}} \right) - \sigma\left( \frac{x_j - b_{j,k}}{\gamma_{j,k}} \right) + c_{j,k}\sigma\left( \frac{x_j - b_{j,k}}{\gamma_{j,k}} \right) - \tilde{c}_{j,k}\sigma\left( \frac{x_j - \tilde{b}_{j,k}}{\tilde{\gamma}_{j,k}} \right) \right) \right|$$

$$= |\tilde{\beta}_k| \prod_{j \in S_k} \left| \sigma\left( \frac{x_j - \tilde{b}_{j,k}}{\tilde{\gamma}_{j,k}} \right) - \sigma\left( \frac{x_j - b_{j,k}}{\gamma_{j,k}} \right) + c_{j,k}\sigma\left( \frac{x_j - b_{j,k}}{\gamma_{j,k}} \right) - \tilde{c}_{j,k}\sigma\left( \frac{x_j - \tilde{b}_{j,k}}{\tilde{\gamma}_{j,k}} \right) \right|$$

$$\leq n \prod_{j \in S_k} \left( \left| \sigma\left( \frac{x_j - \tilde{b}_{j,k}}{\tilde{\gamma}_{j,k}} \right) - \sigma\left( \frac{x_j - b_{j,k}}{\gamma_{j,k}} \right) \right| + \left| c_{j,k}\sigma\left( \frac{x_j - b_{j,k}}{\gamma_{j,k}} \right) - \tilde{c}_{j,k}\sigma\left( \frac{x_j - \tilde{b}_{j,k}}{\tilde{\gamma}_{j,k}} \right) \right| \right).$$

Since $\sigma(\cdot)$ is Lipschitz function, we have

$$\left| \sigma\left( \frac{x_j - \tilde{b}_{j,k}}{\tilde{\gamma}_{j,k}} \right) - \sigma\left( \frac{x_j - b_{j,k}}{\gamma_{j,k}} \right) \right|$$

$$\leq \left| \frac{x_j - \tilde{b}_{j,k}}{\tilde{\gamma}_{j,k}} - \frac{x_j - b_{j,k}}{\gamma_{j,k}} \right|$$

$$\leq \left( \left| \frac{x_j - \tilde{b}_{j,k}}{\tilde{\gamma}_{j,k}} - \frac{x_j - b_{j,k}}{\tilde{\gamma}_{j,k}} \right| + \left| \frac{x_j - b_{j,k}}{\tilde{\gamma}_{j,k}} - \frac{x_j - b_{j,k}}{\gamma_{j,k}} \right| \right)$$

$$\leq 2n^2 \left( |\tilde{b}_{j,k} - b_{j,k}| + |\tilde{\gamma}_{j,k} - \gamma_{j,k}| \right).$$

Similarly, we have

$$\left| c_{j,k}\sigma\left( \frac{x_j - b_{j,k}}{\gamma_{j,k}} \right) - \tilde{c}_{j,k}\sigma\left( \frac{x_j - \tilde{b}_{j,k}}{\tilde{\gamma}_{j,k}} \right) \right|$$

$$\leq \left| c_{j,k}\sigma\left( \frac{x_j - b_{j,k}}{\gamma_{j,k}} \right) - \tilde{c}_{j,k}\sigma\left( \frac{x_j - b_{j,k}}{\gamma_{j,k}} \right) \right| + \left| \tilde{c}_{j,k}\sigma\left( \frac{x_j - b_{j,k}}{\gamma_{j,k}} \right) - \tilde{c}_{j,k}\sigma\left( \frac{x_j - \tilde{b}_{j,k}}{\tilde{\gamma}_{j,k}} \right) \right|$$

$$\leq 4n^3 \left( |c_{j,k} - \tilde{c}_{j,k}| + |\tilde{b}_{j,k} - b_{j,k}| + |\tilde{\gamma}_{j,k} - \gamma_{j,k}| \right).$$

To sum up, the upper bound of (23) is

$$\sup_{\mathbf{x}} |f_{\mathfrak{G}}(\mathbf{x}) - f_{\tilde{\mathfrak{G}}}(\mathbf{x})| \leq K\left( (1 + 2n)^p \epsilon_1 + 2^{p+3} n^{3p+1} \epsilon_1^p \right)$$

$$\leq K(2n)^{3p+1} \epsilon_1.$$

Let $\epsilon = K(2n)^{3p+1} \epsilon_1$. Then, we conclude that

$$\mathcal{N}(\epsilon, \mathcal{G}^n(K), \|\cdot\|_\infty) \leq \left( 1 + \frac{2K(2n)^{3p+2}}{\epsilon} \right)^{K(1+3p)}.$$

$\square$

Using Lemma N.2, we have

$$\mathcal{N}(\delta, \mathcal{F}^n, \|\cdot\|_\infty) \le \sum_{K=1}^{M_n} \left(1 + \frac{2K(2n)^{3p+2}}{\delta}\right)^{K(1+3p)}$$

$$\le M_n \left(1 + \frac{2M_n(2n)^{3p+2}}{\delta}\right)^{M_n(1+3p)}.$$

Let $\delta = \varepsilon/8$. Finally, we choose $C_3$ such that

$$\log \mathcal{N}(\delta, \mathcal{F}^n, \|\cdot\|_\infty) \le \log M_n + M_n(1+3p)\log\left(1 + \frac{2M_n(2n)^{3p+2}}{\delta}\right)$$

$$< n\varepsilon^2/10.$$

Condition (17) is satisfied by letting $C_1 = \varepsilon^2/10$. $\qquad\square$

## N.4 VERIFYING CONDITION (18)

For $S \subseteq [p]$, using Theorem 3.3 in Park et al. (2025), there exist TPNNs such that

$$\left\| f_{0,S} - f_{k_S, \hat{\mathcal{B}}_{S,k_{n,S}}, \hat{\mathbf{b}}_{S,k_{n,S}}, \hat{\Gamma}_{S,k_{n,S}}} \right\|_\infty \le \frac{C_S}{k_{n,S}^{1/|S|} + 1} \tag{24}$$

for some positive constant $C_S$. Here, $\hat{\beta}_{S,k}s$ are uniformly bounded, i.e., $|\hat{\beta}_{S,k}| \le c_S$ for some positive constant $c_S$ and $\hat{\gamma}_{j,k} = 1/k_{n,S}^3$ for all $j, k$ as specified in Theorem 3.3 of Park et al. (2025).

Let $k_{n,S}$ such that

$$\frac{C_S}{k_{n,S}^{1/|S|} + 1} \le \varepsilon\sqrt{2}/(\sqrt{C_A} \cdot 3 \cdot 2^p). \tag{25}$$

Let $k_n = \sum_{S \subseteq [p]} k_{n,S}$ and $f_{k_n, \hat{\mathcal{B}}_{k_n}, \hat{\mathbf{b}}_{k_n}, \hat{\Gamma}_{k_n}} = \sum_{S \subseteq [p]} f_{k_{n,S}, \hat{\mathcal{B}}_{S,k_{n,S}}, \hat{\mathbf{b}}_{S,k_{n,S}}, \hat{\Gamma}_{S,k_{n,S}}}$. For notational simplicity, we write $\hat{\mathcal{B}}_{k_n}, \hat{\mathbf{b}}_{k_n}$ and $\hat{\Gamma}_{k_n}$ simply as $\hat{\mathcal{B}}, \hat{\mathbf{b}}$ and $\hat{\Gamma}$, respectively. Since

$$\|f_0 - f_{k_n, \mathcal{B}, \mathbf{b}, \Gamma}\|_\infty$$
$$\le \|f_0 - f_{k_n, \hat{\mathcal{B}}, \hat{\mathbf{b}}, \hat{\Gamma}}\|_\infty + \|f_{k_n, \hat{\mathcal{B}}, \hat{\mathbf{b}}, \hat{\Gamma}} - f_{k_n, \mathcal{B}, \hat{\mathbf{b}}, \hat{\Gamma}}\|_\infty + \|f_{k_n, \mathcal{B}, \hat{\mathbf{b}}, \hat{\Gamma}} - f_{k_n, \mathcal{B}, \mathbf{b}, \Gamma}\|_\infty, \tag{26}$$

we have

$$\pi\left(f \in \mathcal{F} : \|f - f_0\|_\infty \le \frac{\varepsilon}{3}\sqrt{\frac{2}{C_A}}\right)$$
$$\ge \pi(K = k_n)\left(\prod_{S' \subseteq [p]} \pi(S = S')\right) \tag{27}$$
$$\times \pi\left(\left\{\|f_{k_n, \hat{\mathcal{B}}, \hat{\mathbf{b}}, \hat{\Gamma}} - f_{k_n, \mathcal{B}, \hat{\mathbf{b}}, \hat{\Gamma}}\|_\infty \le \frac{\varepsilon}{3}\sqrt{\frac{2}{C_A}}\right\} \bigcap \left\{\|f_{k_n, \mathcal{B}, \hat{\mathbf{b}}, \hat{\Gamma}} - f_{k_n, \mathcal{B}, \mathbf{b}, \Gamma}\|_\infty \le \frac{\varepsilon}{3}\sqrt{\frac{2}{C_A}}\right\}\right). \tag{28}$$

Therefore, it remains to derive the lower bounds for (27) and (28).

**Lower bound of (27).** We have

$$\pi(K = k_n)\left(\prod_{S' \subseteq [p]} \pi(S = S')\right) = \left(\prod_{S' \subseteq [p]} \pi(S = S')\right) \frac{\exp(-C_0 k_n \log n)}{\sum_{k=0}^{K_{\max}} \exp(-C_0 k \log n)}$$
$$> \exp(-\mathfrak{d}_1 n)$$

for some positive constant $\mathfrak{d}_1$.

**Lower bound of (28).**   For any $\mathcal{B} = (\beta_k, k \in [k_n]) \in \mathbb{R}^k$, we have

$$
\begin{aligned}
\|f_{k_n,\mathcal{B},\hat{\mathbf{b}},\hat{\Gamma}} - f_{k_n,\hat{\mathcal{B}},\hat{\mathbf{b}},\hat{\Gamma}}\|_\infty &\leq \sup_{\mathbf{x}} \sum_{k=1}^{k_n} \left| \beta_k \prod_{j \in S_k} \phi(x_j|\{j\}, \hat{b}_{j,k}, \hat{\gamma}_{j,k}) - \hat{\beta}_k \prod_{j \in S_k} \phi(x_j|\{j\}, \hat{b}_{j,k}, \hat{\gamma}_{j,k}) \right| \\
&\leq \sup_{\mathbf{x}} \sum_{k=1}^{k_n} \left| (\beta_k - \hat{\beta}_k) \prod_{j \in S_k} \phi(x_j|\{j\}, \hat{b}_{j,k}, \hat{\gamma}_{j,k}) \right| \\
&\leq \sum_{k=1}^{k_n} \left| (\beta_k - \hat{\beta}_k)(1 + C_\sigma)^p \right| \\
&\leq (1 + C_\sigma)^p \|\mathcal{B} - \hat{\mathcal{B}}\|_1 \\
&\leq (1 + C_\sigma)^p \sqrt{k_n} \|\mathcal{B} - \hat{\mathcal{B}}\|_2.
\end{aligned}
\tag{29}
$$

That is, we have

$$
\left\{ \|f_{k_n,\hat{\mathcal{B}},\hat{\mathbf{b}},\hat{\Gamma}} - f_{k_n,\mathcal{B},\hat{\mathbf{b}},\hat{\Gamma}}\|_\infty \leq \frac{\varepsilon}{3} \sqrt{\frac{2}{C_A}} \right\} \supseteq \left\{ \|\mathcal{B} - \hat{\mathcal{B}}\|_2 \leq ((1 + C_\sigma)^p \sqrt{k_n})^{-1} \frac{\varepsilon}{3} \sqrt{\frac{2}{C_A}} \right\}.
$$

Furthermore, direct calculation yields

$$
\begin{aligned}
\|f_{k_n,\mathcal{B},\hat{\mathbf{b}},\hat{\Gamma}} - f_{k_n,\mathcal{B},\mathbf{b},\Gamma}\|_\infty &= \sup_{\mathbf{x}} \sum_{k=1}^{k_n} |\beta_k| \left| \prod_{j \in S_k} \left( \phi(x_j|\{j\}, \hat{b}_{j,k}, \hat{\gamma}_{j,k}) - \phi(x_j|\{j\}, b_{j,k}, \gamma_{j,k}) \right) \right| \\
&\leq (1 + C_\sigma) \sup_{\mathbf{x}} \sum_{k=1}^{k_n} |\beta_k| \left| \prod_{j \in S_k} \left( \frac{x_j - \hat{b}_{j,k}}{\hat{\gamma}_{j,k}} - \frac{x_j - b_{j,k}}{\gamma_{j,k}} \right) \right| \\
&= (1 + C_\sigma) \sup_{\mathbf{x}} \sum_{k=1}^{k_n} |\beta_k| \left| \prod_{j \in S_k} \left( \frac{b_{j,k} - \hat{b}_{j,k}}{\hat{\gamma}_{j,k}} + (x_j - b_{j,k}) \frac{\gamma_{j,k} - \hat{\gamma}_{j,k}}{\gamma_{j,k} \hat{\gamma}_{j,k}} \right) \right| \\
&\leq (1 + C_\sigma) \sup_{\mathbf{x}} \sum_{k=1}^{k_n} |\beta_k| \prod_{j \in S_k} \left( \left| \frac{b_{j,k} - \hat{b}_{j,k}}{\hat{\gamma}_{j,k}} \right| + 2 \left| \frac{\gamma_{j,k} - \hat{\gamma}_{j,k}}{\gamma_{j,k} \hat{\gamma}_{j,k}} \right| \right).
\end{aligned}
$$

Let $C_{n,j,k} = \frac{|\hat{\gamma}_{j,k}|}{2} \left( \frac{\varepsilon}{3\xi(1+C_\sigma)k_n} \sqrt{\frac{2}{C_A}} \right)^{1/|S_k|}$. If $|\gamma_{j,k} - \hat{\gamma}_{j,k}| \leq \epsilon_1$, we have

$$
\left| \frac{\gamma_{j,k} - \hat{\gamma}_{j,k}}{\hat{\gamma}_{j,k} \gamma_{j,k}} \right| \leq \frac{\epsilon_1}{\hat{\gamma}_{j,k}(\hat{\gamma}_{j,k} - \epsilon_1)} \leq \frac{1}{4} \left( \frac{\varepsilon}{3\xi k_n} \sqrt{\frac{2}{C_A}} \right)^{1/|S_k|},
$$

where $\epsilon_1 = \frac{C_{n,j,k}|\hat{\gamma}_{j,k}|}{2 + C_{n,j,k}}$. Therefore, if

$$
\begin{aligned}
|\beta_k| &\leq \xi, \\
|b_{j,k} - \hat{b}_{j,k}| &\leq 2C_{n,j,k}, \\
|\gamma_{j,k} - \hat{\gamma}_{j,k}| &\leq \frac{C_{n,j,k_n}|\hat{\gamma}_{j,k}|}{2 + C_{n,j,k}}
\end{aligned}
$$

hold, we have

$$
\|f_{k_n,\mathcal{B},\hat{\mathbf{b}},\hat{\Gamma}} - f_{k_n,\mathcal{B},\mathbf{b},\Gamma}\|_\infty \leq \frac{\varepsilon}{3} \sqrt{\frac{2}{C_A}}.
\tag{30}
$$

That is, we have

$$\left\{\|f_{k_n,\hat{\mathcal{B}},\hat{\mathbf{b}},\hat{\Gamma}} - f_{k_n,\mathcal{B},\hat{\mathbf{b}},\hat{\Gamma}}\|_\infty \le \frac{\varepsilon}{3}\sqrt{\frac{2}{C_A}}\right\} \bigcap \left\{\|f_{k_n,\mathcal{B},\hat{\mathbf{b}},\hat{\Gamma}} - f_{k_n,\mathcal{B},\mathbf{b},\Gamma}\|_\infty \le \frac{\varepsilon}{3}\sqrt{\frac{2}{C_A}}\right\}$$

$$\supseteq \left\{\|\mathcal{B} - \hat{\mathcal{B}}\|_2 \le ((1+C_\sigma)^p\sqrt{k_n})^{-1}\frac{\varepsilon}{3}\sqrt{\frac{2}{C_A}},\right.$$

$$|\beta_j| \le \xi,$$

$$|b_{j,k} - \hat{b}_{j,k}| \le 2C_{n,j,k},$$

$$\left.|\gamma_{j,k} - \hat{\gamma}_{j,k}| \le \frac{C_{n,j,k}|\hat{\gamma}_{j,k}|}{2 + C_{n,j,k}}, \quad \forall j \in S_k, \forall k \in [k_n]\right\}.$$

It implies that

$$\pi\left(\left\{\|f_{k_n,\hat{\mathcal{B}},\hat{\mathbf{b}},\hat{\Gamma}} - f_{k_n,\mathcal{B},\hat{\mathbf{b}},\hat{\Gamma}}\|_\infty \le \frac{\varepsilon}{3}\sqrt{\frac{2}{C_A}}\right\} \bigcap \left\{\|f_{k_n,\mathcal{B},\hat{\mathbf{b}},\hat{\Gamma}} - f_{k_n,\mathcal{B},\mathbf{b},\Gamma}\|_\infty \le \frac{\varepsilon}{3}\sqrt{\frac{2}{C_A}}\right\}\right)$$

$$\ge \pi(\|\mathcal{B} - \hat{\mathcal{B}}\|_2 \le ((1+C_\sigma)^p\sqrt{k_n})^{-1}\varepsilon\sqrt{2}/(3\sqrt{C_A}), |\beta_k| \le \xi, \ \forall k \in [k_n]) \qquad (31)$$

$$\times \pi(|b_{j,k} - \hat{b}_{j,k}| \le 2C_{n,j,k}, \ \forall j \in S_k, \ \forall k \in [k_n]) \qquad (32)$$

$$\times \pi\left(|\gamma_{j,k} - \hat{\gamma}_{j,k}| \le \frac{C_{n,j,k}}{1 + C_{n,j,k}}|\hat{\gamma}_{j,k}|, \ \forall j \in S_k, \ \forall k \in [k_n]\right). \qquad (33)$$

Now, we will show that these three probabilities sufficiently large.

**Lower bound of (31).** Since

$$\left\{\|\mathcal{B} - \hat{\mathcal{B}}\|_2 \le ((1+C_\sigma)^p\sqrt{k_n})^{-1}\varepsilon\sqrt{2}/(3\sqrt{C_A}), |\beta_k| \le \xi, \ \forall k \in [k_n]\right\}$$

$$\supseteq \left\{|\beta_k - \hat{\beta}_k| \le ((1+C_\sigma)^p k_n)^{-1}\varepsilon\sqrt{2}/(3\sqrt{C_A}), |\beta_k| \le \xi, \forall k \in [k_n]\right\}$$

$$\supseteq \left\{|\beta_k - \hat{\beta}_k| \le ((1+C_\sigma)^p k_n)^{-1}\varepsilon\sqrt{2}/(3\sqrt{C_A}), \forall k \in [k_n]\right\} \qquad (34)$$

for sufficiently large $n$, it suffices to get the lower bound of $\pi(|\beta_k - \hat{\beta}_k| \le ((1 + C_\sigma)^p k_n)^{-1}\varepsilon\sqrt{2}/(3\sqrt{C_A}))$ for $k \in [k_n]$.

For $k \in [k_n]$, we let

$$I_k = [\hat{\beta}_k \pm ((1+C_\sigma)^p k_n)^{-1}\varepsilon\sqrt{2}/(3\sqrt{C_A})]$$

and we have

$$\pi(|\beta_k - \hat{\beta}_k| \le ((1+C_\sigma)^p k_n)^{-1}\varepsilon\sqrt{2}/(3\sqrt{C_A}))$$

$$= \int_{I_k} \frac{1}{\sqrt{2\pi}\sigma_\beta}\exp\left(-\frac{\beta_k^2}{2\sigma_\beta^2}\right)d\beta_k$$

$$\ge |I_k|\frac{1}{\sqrt{2\pi}\sigma_\beta}\exp\left(-\frac{(\max_S c_S + ((1+C_\sigma)^p k_n)^{-1}\varepsilon\sqrt{2}/(3\sqrt{C_A}))^2}{2\sigma_\beta^2}\right) \qquad (35)$$

$$> \exp(-\mathfrak{d}_1 n)$$

for some positive constant $\mathfrak{d}_1$, where (35) is derived from $|\hat{\beta}_k| \le \max_S c_S$.

**Lower bound of (32).** Since

$$\pi\big(|b_{j,k} - \hat{b}_{j,k}| \le 2C_{n,j,k}\big) = 4C_{n,j,k}$$

for all $j \in S_k, \ k \in [k_n]$, we have

$$\pi\big(|b_{j,k} - \hat{b}_{j,k}| \le 2C_{n,j,k}, \ \forall j \in S_k, \ \forall k \in [k_n]\big) = \prod_{k \in [k_n], j \in S_k} 4C_{n,j,k}$$

$$> \exp(-\mathfrak{d}_2 n)$$

for some positive constant $\mathfrak{d}_2$.

**Lower bound of (33).** Using direct calculation, we have

$$\pi\left(|\gamma_{j,k} - \hat{\gamma}_{j,k}| \le \frac{C_{n,j,k}}{2 + C_{n,j,k}}\hat{\gamma}_{j,k}\right) \ge \left(\frac{2C_{n,j,k}\hat{\gamma}_{j,k}}{2 + C_{n,j,k}}\right) \min_{x \in [L_n, U_n]} pdf_\gamma(x)$$

$$= \left(\frac{2C_{n,j,k}\hat{\gamma}_{j,k}}{2 + C_{n,j,k}}\right) \frac{b_\gamma^{a_\gamma}}{\Gamma(a_\gamma)} \min_{x \in [L_n, U_n]} x^{a_\gamma - 1} \exp(-b_\gamma x),$$

where $L_n = \hat{\gamma}_{j,k} - \frac{C_{n,j,k}\hat{\gamma}_{j,k}}{2 + C_{n,j,k}}$ and $U_n = \hat{\gamma}_{j,k} + \frac{C_{n,j,k}\hat{\gamma}_{j,k}}{2 + C_{n,j,k}}$.

Note that $1/k_n^3 \le \hat{\gamma}_{i,j} \le 1$. For $a_\gamma > 1$, we have

$$\min_{x \in [L_n, U_n]} x^{a_\gamma - 1} \ge L_n^{a_\gamma - 1}$$

$$= \left(\frac{2\hat{\gamma}_{j,k}}{2 + C_{n,j,k}}\right)^{a_r - 1}$$

$$> \exp(-\mathfrak{d}_3 n)$$

for some positive constant $\mathfrak{d}_3$ and for $a_\gamma < 1$, we have

$$\min_{x \in [L_n, U_n]} x^{a_\gamma - 1} \ge U_n^{a_\gamma - 1}$$

$$= \left(\hat{\gamma}_{j,k}\right)^{1 - a_\gamma}$$

$$> \exp(-\mathfrak{d}_4 n)$$

for some positive constant $\mathfrak{d}_4$. Furthermore, we have

$$\min_{x \in [L_n, U_n]} \exp(-b_\gamma x) \ge \exp(-b_\gamma U_n)$$

$$\ge \exp(-2b_\gamma \hat{\gamma}_{i,j})$$

$$> \exp(-2\mathfrak{d}_5 n)$$

and

$$\frac{2C_{n,j,k}\hat{\gamma}_{j,k}}{2 + C_{n,j,k}} > \exp(-2\mathfrak{d}_7 n)$$

for some positive constants $\mathfrak{d}_6$ and $\mathfrak{d}_7$. Finally, the proof is completed by letting $C_2 = \sum_{i=1}^7 \mathfrak{d}_i$.

$\square$

## N.5 VERIFYING CONDITION (19)

We will verify Condition (19) with the constant $C_3$.

We let

$$Z_1 = \{K > M_n\},$$
$$Z_2 = \{\{K \le M_n\} \cap \{\exists k \in [K] \text{ such that } |\beta_k| > n\}\},$$
$$Z_3 = \{\{K \le M_n\} \cap \{\exists k \in [K] \text{ such that } \Gamma_{S_k, k} \in (n, \infty)^{|S_k|}\}\}.$$

Since

$$\pi(\mathcal{F} \backslash \mathcal{F}^n) = \pi(Z_1 \cup Z_2 \cup Z_3),$$

the upper bound of $\pi(\mathcal{F} \backslash \mathcal{F}^n)$ is

$$\pi(\mathcal{F} \backslash \mathcal{F}^n)$$
$$\le \pi(K > M_n) \tag{36}$$
$$+ \pi(K \le M_n)\pi(\exists k \in [K] \text{ such that } |\beta_k| > n | K \le M_n) \tag{37}$$
$$+ \pi(K \le M_n)\pi(\exists k \in [K] \text{ such that } \Gamma_{S_k, k} \in (n, \infty)^{|S_k|} | K \le M_n). \tag{38}$$

**Upper bound of (36).** For $M_n = \lfloor \frac{C_3 n \varepsilon^2}{\log n} \rfloor$, we have

$$\pi(K > M_n) = \frac{\sum_{k=M_n+1}^{K_{\max}} \exp(-kC_0 \log n)}{\sum_{k=0}^{K_{\max}} \exp(-kC_0 \log n)}$$
$$\leq \exp(-M_n C_0 \log n).$$

Since $C_3 > \frac{C_2+2}{C_0 \log n}$ for sufficiently large $n$, we have

$$\pi(K > M_n) \exp((C_2 + 2)n) \to 0 \ \text{as} \ n \to \infty.$$

**Upper bound of (37).** We have

$$\pi(\exists k \in [K] \text{ such that } |\beta_k| > n | K \leq M_n) \leq M_n \pi(|\beta_1| > n)$$
$$\leq 2M_n \exp\left(-\frac{n^2}{2\sigma_\beta^2}\right),$$

where $\sigma_\beta^2$ is a constant. That is, we conclude that

$$\pi(\exists k \in [K] \text{ such that } |\beta_k| > n | K \leq M_n) \exp((C_2 + 2)n) \to 0 \ \text{as} \ n \to \infty.$$

**Upper bound of (36).** For any $j, k$, using Markov inequality, we have

$$\pi(\gamma_{j,k} > n) \leq \mathbb{E}\left[\exp\left(\frac{b_\gamma \gamma_{j,k}}{2}\right)\right] \exp\left(-\frac{b_\gamma n}{2}\right)$$
$$= \left(\frac{1}{2}\right)^{-a_\gamma} \exp\left(-\frac{b_\gamma n}{2}\right).$$

Since

$$\pi(\exists k \in [K] \text{ such that } \Gamma_{S_k,k} \in (n, \infty)^{|S_k|} | K \leq M_n) \leq M_n \pi(\gamma_{1,1} > n),$$

we have

$$\pi(\exists k \in [K] \text{ such that } \Gamma_{S_k,k} \in (n, \infty)^{|S_k|} | K \leq M_n) \exp((C_2 + 2)n) \to 0 \quad \text{as} \quad n \to \infty,$$

where $a_\gamma$ and $b_\gamma$ are positive constants.

$\square$

## N.6 VERIFICATION OF THE CONDITIONS IN GHOSAL ET AL. (1999)

For given data $\mathbf{x}^{(n)}$, let $q_{f,i}$ be the probability density of $\mathbb{Q}_{f(\mathbf{x}_i)}$ for $i = 1, ..., n$. From Theorem 2 of Ghosal et al. (1999), it suffices to verify that for every $f_0 \in \text{Lip}_{0,L,F}$, there exists a sieve $\mathcal{F}_\xi^n$, constants $\delta < \varepsilon/4, C_5, C_6 > 0$ and $C_1 < \varepsilon^2/8$ such that the following three conditions hold with respect to the $\|\cdot\|_{2,n}$.

$$\log \mathcal{N}\left(\delta, \mathcal{F}_\xi^n, \|\cdot\|_{2,n}\right) < nC_1, \tag{39}$$

$$\pi_\xi\left(f \in \mathcal{F}_\xi : \frac{1}{n}\sum_{i=1}^n K(q_{f_0,i}, q_{f,i}) \leq \varepsilon^2\right) > \exp(-nC_5), \tag{40}$$

$$\pi_\xi\left(\mathcal{F}_\xi \backslash \mathcal{F}_\xi^n\right) < \exp(-nC_6). \tag{41}$$

To complete the proof of Theorem N.1, we will verify that the three conditions (39), (40), and (41) for given data $\mathbf{x}^{(n)}$.

**Verifying Condition (39).**

Condition (39) holds under Condition (17).

**Verifying Condition (40).**

By using a direct calculation, for $i = 1, ..., n$, we have

$$K(q_{f_0,i}, q_{f,i}) = \int \left( (f_0(\mathbf{x}_i) - f(\mathbf{x}_i))y - A(f_0(\mathbf{x}_i)) + A(f(\mathbf{x}_i)) \right) q_{f_0,i}(y) dy \qquad (42)$$

$$= \left( (f_0(\mathbf{x}_i) - f(\mathbf{x}_i))\mathbb{E}[Y_i] - A(f_0(\mathbf{x}_i)) + A(f(\mathbf{x}_i)) \right) \qquad (43)$$

$$= \left( (f_0(\mathbf{x}_i) - f(\mathbf{x}_i))\dot{A}(f_0(\mathbf{x}_i)) - A(f_0(\mathbf{x}_i)) + A(f(\mathbf{x}_i)) \right). \qquad (44)$$

Using Talyor expansion, we have

$$K(q_{f_0,i}, q_{f,i}) = \frac{1}{2}\ddot{A}(\tilde{x})(f_0(\mathbf{x}_i) - f(\mathbf{x}_i))^2,$$

where $\tilde{x} \in [-F, F]$. That is, we have

$$\frac{1}{n} \sum_{i=1}^{n} K(q_{f_0,i}, q_{f,i}) \leq \frac{C_A}{2} \|f_0 - f\|_{2,n}^2.$$

When $\xi \geq 2^P F + \varepsilon\sqrt{\frac{2}{C_A}}$, we have

$$\pi_\xi \left( f \in \mathcal{F}_\xi : \|f - f_0\|_{2,n} \leq \varepsilon\sqrt{\frac{2}{C_A}} \right) \geq \pi \left( f \in \mathcal{F}_\xi : \|f - f_0\|_{2,n} \leq \varepsilon\sqrt{\frac{2}{C_A}} \right).$$

Therefore, the proof is done by Condition (18).

$\square$

**Verifying Condition (41).**

Since

$$\pi_\xi(\mathcal{F}\backslash\mathcal{F}^n) \leq \frac{\pi(\mathcal{F}\backslash\mathcal{F}^n)}{\pi(\|f\|_\infty \leq \xi)}$$

$$\leq \frac{\pi(\mathcal{F}\backslash\mathcal{F}^n)}{\pi\left( \|f - f_0\|_\infty \leq \varepsilon\sqrt{\frac{2}{C_A}} \right)}$$

$$\leq \exp(-(C_5 + 2)n)$$

for $2^p F + \varepsilon\sqrt{\frac{2}{C_A}} \leq \xi$, the condition (41) holds for $C_6 = C_5 + 2$ by condition (18) and (19).

$\square$

## N.7   STEP $(P.2)$

Since $(P.1)$ holds for arbitary $\mathbf{x}^{(n)}$, we conclude that

$$\lim_{n \to \infty} \mathbb{E}_0^n[\pi_\xi(\|f - f_0\|_{2,n} > \varepsilon | \mathbf{X}^{(n)}, Y^{(n)})] = 0$$

for any $\varepsilon > 0$.

$\square$

## O    PROOF OF THEOREM 3.2

The proof consists of the following 4 steps.

**(STEP E.1)**
We first establish the rate at which the posterior concentrates under the population $\ell_2$ norm; specifically, we demonstrate that

$$\mathbb{E}_0^n\left[\pi_\xi\left(f \in \mathcal{F}_\xi^n : \|f - f_0\|_{2,\mathbb{P}_{\mathbf{X}}} > \varepsilon | \mathbf{X}^{(n)}, Y^{(n)}\right)\right] \to 0, \tag{45}$$

for any $\varepsilon > 0$.

**(STEP E.2)**
Based on (45), we establish that the following holds for any subset $S \subseteq [p]$.

$$\mathbb{E}_0^n\left[\pi_\xi\left(f \in \mathcal{F}_\xi^n : \|f_S - f_{0,S}\|_{2,\mathbb{P}_{\mathbf{X}}} > \varepsilon | \mathbf{X}^{(n)}, Y^{(n)}\right)\right] \to 0, \tag{46}$$

for any $\varepsilon > 0$.

**(STEP E.3)**
We reformulate (46) in terms of the empirical $\ell_2$ norm. Specifically, we demonstrate that

$$\mathbb{E}_0^n\left[\pi_\xi\left(f \in \mathcal{F}_\xi^n : \|f_S - f_{0,S}\|_{2,n} > \varepsilon | \mathbf{X}^{(n)}, Y^{(n)}\right)\right] \to 0, \tag{47}$$

for any $\varepsilon > 0$.

**(STEP E.4)**
The last step is to verify

$$\mathbb{E}_0^n\left[\pi_\xi(\mathcal{F}_\xi \backslash \mathcal{F}_\xi^n | \mathbf{X}^{(n)}, Y^{(n)})\right] \to 0 \tag{48}$$

as $n \to \infty$.

### O.1    VERIFYING (**STEP D.1**)

To verify (**STEP D.1**), we rely on the following lemma, whose proof is provided in Theorem 19.3 of Györfi et al. (2006).

**Lemma O.1** (Theorem 19.3 of Györfi et al. (2006)). *Let $\mathbf{X}, \mathbf{X}_1, \ldots, \mathbf{X}_n$ be independent and identically distributed random vectors with values in $\mathbb{R}^d$. Let $K_1, K_2 \geq 1$ be constants and let $\mathcal{G}$ be a class of functions $g : \mathbb{R}^d \to \mathbb{R}$ with the properties*

$$|g(\mathbf{x})| \leq K_1, \quad \mathbb{E}[g(\mathbf{X})^2] \leq K_2\mathbb{E}[g(\mathbf{X})]. \tag{49}$$

*Let $0 < \kappa < 1$ and $\zeta > 0$. Assume that*

$$\sqrt{n}\kappa\sqrt{1-\kappa}\sqrt{\zeta} \geq 288 \max\left\{2K_1, \sqrt{2K_2}\right\} \tag{50}$$

*and that, for all $\mathbf{x}_1, \ldots, \mathbf{x}_n \in \mathbb{R}^d$ and for all $t \geq \frac{\zeta}{8}$,*

$$\frac{\sqrt{n}\kappa(1-\kappa)t}{96\sqrt{2}\max\{K_1, 2K_2\}} \geq \int_{\frac{\kappa(1-\kappa)t}{16\max\{K_1, 2K_2\}}}^{\sqrt{t}} \sqrt{\log \mathcal{N}\left(u, \left\{g \in \mathcal{G} : \frac{1}{n}\sum_{i=1}^n g(\mathbf{x}_i)^2 \leq 16t\right\}, \|\cdot\|_{1,n}\right)} \, du. \tag{51}$$

*Then,*

$$\mathbb{P}_{\mathbf{X}}^n\left(\sup_{g \in \mathcal{G}} \frac{\left|\mathbb{E}[g(\mathbf{X})] - \frac{1}{n}\sum_{i=1}^n g(\mathbf{X}_i)\right|}{\zeta + \mathbb{E}[g(\mathbf{X})]} > \kappa\right) \leq 60\exp\left(-\frac{n\zeta\kappa^2(1-\kappa)}{C_g \max\{K_1^2, K_2\}}\right)$$

*for some positive constant $C_g$.*

Since $\mathcal{F}_\xi^n$ depends on the dataset $\mathbf{X}^{(n)}$, we will apply Lemma O.1 to the function class $\mathcal{G}_\xi^n$ defined as

$$\mathcal{G}_\xi^n = \bigcup_{K=1}^{M_n} \mathcal{G}_\xi^n(K),$$

where $\mathcal{G}_\xi^n(K) = \{f \in \mathcal{G}^n(K) : \|f\|_\infty \le \xi\}$. Here, $\mathcal{G}^n(K)$ is defined in (20).

Since

$$\mathcal{N}(\epsilon, \mathcal{G}_\xi^n, \|\cdot\|_\infty) \le \mathcal{N}(\epsilon, \mathcal{G}^n, \|\cdot\|_\infty)$$

$$\le M_n \left(1 + \frac{M_n 2^{p+3} n^{3p+1}}{\epsilon}\right)^{M_n(1+3p)},$$

we can easily verify that conditions (49), (50), and (51) hold for $K_1 = K_2 = 4\xi^2$, $\kappa = \frac{1}{4}$, $\zeta = \varepsilon^2$, and $\mathcal{G} = \{g : g = (f_0 - f)^2, f \in \mathcal{G}_\xi^n\}$. That is, we have

$$\mathbb{P}_\mathbf{X}^n \left(\sup_{f \in \mathcal{F}_\xi^n} \frac{\left| \|f - f_0\|_{2,\mathbb{P}_\mathbf{X}}^2 - \|f - f_0\|_{2,n}^2 \right|}{\varepsilon^2 + \|f - f_0\|_{2,\mathbb{P}_\mathbf{X}}^2} > \frac{1}{4}\right) \le 60 \exp\left(-\frac{n\varepsilon^2/8}{C_g \cdot 16\xi^4}\right).$$

We define $\mathcal{A}_n := \left\{\mathbf{X}^{(n)} : \sup_{f \in \mathcal{F}_\xi^n} \frac{\left| \|f - f_0\|_{2,\mathbb{P}_\mathbf{X}}^2 - \|f - f_0\|_{2,n}^2 \right|}{\varepsilon^2 + \|f - f_0\|_{2,\mathbb{P}_\mathbf{X}}^2} \le \frac{1}{4}\right\}$. Then, we have

$$\mathbb{E}_0^n \left[\pi_\xi\left(f \in \mathcal{F}_\xi^n : \|f - f_0\|_{2,\mathbb{P}_\mathbf{X}} > \varepsilon | \mathbf{X}^{(n)}, Y^{(n)}\right)\right]$$

$$\le \mathbb{E}_0^n \left[\pi_\xi\left(f \in \mathcal{F}_\xi^n : \|f - f_0\|_{2,\mathbb{P}_\mathbf{X}} > \varepsilon | \mathbf{X}^{(n)}, Y^{(n)}\right) \mathbb{I}(\mathbf{X}^{(n)} \in \mathcal{A}_n)\right] + \mathbb{P}_\mathbf{X}^n(\mathcal{A}_n^c)$$

$$\le \mathbb{E}_0^n \left[\pi_\xi\left(f \in \mathcal{F}_\xi^n : \|f - f_0\|_{2,n} > \varepsilon/\sqrt{2} | \mathbf{X}^{(n)}, Y^{(n)}\right)\right] + \mathbb{P}_\mathbf{X}^n(\mathcal{A}_n^c)$$

$$\to 0$$

as $n \to \infty$.

$\square$

## O.2 VERIFYING (**STEP D.2**)

For $f \in \mathcal{F}_\xi^n$, we have

$$f(\mathbf{x}) = \sum_{S \subseteq [p]} f_S(\mathbf{x}_S),$$

where $f_S$ satisfies the sum-to-zero condition with respect to the uniform distribution on $(0,1)$.

Consider positive constants $C_7$ and $C_8$ such that

$$C_7 \le \inf_{\mathbf{x} \in \mathcal{X}} p_\mathbf{X}(\mathbf{x}) \le \sup_{\mathbf{x} \in \mathcal{X}} p_\mathbf{X}(\mathbf{x}) \le C_8. \tag{52}$$

Therefore, using the inequality (52), for all $S \subseteq [p]$, we have

$$\|f_0 - f\|_{2,\mathbb{P}_\mathbf{X}} \ge \sqrt{C_7 \int_\mathcal{X} (f_0(\mathbf{x}) - f(\mathbf{x}))^2 d\mathbf{x}}$$

$$= \sqrt{C_7 \sum_{S \subseteq [p]} \int_{\mathcal{X}_S} (f_{0,S}(\mathbf{x}_S) - f_S(\mathbf{x}_S))^2 d\mathbf{x}_S} \tag{53}$$

$$\ge C_9 \|f_{P,S} - f_{0,S}\|_{2,\mathbb{P}_\mathbf{X}},$$

where (53) is derived from the sum-to-zero condition with respect to the uniform distribution on $(0,1)$ and $C_9 = \sqrt{C_7/C_8}$.

Hence, we conclude that

$$\mathbb{E}_0^n\Big[\pi_\xi\Big(f \in \mathcal{F}_\xi^n : \|f_S - f_{0,S}\|_{2,\mathbb{P}_\mathbf{X}} > \varepsilon|\mathbf{X}^{(n)}, Y^{(n)}\Big)\Big]$$
$$\leq \mathbb{E}_0^n\Big[\pi_\xi\Big(f \in \mathcal{F}_\xi^n : \|f - f_0\|_{2,\mathbb{P}_\mathbf{X}} > \varepsilon C_9|\mathbf{X}^{(n)}, Y^{(n)}\Big)\Big]$$
$$\to 0,$$

as $n \to \infty$.

$\square$

## O.3  VERIFYING (STEP D.3)

Following the same approach as in the proof of (**STEP D.1**), and applying Lemma O.1 to the function class $\mathcal{G} = \{g : g = (f_{0,S} - f_S)^2, f \in \mathcal{G}_\xi^n\}$, we have

$$\lim_{n\to\infty} \mathbb{E}_0^n\Big[\pi_\xi\Big(f \in \mathcal{F}_\xi^n : \|f_S - f_{0,S}\|_{2,n} > \varepsilon\Big|\mathbf{X}^{(n)}, Y^{(n)}\Big)\Big] = 0.$$

$\square$

## O.4  VERIFYING THE (STEP D.4)

Since

$$\frac{\pi_\xi(\mathcal{F}_\xi \backslash \mathcal{F}_\xi^n)}{\pi_\xi(\mathbb{B}_n)} \leq \exp(-2n)$$

for given data $\mathbf{x}^{(n)}$, using Lemma 1 in Ghosal & Van Der Vaart (2007), we conclude that

$$\lim_{n\to\infty} \mathbb{E}_0^n\Big[\pi_\xi(\mathcal{F}_\xi \backslash \mathcal{F}_\xi^n|\mathbf{X}^{(n)}, Y^{(n)})\Big|\mathbf{X}^{(n)} = \mathbf{x}^{(n)}\Big] = 0.$$

Since it holds for arbitrary $\mathbf{x}^{(n)}$, the proof is completed.

$\square$

