# OpenReview forum: "Bayesian Neural Networks for Functional ANOVA Model"
_ICLR.cc/2026/Conference — ICLR 2026 Poster_

### Official Review · Reviewer_hVKC · 2025-10-22

**Soundness:** 3
**Presentation:** 3
**Contribution:** 2
**Rating:** 4
**Confidence:** 4

**Summary:**

This paper investigates functional ANOVA decomposition, a technique for decomposing a high dimensional complex function into lower dimensional components to enhance model interpretability. Existing methods typically require pre specifying the maximum order of interaction in the low dimensional functions and exhaustively considering all feature combinations up to that order. In practice, this often limits analysis to pairwise interaction terms. To address this limitation, the authors propose Bayesian TPNN, a Bayesian framework that adaptively selects the most relevant features for inclusion in the low dimensional components without pre-specifying the maximum order. The method is evaluated across diverse domains, including real world tabular datasets, simulated data, and toy image datasets.

**Strengths:**

This paper is well written and easy to follow. Using a Bayesian approach to select features for functional ANOVA decomposition provides a theoretically principled framework that could be of significant interest to the interpretability community and potentially a broader audience.

**Weaknesses:**

My major concern is its evaluation and experiments. Most experiments are conducted on toy or tabular datasets, where the application of deep learning is arguably less compelling. Please refer to my detailed questions below.

**Questions:**

1. In Table 1, the authors compare the proposed Bayesian TPNN approach with existing methods in terms of prediction accuracy. However, the proposed method does not appear to outperform existing baselines significantly, particularly when taking the standard errors into account.

2. In Table 2, the authors evaluate uncertainty quantification against existing methods. It would strengthen the comparison to include deep ensembles, which are widely recognized as a standard baseline for uncertainty estimation.

3. Most experiments are conducted on simulated or tabular datasets, with the exception of Section 4.4, which uses CelebA HQ and Catdog datasets. However, in these cases, the approach relies on another CBM model to generate interpretable concepts. Could the authors apply Bayesian TPNN directly to these datasets and produce interpretations for the low dimensional functions?

Overall, it is challenging to identify practical scenarios where the proposed method would be preferred. I recommend extending the evaluation to additional domains, such as the genomics datasets used in Martens & Yau (2020), to better demonstrate applicability.

---

> ### Author Response · Authors · 2025-11-17
>
> Thank you very much for your valuable feedback and thoughtful questions.
>  We have carefully reviewed all of your insightful comments and have made every effort to address them thoroughly.
> For reference, we have newly uploaded the revised paper, and the revised parts are highlighted in red.
>
> >**Question 1.**
> In Table 1, the authors compare the proposed Bayesian-TPNN approach with existing methods in terms of prediction accuracy.
> However, the proposed method does not appear to outperform existing baselines significantly, particularly when taking the standard errors into account.
>
> **Response.**
> The functional ANOVA model is a white box model and so we do not expect that its prediction performance is better than black box ones.
> The motivation of Bayesian-TPNN is that existing learning algorithms for the functional ANOVA model have too much worse prediction performance for certain data such as *Madelon* and *Servo* datasets.
> We have found that there are higher-order interactions in such datasets and existing algorithms fail to detect such higher-order interactions.
> **The aim of Bayesian-TPNN is to learn a white box model whose prediction performances are competitive to black box models**.
>
> As shown in Table 1, for the *Madelon* and *Servo* datasets where higher-order interactions are important, Bayesian-TPNN achieves markedly superior predictive performance compared to other interpretable models that consider only up to second-order interactions (e.g., NAM, ANOVA-TPNN, and linear models), and performs comparably to XGB.
> In addition, Table 4 of the paper presents that this improvement in the prediction performance is attributed to Bayesian-TPNN’s ability to capture higher-order interactions.
>
> >**Question 2.**
> In Table 2, the authors evaluate uncertainty quantification against existing methods.
> It would strengthen the comparison to include deep ensembles, which are widely recognized as a standard baseline for uncertainty estimation.
>
> **Response.**
> As the reviewer suggested, we conducted additional experiments on Deep Ensemble ([1]).
> In Deep Ensemble, the hyperparameters include the number of Multilayer Perceptrons (MLPs) for the ensemble, the dimension of hidden layer for each MLP, the number of training epochs, the learning rate, and the weight of the $L_{2}$ regularization term.
> We considered candidates for each hyperparmeter as below
>
>  -- The number of MLPs : {5, 50, 100}
>
>  -- MLP architectures : {(50), (100), (256,128,64), (512,256,128)}
>
>  -- Learning rates : {1e-4,1e-3,1e-2}
>
>  -- Epochs : {100, 200, 500, 1000}
>
>  -- Weight for $L_{2}$ regularization : {1e-3,1e-2,1e-1}
>
>
> As in the experiments of our paper, these hyperparameters were optimized using 5-fold cross-validation.
> Table C.1 shows the averages of RMSE, AUROC, CRPS, ECE and NLLs for 5 trials on real datasets.
> These results show that the performance of Bayesian-TPNN is comparable to that of Deep Ensemble ([1]) in terms of both prediction performance and uncertainty quantification.
> We included these results in Section K of the revised paper.
> We appreciate your thoughtful feedback.
>
>
>
> **Table C.1: Results of Deep Ensemble**
>
> | Dataset | RMSE/AUROC (Bayesian-TPNN) | RMSE/AUROC (Deep Ensemble) | CRPS/ECE (Bayesian-TPNN) | CRPS/ECE (Deep Ensemble) | NLL (Bayesian-TPNN) | NLL (Deep Ensemble) |
> |---------|-----------------------------|-----------------------------|---------------------------|---------------------------|----------------------|----------------------|
> | **Abalone** | **2.053 (0.26)** | 2.121 (0.23) | **1.372 (0.19)** | 1.498 (0.17) | 2.260 (0.16) | **2.036 (0.15)** |
> | **Boston** | **3.654 (0.49)** | 3.922 (0.57) | **2.202 (0.23)** | 2.458 (0.22) | **3.411 (0.37)** | 3.747 (0.40) |
> | **Mpg** | 2.386 (0.41) | **2.257 (0.14)** | 1.510 (0.43) | **1.481 (0.11)** | **2.511 (0.21)** | 2.769 (0.47) |
> | **Servo** | **0.351 (0.02)** | 0.398 (0.03) | 0.194 (0.01) | **0.179 (0.01)** | 0.836 (0.10) | **0.701 (0.04)** |
> | **FICO** | **0.793 (0.009)** | 0.773 (0.024) | **0.036 (0.004)** | 0.057 (0.033) | **0.554 (0.007)** | 0.577 (0.034) |
> | **Breast** | **0.998 (0.001)** | 0.993 (0.003) | 0.129 (0.009) | **0.075 (0.017)** | 0.211 (0.014) | **0.133 (0.041)** |
> | **Churn** | **0.849 (0.008)** | 0.841 (0.013) | **0.031 (0.001)** | 0.039 (0.002) | **0.418 (0.008)** | 0.424 (0.018) |
> | **Madelon** | **0.854 (0.013)** | 0.616 (0.029) | **0.076 (0.004)** | 0.137 (0.061) | **0.478 (0.009)** | 0.719 (0.049) |

---

> ### Author Response · Authors · 2025-11-17
>
> >**Question 3.**
> Most experiments are conducted on simulated or tabular datasets, with the exception of Section 4.4, which uses *CelebA-HQ* and *Catdog* datasets. However, in these cases, the approach relies on another CBM model to generate interpretable concepts. Could the authors apply Bayesian-TPNN directly to these datasets and produce interpretations for the low dimensional functions?
>
> **Response.**
> In principle, Bayesian-TPNN could be directly applied to image datasets such as *CelebA-HQ* or *Catdog* by flattening the image data into pixel-level tabular features.
> However, this approach would likely lead to degraded prediction performance due to the loss of spatial information, which is crucial for image understanding.
>
> However, for  image or natural language datasets, numerous backbone models capable of extracting meaningful representations have already been extensively developed and studied.(e.g., CNNs for images or Transformers for text) and we believe that the learned representations obtained by backbone models
> are composed of super higher-order interactions.
> Usually, a simple model such as a linear model is applied to the learned representations.
> Therefore, it may fail to capture higher-order interactions.
> Bayesian-TPNN is designed to operate on such low-dimensional, semantically structured representations rather than raw inputs.
> Thus, our framework is intended to be used in combination with a suitable backbone that transforms raw data into interpretable concept-level features.
>
> A representative example is the integration with the CBM, as demonstrated in this paper.
> We empirically demonstrated that coupling Bayesian-TPNN with a strong backbone (CBM) can achieve both strong prediction performance and high interpretability.
> In particular, we have already demonstrate that less representations (low number of concepts) are required when a backbone model is combined with Bayesian-TPNN in Section E.1 of the paper.

---

> ### Author Response · Authors · 2025-11-17
>
> >**Question 3.**
> Overall, it is challenging to identify practical scenarios where the proposed method would be preferred. I recommend extending the evaluation to additional domains, such as the genomics datasets used in Martens \& Yau (2020), to better demonstrate applicability.
>
> **Response.**
> As demonstrated in Sections 4.4 and E.1, Bayesian-TPNN can be effectively utilized within Concept Bottleneck Models (CBMs).
> In CBMs, generating appropriate concepts for images is challenging, and annotating these concepts for each image incurs substantial cost.
> That is, increasing the number of concepts in CBMs can improve prediction performance, but it substantially increases the associated costs.
>
> This problem can be alleviated by replacing the linear classifier with Bayesian-TPNN.
> Since Bayesian-TPNN estimates higher-order interactions among concepts, it can effectively capture—and in a sense generate—richer, composite concepts from these interactions.
> As a result, a CBM equipped with Bayesian-TPNN can achieve strong prediction performance even with fewer annotated concepts.
> The results presented in Table 22 of the paper support this claim—showing that a CBM combined with Bayesian-TPNN, even when using fewer concepts, achieves better prediction performance than a CBM with a linear classifier that using a larger number of concepts.
>
> Furthermore, inspired by the reviewer’s insightful comment, we conducted additional experiment to explore the applicability of Bayesian-TPNN to genomics dataset *GSE43358* ([2]).
> GSE43358 is a gene expression dataset with $n=57$ samples and $p=54,675$ features and we perform a classification task distinguishing between HER2-positive and non–HER2-positive cases.
> Table C.2 below shows that the averages and standard errors of prediction performance for Bayesian-TPNN, Linear model and XGB for 5 trials.
> For Bayesian-TPNN and XGB, the hyperparameters are optimized as in the paper.
> Note that because the input dimension $p$ is too large, both ANOVA-TPNN and NAM could not be trained within our computational environment.
> The results in Table C.2 indicate that the interpretable Bayesian-TPNN achieves prediction performance comparable to that of the black-box model XGB on *GSE43358* dataset.
>
> Table C.3 reports the top 10 most important components in Bayesian-TPNN with the normalized importance score.
> Here, we use the importance score defined in Section 4.2 of the paper, and the normalized score represents each importance value divided by the maximum importance score.
> Note that one of the third order interactions is detected by Bayesian-TPNN.
> The results in Table C.3 indicate that higher-order interactions (beyond the second order) play a crucial role, which provides a plausible explanation for the inferior prediction performance of the linear model.
> Moreover, this highlights the necessity of an interpretable model such as Bayesian-TPNN, which is capable of estimating such higher-order interactions.
>
> In summary, Bayesian-TPNN is applicable to domains such as genomics, credit scoring, and proteomics, where the input dimension $p$ is extremely large and understanding higher-order interactions is crucial.
> Moreover, Bayesian-TPNN can also be employed to estimate the density of high-dimensional data and thereby infer graphical models ([3]).
> We added these results to Section L of the revised manuscript. Thank you for your valuable feedback.
>
>
> **Table C.2: Results for genomic dataset**
>
> | Metric | Bayesian-TPNN | ANOVA-TPNN | NAM | Linear | XGB |
> |--------|----------------|-------------|-----|---------|------|
> | **AUROC** | 0.949 (0.017) | -- | -- | 0.545 (0.001) | 0.953 (0.041) |
>
> $\newline$
> $\newline$
> $\newline$
>
> **Table C.3: Important components**
>
> | Rank | Component of GenBank accession numbers | Normalized Score |
> |------|-----------------------------------------|------------------|
> | 1 | S69189 | 1.000 |
> | 2 | BF357738 | 0.924 |
> | 3 | (BC000129, R80390) | 0.701 |
> | 4 | AF307338 | 0.569 |
> | 5 | NM_018297 | 0.410 |
> | 6 | BF061275 | 0.375 |
> | 7 | AF319440 | 0.365 |
> | 8 | (BE741754, AB037854, AK024890) | 0.334 |
> | 9 | AI368358 | 0.292 |
> | 10 | BE672684 | 0.218 |
>
>
> **References.**
>
> [1]. Lakshminarayanan, Balaji, Alexander Pritzel, and Charles Blundell. "Simple and scalable predictive uncertainty estimation using deep ensembles." Advances in neural information processing systems 30 (2017).
>
> [2]. https://www.ncbi.nlm.nih.gov/geo/query/acc.cgi?acc=GSE43358}[https://www.ncbi.nlm.nih.gov/geo/query/acc.cgi?acc=GSE43358
>
> [3]. Jeon, Yongho, and Yi Lin. "An effective method for high-dimensional log-density anova estimation, with application to nonparametric graphical model building." Statistica Sinica (2006): 353-374.

---

### Official Review · Reviewer_Jnxk · 2025-10-31

**Soundness:** 4
**Presentation:** 4
**Contribution:** 2
**Rating:** 4
**Confidence:** 3

**Summary:**

The authors propose a Bayesian method for estimating a Tensor Product Neural Network, a method for estimating a functional ANOVA with a neural net proposed at ICML this year.
Authors develop an MCMC algo using Langevin dynamics for continuous parameters and bespoke proposals for the discrete ones.

**Strengths:**

The article is logically organized and easy to follow overall. It's clear what the goals are. From a structural perspective this piece is very well written.

The contributions of the article are clear, and the proposed methodology is thoroughly investigated.

The numerical experiments seem adequate to me; they use several datasets and compare against a reasonable suite of alternative methods.
For instance, BART is a tough competitor, and matching/beating it while providing for high interpretability is notable.

Though primarily a computation/applied article, the authors prove a basic asymptotic result of their method.

**Weaknesses:**

The biggest weakness of this article is that this work is fundamentally incremental: it is a straightforward Bayesian version of an existing method, for which the Bayesian inference is straightforward.

Some of the limitations of Bayesian inference in the neural setting, notably the lack of scalability due to the nonexistence of a stochaastic version of the MH algorithm, are not addressed.
Discussion of how to mitigate this would have improved the article.

Needs spellchecking and grammar correction.

**Questions:**

I think this work was very clearly presented and the motivation for the method itself is clear, so I don't have any questions, but I invite the authors to nevertheless answer:
1) What did I get wrong in my review?

---

> ### Author Response · Authors · 2025-11-17
>
> Thank you for your insightful comments and questions. We have carefully addressed each of your thoughtful suggestions and concerns. We have uploaded the revised version, and all modifications are highlighted in red for the convenience.
>
> > **Weakness 1.**
> The biggest weakness of this article is that this work is fundamentally incremental: it is a straightforward Bayesian version of an existing method, for which the Bayesian inference is straightforward.
>
> **Response.**
> A novel feature of Bayesian-TPNN compared to ANOVA-TPNN is its ability to learn the architecture as well as the parameters (i.e. weights and biases) while
> ANOVA-TPNN uses a fixed architecture and only learns the parameters. Ability of learning the architecture of Bayesian-TPNN makes it possible to
> search higher-order interactions which is not possible for ANOVA-TPNN due to memory and computational resources
> because the number of components to be estimated grows exponentially with the increase of interaction orders.
>
>
> Note that Bayesian-TPNN can be understood as an edge sparse one-hidden layer Bayesian neural network, where
> the number of incoming edges at each node is the order of interactions.
> A novel feature of Bayesian-TPNN is that
> the proposed MCMC algorithm can update the architecture (i.e. the number of hidden nodes) as well as the parameters (i.e. weights and biases).
> Specifically, updating $K$ is equivalent to delete or add new node into the architecture.
> When we add a new node, we devise a special step where the new edges corresponding to the new node are either completely new **Random** or adding a new edge to an existing node to introduce the new node **Stepwise**.
> The **Stepwise** step is a key tool to search higher-order interactions  since the order of interactions (i.e. the number of incoming edges) corresponding to a newly added node is always larger than the order of the old node by 1.
> Also, updating $S_{K}$ through **Adding**, **Deleting**, or **Changing** corresponds to adding, removing, or modifying edges among the given nodes.
>
>
> Empirically, as shown in Table 3 of the paper, our model efficiently detects signal higher-order interactions, and Table 25 of the paper demonstrates that, as $p$ increases, the runtime of Bayesian-TPNN becomes significantly faster than that of ANOVA-TPNN.
> These results suggest that our work does not merely present a straightforward Bayesian reformulation — it introduces a novel trans-dimensional Bayesian learning algorithm that efficiently estimates higher-order interactions.
>
>
> Figure 1 in the revised manuscript explains the architecture of Bayesian-TPNN visually.
> In addition, we added Appendix I in the revised manuscript to illustrate how the architecture evolves with emphasizing the role of the **Stepwise** step to explore higher-order interactions since the order of interactions (i.e. the number of incoming edges) corresponding to a newly added node is always larger than the order of the old node by 1.
> Finally, we emphasize the statement that Bayesian-TPNN can learn the architecture in the revised manuscript by presenting it in italics.

---

> ### Author Response · Authors · 2025-11-17
>
> > **Weakness 2.**
> Some of the limitations of Bayesian inference in the neural setting, notably the lack of scalability due to the nonexistence of a stochastic version of the MH algorithm, are not addressed. Discussion of how to mitigate this would have improved the article.
>
> **Response.**
> As far as we know, there have been many studies on stochastic versions of MCMC algorithms based on the Metropolis–Hastings (MH) framework ([1], [2], [3], [4]).
> Specifically, Stochastic Gradient Langevin Dynamics (SGLD, [1]) extends Langevin Monte Carlo to the mini-batch setting.
> They proved the acceptance probability of MH algorithm becomes 1 as the step size decreases.
> That is, the MH accept/reject step becomes nearly redundant ([1], [2]).
> In this sense, stochastic gradient MCMC methods ([1], [2], [3]) are scalable MCMC compared to the traditional MH.
>
>
> Recently, research has aimed to extend RJMCMC—which allows changes in parameter dimensionality—into a stochastic MCMC framework.
> In particular, [4] extends an MCMC algorithm that updates both discrete and continuous parameters to its stochastic version.
> Therefore, we believe that our MCMC algorithm could also be extended to a stochastic framework, and establishing such an extension would be a highly promising direction for future work.
>
>
> Instead, we conduct an additional experiment to empirically verify that our MCMC algorithm performs well when mini-batches are used.
> When estimating *Bayesian-TPNN* with mini-batched data, we refer to it as *MBayesian-TPNN*.
> Here, for *Abalone*, *FICO* datasets, we set the size of mini-batch as 1,000 and 2,000, respectively.
> Table B.1 presents the averages and standard errors of prediction performance and the uncertainty quantifications of *Bayesian-TPNN* and *MBayesian-TPNN* for 5 trials on *Abalone* and *FICO* datasets.
> These results suggest that training with mini-batches does not significantly reduce prediction performance and uncertainty quantification.
> In practice, these findings indicate that using mini-batches is practically acceptable, as it does not lead to meaningful degradation in performance or uncertainty estimation.
> We included these results in Section J of the revised paper. Thank you for your valuable comments.
>
> **Table B.1. Results of MBayesian-TPNN**
> |  | RMSE/AUROC |                          | CRPS/ECE |                                | NLL |                                  |
> |------------|----------------------|----------------------|-----------------|---------------------------|-------|----------------------------|
> |               |  **Bayesian-TPNN** | **MBayesian-TPNN** | **Bayesian-TPNN** | **MBayesian-TPNN** |**Bayesian-TPNN** | **MBayesian-TPNN** |
> | **Abalone** | 2.053 (0.26) | 2.081 (0.24) | 1.372 (0.19) | 1.391 (0.17) | 2.260 (0.16) | 2.280 (0.18) |
> | **FICO** | 0.793 (0.009) | 0.788 (0.005) | 0.036 (0.004) | 0.038 (0.003) | 0.554 (0.007) | 0.564 (0.003) |
>
> >**Question 1.**
> I think this work was very clearly presented and the motivation for the method itself is clear, so I don't have any questions, but I invite the authors to nevertheless answer: What did I get wrong in my review?
>
> **Response.**
> As mentioned in Weakness 1, this paper is not merely a Bayesian version of ANOVA-TPNN. Rather, we try to learn the architecture also. The key contribution of our work lies in the development of a specialized RJMCMC algorithm designed to overcome the limitations of ANOVA-TPNN. We also experimentally demonstrate that the proposed MCMC effectively captures higher-order interactions in high-dimensional dataset.
>
>
> It is also worth noting that designing proposals in the RJMCMC family is a particularly challenging problem ([5], [6]).
>
>
> Finally, to the best of our knowledge, this is the first Bayesian functional ANOVA model that can successfully identify higher-order interactions.
>
>
> **References.**
>
>
> + **[1]** Welling, Max, and Yee W. Teh. "Bayesian learning via stochastic gradient Langevin dynamics." Proceedings of the 28th international conference on machine learning (ICML-11). 2011.
> + **[2]** Nemeth, Christopher, and Paul Fearnhead. "Stochastic gradient markov chain monte carlo." Journal of the American Statistical Association 116.533 (2021): 433-450.
> + **[3]**  Chen, Tianqi, Emily Fox, and Carlos Guestrin. "Stochastic gradient hamiltonian monte carlo." International conference on machine learning. PMLR, 2014.
> + **[4]** Song, Qifan, et al. "Extended stochastic gradient Markov chain Monte Carlo for large-scale Bayesian variable selection." Biometrika 107.4 (2020): 997-1004.
> + **[5]** L. Davies, R. Salomone, M. Sutton, and C. Drovandi, “Transport Reversible Jump Proposals,” 26th International Conference Artificial Intelligence and Statistics (AISTATS), PMLR, vol. 206, pp. 6839–6852, 2023.
> + **[6]** Moumita Das. Sourabh Bhattacharya. "Transdimensional transformation based Markov chain Monte Carlo." Braz. J. Probab. Stat. 33 (1) 87 - 138, February 2019.

---

### Official Review · Reviewer_7sy8 · 2025-11-01

**Soundness:** 3
**Presentation:** 2
**Contribution:** 3
**Rating:** 6
**Confidence:** 2

**Summary:**

This paper proposes Bayesian Tensor Product Neural Networks (Bayesian-TPNN), a Bayesian neural network architecture for the functional ANOVA model. The key innovation is incorporating Bayesian inference over both network parameters and architectures (i.e., the subsets of input variables forming each component). This allows efficient detection of higher-order interactions without predefining component structures, addressing the scalability issue of ANOVA-TPNN (Park et al., 2025), whose number of sub-networks grows exponentially with interaction order.
The authors design a specialized MCMC algorithm that alternates between growing/pruning architectures and updating continuous parameters via Langevin proposals. Theoretically, the paper establishes posterior consistency for both the overall regression function and its individual ANOVA components.
Empirically, Bayesian-TPNN is evaluated on eight real tabular datasets, synthetic benchmarks, and concept bottleneck image tasks. It delivers comparable or superior predictive accuracy to both interpretable and black-box baselines (NAM, BART, XGB, mBNN) and significantly better uncertainty quantification and component selection, particularly for higher-order interactions.

**Strengths:**

- Clear motivation and significance: the paper tackles the long-standing scalability bottleneck of functional ANOVA neural models and provides a principled Bayesian alternative capable of learning higher-order terms without combinatorial explosion.

- Novel integration of architecture learning into ANOVA-structured modes: treating subsets of input variables as random variables and exploring them via reversible-jump-style MCMC is a clever idea. The stepwise proposal mechanism guided by feature importance is intuitive and empirically validated.

- The proof of posterior consistency for both the global function and each ANOVA component is nontrivial and strengthens the paper’s soundness.

**Weaknesses:**

- The paper spends many pages detailing MCMC updates and proofs but offers limited high-level intuition on why the proposed priors or proposal mechanisms work. As I'm not an expert of XAI and ANOVA, more discussion on background and a small synthetic visual example illustrating architecture evolution would help accessibility.

- While results are comprehensive, the gains in predictive accuracy are modest compared to ANOVA-TPNN, and the improvements in uncertainty quantification, though consistent, are small in magnitude. It would strengthen the empirical narrative to include a more demanding high-dimensional or noisy regime where the Bayesian approach truly shines.

Minors:
- Some figures (e.g., component plots) and tables are crowded or use small legends.

- The exposition occasionally repeats prior work descriptions or defers too much to appendices.

- The paper would benefit from clearer separation between methodological explanation and algorithmic detail.

**Questions:**

Could the proposed method scale to higher-order input, like p=4 or 5?

---

> ### Author Response · Authors · 2025-11-17
>
> Thank you very much for your valuable feedback and thoughtful questions. We have made
> every effort to carefully address all of your insightful comments and concerns.
> For reference, we have newly uploaded the revised paper, and the revised parts are highlighted in red.
>
> > **Weakness 1.**
> The paper spends many pages detailing MCMC updates and proofs but offers limited high-level intuition on why the proposed priors or proposal mechanisms work. As I’m not an expert of XAI and ANOVA, more discussion on background and a small synthetic visual
> example illustrating architecture evolution would help accessibility.
>
> **Response.**
> Note that Bayesian-TPNN can be understood as an edge sparse one-hidden layer Bayesian neural network, where
> the number of incoming edges at each node is the order of interactions.
> A novel feature of Bayesian-TPNN is that
> the proposed MCMC algorithm can update the architecture (i.e. the number of hidden nodes) as well as the parameters (i.e. weights and biases).
> Specifically, updating $K$ is equivalent to delete or add new node into the architecture.
> When we add a new node, we devise a special step where the new edges corresponding to the new node are either completely new ($\textbf{Random}$) or adding a new edge to an existing node to introduce the new node ($\textbf{Stepwise}$).
> The $\textbf{Stepwise}$ step is a key tool to search higher-order interactions.
> Also, updating $S_{K}$ through $\textbf{Adding}, \textbf{Deleting}$, or $\textbf{Changing}$ corresponds to adding, removing, or modifying edges among the given nodes.
>
> Figure 1 explains the architecture of Bayesian-TPNN visually and an explanation of why estimating $K$ and $S_{K}$ corresponds to estimating the architecture of the neural network is provided in lines 186–193 in the revised paper.
> Also, we added Appendix I in the revised manuscript to visually illustrate how the architecture evolves, with an emphasis on the role of the $\textbf{Stepwise}$ step in exploring higher-order interactions.
> Finally, we emphasize the statement that Bayesian-TPNN can learn the architecture in the revised manuscript by presenting it in italics.
>
> $\newline$
> $\newline$
>
> > **Weakness 2.**
> While results are comprehensive, the gains in predictive accuracy are modest compared to ANOVA-TPNN, and the improvements in uncertainty quantification, though consistent, are small in magnitude. It would strengthen the empirical narrative to include a more demanding high-dimensional or noisy regime where the Bayesian approach truly shines.
>
> **Response.**
> Our main contribution is proposing an interpretable model that achieves strong prediction performance even on datasets where higher-order interactions play an important role.
> As shown in Table 1, the prediction performance of Bayesian-TPNN is significantly superior to that of ANOVA-TPNN, NAM, and Linear models, which consider only up to second-order interactions, on the
> *Madelon* and *Servo* datasets.
>
> Table 4 of the paper provides evidence that this superior performance arises from Bayesian-TPNN's ability to effectively capture higher-order interactions.
> Specifically, for the *Madelon* dataset, Table 4 shows that the most influential interaction is of order 4.
>
> However, if NAM or ANOVA-TPNN were to estimate up to fourth-order interactions on *Madelon* dataset, the number of interactions to estimate would be 2,593,864,875, which makes such estimation computationally infeasible.
> In contrast, due to the proposed novel MCMC algorithm, Bayesian-TPNN can efficiently estimate these higher-order interactions in high-dimensional datasets, leading to significantly enhanced prediction performance and improved interpretability.

---

> ### Author Response · Authors · 2025-11-17
>
> >**Minor Weakness 1.**
> Some figures (e.g., component plots) and tables are crowded or use small legends.
>
> >**Minor Weakness 2.**
> The exposition occasionally repeats prior work descriptions or defers too much to appendices.
>
> >**Minor Weakness 3.**
> The paper would benefit from clearer separation between methodological explanation and algorithmic detail.
>
> **Response.**
> We acknowledge the reviewer’s feedback.
> Due to strict page limits, some figures and tables used smaller font sizes, and certain methodological explanations were placed in the appendix.
> In the revised version, we have newly added content in Section 3.2 in red to clarify the proposed method, and we have also included additional visual explanations of the architecture evolution in Section I.
> Due to the current page limit, we were unable to incorporate all of the points raised by the reviewer.
> However, we will make sure to address all of the reviewer’s suggestions later, including the separation between methodological explanations and algorithmic details, as well as improving the clarity and visibility of the figures and tables.
>
> $\newline$
> $\newline$
> >**Question 1.**
> Could the proposed method scale to higher-order input, like p=4 or 5?
>
> **Response.**
> Our model can efficiently identify higher-order interactions (up to order $p$) even when the input dimension $p$ is large.
> For instance, in the real-data experiment using Madelon dataset, which has 500-dimensional input features, our model successfully estimated higher-order (4th-order) interactions, as demonstrated in Table 4 of the paper.
> These results indicate that our model can effectively capture higher-order components (interactions) even in high-dimensional datasets.
> In addition, we observed that as $p$ increases, the runtime becomes significantly faster than that of ANOVA-TPNN as shown in Table 25 of the paper.
>
> Finally, we emphasize that our model is not limited to detecting up to 4th or 5th-order interactions; it is, in principle, capable of capturing interactions of even higher-orders when they are present in the data.
> To the best of our knowledge, this is the first Bayesian functional ANOVA model that can successfully identify such higher-order interactions.

---

### Meta-Review · Area_Chair_tpbe · 2025-12-13

**Summary:**

The reviewers’ main concerns centered on three aspects: (i) whether the proposed method constitutes a substantive contribution beyond existing models, (ii) whether the empirical evaluation sufficiently demonstrates clear advantages, and (iii) questions about scalability and applicability. Across reviews, there was agreement that the paper is clearly written, technically sound, and addresses an important limitation. The authors’ rebuttal and subsequent revisions provided additional clarification.

**Reviewer Concerns:**

Addressed

* The concern that the method is primarily an incremental Bayesian extension was addressed by clarifying that the proposed approach learns the model architecture itself
* Requests for higher-level intuition and improved accessibility were addressed through added visual explanations
* Concerns that predictive improvements were modest or unclear were addressed
* The absence of strong uncertainty quantification baselines was addressed by adding comparisons with deep ensembles and reporting results across multiple datasets using standard uncertainty metrics. Here authors could include scalable MCMC approaches such as Microcanonical Langevin Ensembles (Sommer et al., ICLR 2025) or their stochastic version as a baseline
* Questions regarding scalability and the lack of stochastic MH were addressed through discussion as well as new empirical results showing that mini-batch training does not substantially degrade predictive performance or uncertainty estimates
* Concerns about practical applicability were addressed by adding experiments on high-dimensional genomics data and by expanding discussion of use cases in concept bottleneck models

Unresolved

* Despite added experiments and discussion, the empirical evaluation remains primarily focused on tabular data and concept-level representations, which may limit conclusions about applicability to raw high-dimensional modalities.
* Some presentation-related issues, such as figure density and separation between methodological explanation and algorithmic detail, were only partially addressed due to page limits.
* The degree to which architectural learning via RJMCMC constitutes a sufficiently distinct contribution beyond existing methods remains a matter of interpretation, depending on how novelty is assessed.

**Reviewer Scores:**

- Reviewer 7sy8: Likely unchanged or slightly higher, given that the main requests for intuition, visualization, and stronger high-dimensional evidence were directly addressed in the revision.
- Reviewer Jnxk: Likely modestly higher, as the rebuttal directly engages with the “incremental” concern and provides both conceptual clarification and new empirical evidence regarding scalability.
- Reviewer hVKC: Likely modestly higher, given the addition of deep ensemble comparisons, genomics experiments, and clearer articulation of practical use cases, though some reservations about scope may remain.

---

### Decision · Program_Chairs · 2026-01-26

Accept (Poster)